# Methods for Convex $(L_0, L_1)$-Smooth Optimization: Clipping, Acceleration, and Adaptivity

**Eduard Gorbunov**[*]
MBZUAI

**Nazarii Tupitsa**[*]
MBZUAI
Innopolis University [†]

**Sayantan Choudhury**
Johns Hopkins University

**Alen Aliev**
MBZUAI

**Peter Richtárik**
KAUST

**Samuel Horváth**
MBZUAI

**Martin Takáč**
MBZUAI

## Abstract

Due to the non-smoothness of optimization problems in Machine Learning, generalized smoothness assumptions have been gaining a lot of attention in recent years. One of the most popular assumptions of this type is $(L_0, L_1)$-smoothness (Zhang et al., 2020b). In this paper, we focus on the class of (strongly) convex $(L_0, L_1)$-smooth functions and derive new convergence guarantees for several existing methods. In particular, we derive improved convergence rates for Gradient Descent with (Smoothed) Gradient Clipping and for Gradient Descent with Polyak Stepsizes. In contrast to the existing results, our rates do not rely on the standard smoothness assumption and do not suffer from the exponential dependency on the initial distance to the solution. We also extend these results to the stochastic case under the over-parameterization assumption, propose a new accelerated method for convex $(L_0, L_1)$-smooth optimization, and derive new convergence rates for Adaptive Gradient Descent (Malitsky & Mishchenko, 2020).

## 1 Introduction

Modern optimization problems arising in Machine Learning (ML) and Deep Learning (DL) are typically non-smooth, i.e., the gradient of the objective function is not necessarily Lipschitz continuous. In particular, the gradient of the standard $\ell_2$-regression loss computed for simple networks is not Lipschitz continuous (Zhang et al., 2020b). Moreover, the methods that are designed to benefit from the smoothness of the objective often perform poorly in Deep Learning, where problems are non-smooth. For example, variance-reduced methods (Schmidt et al., 2017; Johnson & Zhang, 2013; Defazio et al., 2014; Nguyen et al., 2017; 2021; Beznosikov & Takáč, 2021; Shi et al., 2023) are known to be faster in theory (for finite sums of smooth functions) but are outperformed by slower theoretically non-variance-reduced methods (Defazio & Bottou, 2019). All of these reasons motivate researchers to consider different assumptions to replace the standard smoothness assumption.

One such assumption is $(L_0, L_1)$-*smoothness* originally introduced by Zhang et al. (2020b) for twice differentiable functions. This assumption allows the norm of the Hessian of the objective to increase linearly with the growth of the norm of the gradient. In particular, $(L_0, L_1)$-smoothness can hold even for functions with polynomially growing gradients – a typical behavior for DL problems. Moreover, the notion of $(L_0, L_1)$-smoothness can also be extended to the class of differentiable but not necessarily twice differentiable functions (Zhang et al., 2020a; Chen et al., 2023).

Although Zhang et al. (2020b) focus on the non-convex problems as well as more recent works such as (Zhang et al., 2020a; Zhao et al., 2021; Faw et al., 2023; Wang et al., 2023; Li et al., 2024b; Chen et al., 2023; Hübler et al., 2024), the class of $(L_0, L_1)$-smooth *convex*[1] function is much weaker explored. In particular, the existing convergence results for the methods such as Gradient Descent with

---

[*]Equal contribution. Corresponding author – E. Gorbunov (`eduard.gorbunov@mbzuai.ac.ae`).

[†]Research Center of the Artificial Intelligence Institute of Innopolis University, Innopolis, Russia

[1]Although many existing problems are not convex, it is useful to understand methods behavior under the convexity assumption as well due to several reasons; see further details in Appendix A.

Clipping (Pascanu et al., 2013) and Gradient Descent with Polyak Stepsizes (Polyak, 1987) applied to $(L_0, L_1)$-smooth convex problems either rely on additional smoothness assumption (Koloskova et al., 2023; Takezawa et al., 2024) or require (potentially) small stepsizes to ensure that the method stays in the compact set where the gradient is bounded and, as a consequence of $(L_0, L_1)$-smoothness of the objective, Lipschitz continuous (Li et al., 2024a). This leads us to the following natural question:

*How the convergence bounds for different versions of Gradient Descent depend on $L_0$ and $L_1$ when the objective function is convex, $(L_0, L_1)$-smooth but not necessarily $L$-smooth?*

In this paper, we address the above question for Gradient Descent with Smoothed Gradient Clipping, Polyak Stepsizes, Similar Triangles Method (Gasnikov & Nesterov, 2016), and Adaptive Gradient Descent (Malitsky & Mishchenko, 2020): *for each of the mentioned methods, we either improve the existing convergence results or derive the first convergence results under $(L_0, L_1)$-smoothness. We also derive new results for the stochastic versions of Gradient Descent with Smoothed Gradient Clipping and Polyak Stepsizes.*

## 1.1 PROBLEM SETUP

Before we continue the discussion of the related work and our results, we need to formalize the problem setup. That is, we consider the unconstrained minimization problem

$$\min_{x \in \mathbb{R}^d} f(x), \tag{1}$$

where $f : \mathbb{R}^d \to \mathbb{R}$ is a (strongly) convex differentiable function.

**Assumption 1** (Convexity). *Function $f : \mathbb{R}^d \to \mathbb{R}$ is $\mu$-strongly convex with[2] $\mu \geqslant 0$:*

$$f(y) \geqslant f(x) + \langle \nabla f(x), y - x \rangle + \frac{\mu}{2} \|x - y\|^2, \quad \forall x, y \in \mathbb{R}^d. \tag{2}$$

As we already mentioned earlier, in addition to convexity, we assume that the objective function is $(L_0, L_1)$-smooth. Following[3] Chen et al. (2023), we consider two types of $(L_0, L_1)$-smoothness.

**Assumption 2** (Asymmetric $(L_0, L_1)$-smoothness). *Function $f : \mathbb{R}^d \to \mathbb{R}$ is asymmetrically $(L_0, L_1)$-smooth ($f \in \mathcal{L}_{asym}(L_0, L_1)$), i.e., for all $x, y \in \mathbb{R}^d$ we have*

$$\|\nabla f(x) - \nabla f(y)\| \leqslant (L_0 + L_1 \|\nabla f(y)\|) \|x - y\|. \tag{3}$$

**Assumption 3** (Symmetric $(L_0, L_1)$-smoothness). *Function $f : \mathbb{R}^d \to \mathbb{R}$ is symmetrically $(L_0, L_1)$-smooth ($f \in \mathcal{L}_{sym}(L_0, L_1)$), i.e., for all $x, y \in \mathbb{R}^d$ we have*

$$\|\nabla f(x) - \nabla f(y)\| \leqslant \left( L_0 + L_1 \sup_{u \in [x,y]} \|\nabla f(u)\| \right) \|x - y\|. \tag{4}$$

Clearly, Assumption 3 is more general than Assumtpion 2. Due to this reason, we mostly focus on Assumption 3, and by $(L_0, L_1)$-smooth functions, we mean functions satisfying Assumption 3 if the opposite is not specified. Nevertheless, it is worth mentioning that asymmetric $(L_0, L_1)$-smoothness (under some extra assumptions) is satisfied for a certain problem formulation appearing in Distributionally Robust Optimization (Jin et al., 2021). Chen et al. (2023) also show that exponential function satisfies (4), and, more generally, for twice differentiable functions Assumption 3 is equivalent to

$$\|\nabla^2 f(x)\|_2 \leqslant L_0 + L_1 \|\nabla f(x)\|, \quad \forall x \in \mathbb{R}^d. \tag{5}$$

Moreover, below, we provide some examples of functions satisfying Assumption 3 but either not satisfying standard $L$-smoothness, i.e., (4) with $L_1 = 0$, or satisfying $L$-smoothness with larger constants than $L_0$ and $L_1$ respectively. The detailed proofs are deferred to Appendix B.

**Example 1.1** (Power of Norm). *Let $f(x) = \|x\|^{2n}$, where $n$ is a positive integer. Then, $f(x)$ is convex and $(2n, 2n - 1)$-smooth. Moreover, $f(x)$ is not $L$-smooth for $n \geqslant 2$ and any $L \geqslant 0$.*

---

[2]In this paper, we consider standard $\ell_2$-norm for vectors and spectral norm for matrices.

[3]The first version of Assumptions 2 and 3 is proposed by Zhang et al. (2020a).

**Example 1.2** (Exponent of the Inner Product). *Function $f(x) = \exp(a^\top x)$ for some $a \in \mathbb{R}^d$ is convex, $(0, \|a\|)$-smooth, but not $L$-smooth for $a \neq 0$ and any $L \geqslant 0$.*

These two examples illustrate that $(L_0, L_1)$-smoothness is quite a mild assumption, and it is strictly weaker than $L$-smoothness. However, the next example shows that even when $L$-smoothness holds, it makes sense to consider $(L_0, L_1)$-smoothness as well.

**Example 1.3** (Logistic Function). *Consider logistic function: $f(x) = \log\left(1 + \exp(-a^\top x)\right)$, where $a \in \mathbb{R}^d$ is some vector. It is known that this function is $L$-smooth and convex with $L = \|a\|^2$. However, one can show that $f$ is also $(L_0, L_1)$-smooth with $L_0 = 0$ and $L_1 = \|a\|$. For $\|a\| \gg 1$, both $L_0$ and $L_1$ are much smaller than $L$.*

## 1.2 RELATED WORKS

We overview closely related works below and defer the additional discussion to Appendix A.

**Results in the non-convex case.** Zhang et al. (2020b) introduce $(L_0, L_1)$-smoothness in the form (5) and show that Clipped Gradient Descent (Clip-GD) has iteration complexity $\mathcal{O}\left(\max\left\{L_0\Delta/\varepsilon^2, (1+L_1^2)\Delta/L_0\right\}\right)$ with $\Delta := f(x) - \inf_{x \in \mathbb{R}^d} f(x)$ for finding $\varepsilon$-approximate first-order stationary point of $(L_0, L_1)$-smooth function. The asymptotically dominant term in this complexity $\mathcal{O}\left(L_0\Delta/\varepsilon^2\right)$ is independent of $L_1$, and thus, this term can be much smaller than $\mathcal{O}\left(L\Delta/\varepsilon^2\right)$, where $L$ is a Lipschitz constant of the gradient (if finite). Under the assumption that $M := \sup\{\|\nabla f(x)\| \mid x \in \mathbb{R}^d \text{ such that } f(x) \leqslant f(x^0)\} < +\infty$ Zhang et al. (2020b) also show that GD with stepsize $\Theta\left(1/(L_0+ML_1)\right)$ has complexity $\mathcal{O}\left((L_0+ML_1)\Delta/\varepsilon^2\right)$, which is natural to expect since on $\{x \in \mathbb{R}^d \mid f(x) \leqslant f(x^0)\}$ the norm of the Hessian is bounded as $L_0 + ML_1$ (see (5)), i.e., function is $(L_0 + ML_1)$-smooth. Zhang et al. (2020a) generalize the results from (Zhang et al., 2020b) to the method with heavy-ball momentum (Polyak, 1964) and clipping of both momentum and gradient. Similar results are derived for Normalized GD (Zhao et al., 2021; Chen et al., 2023), SignGD (Crawshaw et al., 2022), AdaGrad-Norm/AdaGrad (Faw et al., 2023; Wang et al., 2023), Adam (Wang et al., 2022; Li et al., 2024b), and Normalized GD with Momentum (Hübler et al., 2024). Notably, all papers in this paragraph also address stochastic method versions.

**Results in the convex case.** To the best of our knowledge, convex $(L_0, L_1)$-smooth optimization is studied in three papers[4] (Koloskova et al., 2023; Takezawa et al., 2024; Li et al., 2024a). In particular, under convexity, $L$-smoothness, and $(L_0, L_1)$-smoothness, Koloskova et al. (2023) show that Clip-GD with clipping level $c$ has $\mathcal{O}\left(\max\left\{(L_0+cL_1)R_0^2/\varepsilon, \sqrt{R_0^4 L(L_0+cL_1)^2/c^2\varepsilon}\right\}\right)$ complexity of finding $\varepsilon$-solution, i.e., $x$ such that $f(x) - f(x^*) \leqslant \varepsilon$, where $x^* \in \arg\min_{x \in \mathbb{R}^d} f(x)$ and $R_0 := \|x^0 - x^*\|$. In particular, if $c \sim L_0/L_1$, then the asymptotically dominant term in the complexity is $\mathcal{O}\left(L_0 R_0^2/\varepsilon\right)$, i.e., it is independent of $L_1$ and $L$, which can be significantly better than the complexity of GD of $\mathcal{O}\left(LR_0^2/\varepsilon\right)$ for convex $L$-smooth functions. In the same setting, Takezawa et al. (2024) prove $\mathcal{O}\left(\max\left\{L_0 R_0^2/\varepsilon, \sqrt{R_0^4 L L_1^2/\varepsilon}\right\}\right)$ complexity bound for GD with Polyak Stepsizes (GD-PS). Finally, under convexity and $(L_0, L_1)$-smoothness Li et al. (2024a) show that for sufficiently small stepsizes standard GD and Nesterov's method (NAG) (Nesterov, 1983) have complexities $\mathcal{O}\left(\ell R_0^2/\varepsilon\right)$ and $\mathcal{O}\left(\sqrt{\ell R_0^2/\varepsilon}\right)$ respectively, where $\ell := L_0 + L_1 G$ and $G$ is some constant depending on $L_0, L_1, R_0, \|\nabla f(x^0)\|$, and $f(x^0) - f(x^*)$. In particular, constant $G$ and stepsizes are chosen in such a way that it is possible to show via induction that in all points generated by GD/NAG and where $(L_0, L_1)$-smoothness is used the norm of the gradient is bounded by $G$. However, these results have a common limitation: constants $L$ (if finite) and $\ell$ can be much larger than $L_0$ and $L_1$. Moreover, for Clip-GD and GD-PS, these results lead to a natural question of whether it is possible to achieve $\mathcal{O}\left(LR_0^2/\varepsilon\right)$ complexity without $L$-smoothness non-asymptotically.

## 1.3 OUR CONTRIBUTION

- **Tighter rates for Gradient Descent with (Smoothed) Clipping.** We prove that Gradient Descent with (Smoothed) Clipping, which we call $(L_0, L_1)$-GD, has $\mathcal{O}\left(\max\left\{L_0 R_0^2/\varepsilon, L_1^2 R_0^2\right\}\right)$ worst-case

---

[4]After the first version of our paper appeared online, another highly-relevant paper appeared online (Vankov et al., 2024). In particular, Vankov et al. (2024) independently derive similar results to ours for $(L_0, L_1)$-GD and GD-PS, and also obtained new convergence bounds for Normalized GD and improved accelerated rates.

complexity of finding $\varepsilon$-solution for convex $(L_0, L_1)$-smooth functions. In contrast to the previous results (Koloskova et al., 2023; Li et al., 2024a), our bound is derived without $L$-smoothness assumption and does not depend on any bound for $\|\nabla f(x^k)\|$. To achieve this, we prove that $(L_0, L_1)$-GD has non-increasing gradient norm and show that the method's behavior consists of two phases: initial (and finite) phase when $\|\nabla f(x^k)\| \geqslant L_0/L_1$ (large gradient), and final phase when $\|\nabla f(x^k)\| < L_0/L_1$ and the method behaves similarly to GD applied to $2L_0$-smooth problem. We also extend the result to the strongly/stochastic convex cases.

- **Tighter rates for Gradient Descent with Polyak Stepsizes.** For GD-PS, we also derive $\mathcal{O}\left(\max\left\{L_0 R_0^2/\varepsilon, L_1^2 R_0^2\right\}\right)$ worst-case complexity of finding $\varepsilon$-solution for convex $(L_0, L_1)$-smooth functions. In contrast to the existing result (Takezawa et al., 2024), our bound is derived without $L$-smoothness assumption. We also extend the result to the strongly/stochastic convex cases.

- **New accelerated method: $(L_0, L_1)$-Similar Triangles Method.** We propose a version of Similar Triangles Method (Gasnikov & Nesterov, 2016) for convex $(L_0, L_1)$-smooth optimization, and prove $\mathcal{O}\left(\sqrt{L_0(1+L_1 R_0 \exp(L_1 R_0))R_0^2/\varepsilon}\right)$ complexity of finding $\varepsilon$-solution for convex $(L_0, L_1)$-smooth functions. In contrast to the accelerated result from (Li et al., 2024a), our bound is derived without the usage of stepsizes depending on $R_0$ and $f(x^0) - f(x^*)$.

- **New convergence results for Adaptive Gradient Descent.** We also show new convergence result for Adaptive Gradient Descent (Malitsky & Mishchenko, 2020) for convex $(L_0, L_1)$-smooth problems: we prove $\mathcal{O}\left(\max\left\{L_0 D^2/\varepsilon, m^2(L_1^2 D^2 + L_1^4 D_1^4)\right\}\right)$ complexity of finding $\varepsilon$-solution, where $D$ is a constant depending on initial suboptimality of the starting point, and $m$ is a logarithmic factor depending on $L_1$ and $D$. We also extend the result to the strongly convex case.

- **New technical results for $(L_0, L_1)$-smooth functions.** We derive several useful inequalities for the class of (convex) $(L_0, L_1)$-smooth functions.

## 2 TECHNICAL LEMMAS

In this section, we provide some useful facts about $(L_0, L_1)$-smooth functions. We start with the following result from (Chen et al., 2023).

**Lemma 2.1** (Proposition 1 from (Chen et al., 2023)). *Assumption 3 holds if and only if for*

$$\|\nabla f(x) - \nabla f(y)\| \leqslant (L_0 + L_1\|\nabla f(y)\|)\exp(L_1\|x - y\|)\|x - y\|, \quad \forall x, y \in \mathbb{R}^d. \quad (6)$$

*Moreover, Assumption 3 implies for all $x, y \in \mathbb{R}^d$*

$$f(y) \leqslant f(x) + \langle \nabla f(x), y - x \rangle + \frac{L_0 + L_1\|\nabla f(x)\|}{2}\exp(L_1\|x - y\|)\|x - y\|^2. \quad (7)$$

Inequality (6) removes the supremum from (4), but the price for this is a factor of $\exp(L_1\|x - y\|)$. When $\|x - y\| \leqslant 1/L_1$, this factor is upper-bounded as $e$. However, in general, it cannot be removed since (6) is equivalent to (4). Inequality (7) can be seen as a generalization of standard quadratic upper-bound for $L$-smooth functions (Nesterov, 2018) to the class of $(L_0, L_1)$-smooth functions. Note, that (Zhang et al., 2020a) provides a Hessian free assumption which is equivalent to (5), it can be seen as (6) and (7) with improved constants.

Using the above lemma, we derive several useful inequalities that we actively use throughout our proofs. Most of these inequalities can be further simplified in the case of Assumption 2.

**Lemma 2.2.** *Let Assumption 3 hold and $\nu$ satisfy[5] $\nu = e^{-\nu}$. Then, the following statements hold.*

1. *For $f_* := \inf_{x \in \mathbb{R}^d} f(x)$ and arbitrary $x \in \mathbb{R}^d$, we have*

$$\frac{\nu\|\nabla f(x)\|^2}{2(L_0 + L_1\|\nabla f(x)\|)} \leqslant f(x) - f_*. \quad (8)$$

2. *If additionally Assumption 1 holds with $\mu = 0$, then for any $x, y \in \mathbb{R}^d$ such that*

$$L_1\|x - y\|\exp(L_1\|x - y\|) \leqslant 1, \quad (9)$$

---

[5]One can check numerically that $0.56 < \nu < 0.57$.

*we have*

$$\frac{\nu\|\nabla f(x) - \nabla f(y)\|^2}{2(L_0 + L_1\|\nabla f(y)\|)} \leqslant f(y) - f(x) - \langle \nabla f(x), y - x\rangle, \tag{10}$$

*and*

$$\frac{\nu\|\nabla f(x) - \nabla f(y)\|^2}{2(L_0 + L_1\|\nabla f(y)\|)} + \frac{\nu\|\nabla f(x) - \nabla f(y)\|^2}{2(L_0 + L_1\|\nabla f(x)\|)} \leqslant \langle \nabla f(x) - \nabla f(y), x - y\rangle. \tag{11}$$

This lemma provides us with a set of useful inequalities that can be viewed as generalizations of analogous inequalities that hold for smooth (convex) functions. We provide the complete proof in Appendix C. Moreover, when Assumption 2 holds, all inequalities from Lemma 2.2 hold with $\nu = 1$, and requirement (9) is not needed for (10) and (11) to hold. An analog of (8) for a local version of $(L_0, L_1)$-smoothness can be found in (Koloskova et al., 2023). We also refer to (Li et al., 2024a) for an analog of inequality (11) for $(r, \ell)$-smooth functions. However, in contrast to the bound from (Koloskova et al., 2023), bound (8) is derived for a global version of $(L_0, L_1)$-smoothness and thus differs in numerical constants, and, in contrast to the proof from (Li et al., 2024a), we do not use local Lipshitzness of the gradient.

## 3 Smoothed Gradient Clipping

The first method that we consider is closely related to Clip-GD and can be seen as a smoothed version[6] of it – see Algorithm 1. Alternatively, this method can be seen as a version of Gradient Descent designed for $(L_0, L_1)$-smooth functions. Therefore, we call this algorithm $(L_0, L_1)$-GD.

---

**Algorithm 1** $(L_0, L_1)$-Gradient Descent ($(L_0, L_1)$-GD)

---

**Input:** starting point $x^0$, number of iterations $N$, stepsize parameter $\eta > 0$, $L_0 > 0$, $L_1 \geqslant 0$
 1: **for** $k = 0, 1, \ldots, N - 1$ **do**
 2:  $x^{k+1} = x^k - \frac{\eta}{L_0 + L_1\|\nabla f(x^k)\|}\nabla f(x^k)$
 3: **end for**
**Output:** $x^N$

---

Similarly to standard GD, $(L_0, L_1)$-GD satisfies two useful properties, summarized below.

**Lemma 3.1** (Monotonicity of function value). *Let Assumption 3 hold. Then, for all $k \geqslant 0$ the iterates generated by $(L_0, L_1)$-GD with $\eta \leqslant \nu$, $\nu = e^{-\nu}$ satisfy*

$$f(x^{k+1}) \leqslant f(x^k) - \frac{\eta\|\nabla f(x^k)\|^2}{2(L_0 + L_1\|\nabla f(x^k)\|)} \leqslant f(x^k). \tag{12}$$

*Proof sketch.* The inequality follows from (7) applied to $y = x^{k+1}$ and $x = x^k$, see the complete proof in Appendix D. □

**Lemma 3.2** (Monotonicity of gradient norm). *Let Assumptions 1 with $\mu = 0$ and 3 hold. Then, for all $k \geqslant 0$ the iterates generated by $(L_0, L_1)$-GD with $\eta \leqslant \nu$, $\nu = e^{-\nu}$ satisfy*

$$\|\nabla f(x^{k+1})\| \leqslant \|\nabla f(x^k)\|. \tag{13}$$

*Proof sketch.* The inequality follows from (11) applied to $x = x^{k+1}$ and $y = x^k$, see the complete proof in Appendix D. □

We notice that a similar result to Lemma 3.2 is shown in (Li et al., 2024a) for GD with sufficiently small stepsize. With these lemmas in hand, we derive the convergence result for $(L_0, L_1)$-GD.

---

[6]Indeed, when $\|\nabla f(x^k)\| < L_0/L_1$, the denominator of the stepsize in $(L_0, L_1)$-GD lies in $[L_0, 2L_0]$, and when $\|\nabla f(x^k)\| \geqslant L_0/L_1$, this denominator lies in $[L_1\|\nabla f(x^k)\|, 2L_1\|\nabla f(x^k)\|]$. Such a behavior is very similar to the behavior of Clip-GD with clipping level $L_0/L_1$ and stepsize $\eta/L_0$.

**Theorem 3.1.** *Let Assumptions 1 with $\mu = 0$ and 3 hold. Then, the iterates generated by $(L_0, L_1)$-GD with $0 < \eta \leqslant \frac{\nu}{2}$, $\nu = e^{-\nu}$ satisfy the following implication:*

$$\|\nabla f(x^k)\| \geqslant \frac{L_0}{L_1} \implies k \leqslant \frac{8L_1^2\|x^0 - x^*\|^2}{\nu\eta} - 1 \;\; and \;\; \|x^{k+1} - x^*\|^2 \leqslant \|x^k - x^*\|^2 - \frac{\nu\eta}{8L_1^2}. \quad (14)$$

*Moreover, the output after $N > \frac{8L_1^2\|x^0 - x^*\|^2}{\eta} - 1$ iterations satisfies*

$$f(x^N) - f(x^*) \leqslant \frac{2L_0\|x^0 - x^*\|^2}{\eta(N + 1 - T)} - \frac{\nu L_0 T}{4L_1^2(N + 1 - T)} \leqslant \frac{2L_0\|x^0 - x^*\|^2}{\eta(N + 1)}, \quad (15)$$

*where $T := |\mathcal{T}|$ for the set $\mathcal{T} := \{k \in \{0, 1, \dots N - 1\} \mid \|\nabla f(x^k)\| \geqslant \frac{L_0}{L_1}\}$.*

*Proof sketch.* Similarly to the proofs from (Koloskova et al., 2023; Takezawa et al., 2024), our proof is based on careful consideration of two possible situations: either $\|\nabla f(x^k)\| \geqslant L_0/L_1$ or $\|\nabla f(x^k)\| < L_0/L_1$. When the first situation happens, the squared distance to the solution decreases by $\eta/8L_1^2$. Since the squared distance is non-negative and non-increasing, this cannot happen more than $8L_1^2\|x^0 - x^*\|^2/\nu\eta$ times, which gives the first part of the result. Next, when $\|\nabla f(x^k)\| < L_0/L_1$, the method behaves as GD on convex $2L_0$-smooth problem and the analysis is also similar. Together with Lemmas 3.1 and 3.2, this gives the second part of the proof, see Appendix D for the details. □

Bound (15) implies that $(L_0, L_1)$-GD with $\eta = \nu/2$ satisfies $f(x^N) - f(x^*) \leqslant \varepsilon$ after $N = \mathcal{O}\left(\max\left\{L_0 R_0^2/\varepsilon, L_1^2 R_0^2\right\}\right)$ iterations. In contrast, Koloskova et al. (2023); Takezawa et al. (2024) derive $\mathcal{O}\left(\max\left\{L_0 R_0^2/\varepsilon, \sqrt{R_0^4 LL_1^2/\varepsilon}\right\}\right)$ complexity bound that depends on the smoothness constant $L$, which can be much larger than $L_0$ and $L_1$, e.g., when $f(x) = \|x\|^4$ constant $L$ depends on the starting point (since it defines a compact set, where the method stays) as $L_0 + L_1\|\nabla f(x^0)\| = \mathcal{O}(1 + \|x^0\|^3)$ (see Appendix B), while $L_0 = 4$ and $L_1 = 3$. That is, by moving $x^0$ away from the solution, one can make our bound arbitrarily better than the previous one, even for this simple example. Moreover, unlike the result from (Li et al., 2024a) for GD with small enough stepsize, our bound depends neither on $f(x^0) - f(x^*)$ nor on $\|\nabla f(x^0)\|$ that can be significantly larger than $R_0$ (according to Lemma 2.1 – exponentially larger). Finally, we highlight that our analysis shows that $(L_0, L_1)$-GD exhibits a two-stage behavior: during the first stage, the gradient is large (this stage can be empty), and the squared distance to the solution decreases by a constant, and during the second stage, the method behaves as standard GD. This observation is novel on its own and gives a better understanding of the method's behavior. We also provide the result for the strongly convex case in Appendix D.

## 4 GRADIENT DESCENT WITH POLYAK STEPSIZES

Next, we provide an improved analysis under $(L_0, L_1)$-smothness for celebrated Gradient Descent with Polyak Stepsizes (GD-PS, Algorithm 2).

---

**Algorithm 2** Gradient Descent with Polyak Stepsizes (GD-PS)

---

**Input:** starting point $x^0$, number of iterations $N$, minimal value $f(x^*) := \min_{x \in \mathbb{R}^d} f(x)$
1: **for** $k = 0, 1, \dots, N - 1$ **do**
2: $\quad x^{k+1} = x^k - \frac{f(x^k) - f(x^*)}{\|\nabla f(x^k)\|^2} \nabla f(x^k)$
3: **end for**
**Output:** $x^N$

---

**Theorem 4.1.** *Let Assumptions 1 with $\mu = 0$ and 3 hold. Then, the iterates generated by GD-PS satisfy the following implication:*

$$\|\nabla f(x^k)\| \geqslant \frac{L_0}{L_1} \implies \|x^{k+1} - x^*\|^2 \leqslant \|x^k - x^*\|^2 - \frac{\nu^2}{16L_1^2}. \quad (16)$$

*Moreover, the output after $N$ steps the iterates satisfy*

$$\frac{4L_0}{\nu}\|x^{N+1} - x^*\|^2 + \sum_{k \in \{0, 1, \dots, N\} \setminus \mathcal{T}} \left(f(x^k) - f(x^*)\right) \leqslant \frac{4L_0}{\nu}\|x^0 - x^*\|^2 - \frac{\nu L_0 T}{4L_1^2}, \quad (17)$$

where $\mathcal{T} := \{k \in \{0, 1, \ldots, N\} \mid \|\nabla f(x^k)\| \geqslant \frac{L_0}{L_1}\}$, $T := |\mathcal{T}|$, and if $N > T - 1$, it holds that

$$f(\hat{x}^N) - f(x^*) \leqslant \frac{4L_0\|x^0 - x^*\|^2}{\nu(N - T + 1)} - \frac{\nu L_0 T}{4L_1^2(N - T + 1)} \tag{18}$$

where $\hat{x}^N \in \{x^0, x^1, \ldots, x^N\}$ is such that $f(\hat{x}^N) = \min_{x \in \{x^0, x^1, \ldots, x^N\}} f(x)$. In particular, for $N > \frac{16L_1^2\|x^0 - x^*\|^2}{\nu^2} - 1$ inequality $N > T - 1$ is guaranteed and

$$f(\hat{x}^N) - f(x^*) \leqslant \frac{4L_0\|x^0 - x^*\|^2}{\nu(N + 1)}. \tag{19}$$

*Proof sketch.* The proof is similar to the one for $(L_0, L_1)$-GD, see the details in Appendix E. □

In other words, the above result shows that GD-PS has the same worst-case complexity as $(L_0, L_1)$-GD, and the comparison with the results from (Koloskova et al., 2023; Takezawa et al., 2024; Li et al., 2024a) that we provided after Theorem 3.1 is valid for GD-PS as well. However, in contrast to $(L_0, L_1)$-GD, GD-PS requires to know $f(x^*)$ only. In some cases, the optimal value is known in advance, e.g., for over-parameterized problems (Vaswani et al., 2019a) $f(x^*) = 0$. In such situations GD-PS can be called parameter-free. The price for this is the potential non-monotonic behavior of GD-PS, which we observed in our preliminary computer-aided analysis using PEPit (Goujaud et al., 2024) even in the case of $L$-smooth functions. Therefore, unlike Theorem 3.1, Theorem 4.1 does not provide last-iterate convergence rates in the convex case and also does not imply that GD-PS has a clear two-stage behavior (although the iterates can be split into two groups based on the norm of the gradient as well). We also provide the result for the strongly convex case in Appendix E.

## 5 ACCELERATION: $(L_0, L_1)$-SIMILAR TRIANGLES METHOD

In this section, we present an accelerated version of $(L_0, L_1)$-GD called $(L_0, L_1)$-Similar Triangles Method ($(L_0, L_1)$-STM, Algorithm 3). This method can be seen as an adaptation of STM (Gasnikov & Nesterov, 2016) to the case of $(L_0, L_1)$-smooth functions. The main modification in comparison to the standard STM is in Line 5: stepsize for GD-type step is now proportional to $1/G_{k+1}$, where $G_{k+1}$ is some upper bound on $L_0 + L_1\|\nabla f(x^{k+1})\|$, while in STM $G_{k+1}$ should be an upper bound for the smoothness constant.

---

**Algorithm 3** $(L_0, L_1)$-Similar Triangles Method ($(L_0, L_1)$-STM)

**Input:** starting point $x^0$, number of iterations $N$, stepsize parameter $\eta > 0$
1: $y^0 = z^0 = x^0$, $A_k = 0$
2: **for** $k = 0, 1, \ldots, N - 1$ **do**
3:     Set $\alpha_{k+1} = \frac{\eta(k+2)}{2}$ and $A_{k+1} = A_k + \alpha_{k+1}$
4:     $x^{k+1} = \frac{A_k y^k + \alpha_{k+1} z^k}{A_{k+1}}$
5:     $z^{k+1} = z^k - \frac{\alpha_{k+1}}{G_{k+1}}\nabla f(x^{k+1})$, where $G_{k+1} \geqslant L_0 + L_1\|\nabla f(x^{k+1})\|$
6:     $y^{k+1} = \frac{A_k y^k + \alpha_{k+1} z^{k+1}}{A_{k+1}}$
7: **end for**
**Output:** $y^N$

---

The next lemma is valid for any choice of $G_{k+1} \geqslant L_0 + L_1\|\nabla f(x^{k+1})\|$.

**Lemma 5.1.** *Let $f$ satisfy Assumptions 1 with $\mu = 0$ and 3. Then, the iterates generated by $(L_0, L_1)$-STM with $0 < \eta \leqslant \frac{\nu}{2}$, $\nu = e^{-\nu}$ satisfy for all $N \geqslant 0$*

$$A_N\left(f(y^N) - f(x^*)\right) + \frac{G_N}{2}R_N^2 \leqslant \underbrace{\frac{G_1}{2}R_0^2 + \sum_{k=1}^{N-1} \frac{G_{k+1} - G_k}{2}R_k^2}_{(20)} - \underbrace{\sum_{k=0}^{N-1} \frac{\alpha_{k+1}^2}{4G_{k+1}}\|\nabla f(x^{k+1})\|^2}_{(21)},$$

*where $R_k := \|z^k - x^*\|$ for all $k \geqslant 0$.*

Since $A_N \geqslant \frac{\eta N(N+3)}{4}$ (see Lemma F.1) and the term from (21) is non-positive, the above lemma gives an accelerated convergence rate, *if we manage to bound the sum from* (20). Unfortunately, in the case of $G_{k+1} = L_0 + L_1 \|\nabla f(x^{k+1})\|$, it is unclear whether this sum is bounded due to the well-known non-monotonic behavior (in particular, in terms of the gradient norm) of accelerated methods. Nevertheless, if we enforce $G_{k+1}$ to be non-decreasing as a function of $k$, then from the above lemma one can show that $R_k$ remains bounded by $R_0$ and all iterates generated by $(L_0, L_1)$-STM lie in the ball centered at $x^*$ with radius $R_0$. This observation is formalized in the theorem below (see the complete proof in Appendix F).

**Theorem 5.1.** *Let $f$ satisfy Assumptions 1 with $\mu = 0$ and 3. Then, the iterates generated by $(L_0, L_1)$-STM with $0 < \eta \leqslant \frac{\nu}{2}$, $\nu = e^{-\nu}$, $G_1 = L_0 + L_1 \|\nabla f(x^0)\|$, and*

$$G_{k+1} = \max\{G_k, L_0 + L_1 \|\nabla f(x^{k+1})\|\}, \quad k \geqslant 0, \tag{22}$$

*satisfy*

$$f(y^N) - f(x^*) \leqslant \frac{2L_0(1 + L_1 \|x^0 - x^*\| \exp(L_1 \|x^0 - x^*\|)) \|x^0 - x^*\|^2}{\eta N(N+3)}. \tag{23}$$

In the special case of $L_0$-smooth functions ($L_1 = 0$), the above result recovers the standard accelerated convergence rate (Gasnikov & Nesterov, 2016). In the general $(L_0, L_1)$-smooth case, the rate is also accelerated and implies an optimal $\mathcal{O}\left(\sqrt{L_0(1+L_1 R_0 \exp(L_1 R_0)) R_0^2/\varepsilon}\right)$ in $\varepsilon$ complexity. In the case of $(L_0, L_1)$-smooth functions, the complexity $\mathcal{O}\left(\sqrt{\ell R_0^2/\varepsilon}\right)$ from (Li et al., 2024a) derived for Nesterov's method applied to convex $(r, \ell)$-smooth problem coincides with our result in the worst-case. Indeed, in this special case, $\ell = L_0 + 2L_1 G$, where $G \sim \|\nabla f(x^0)\|$ (Li et al., 2024a, Theorem 4.4). However, according to Lemma 2.1, $\|\nabla f(x^0)\| \sim L_0 R_0 \exp(L_1 R_0)$ in the worst case, implying that $\ell \sim L_0(1 + 2L_1 R_0 \exp(L_1 R_0))$ in the worst case. Nevertheless, the derived complexity is clearly not optimal if $L_1$ is large, $R_0$ is large, and $\varepsilon$ is not too small since $\sqrt{L_0(1+L_1 R_0 \exp(L_1 R_0)) R_0^2/\varepsilon}$ can be larger than $\max\left\{L_0 R_0^2/\varepsilon, L_1^2 R_0^2\right\}$, i.e., $(L_0, L_1)$-GD and GD-PS can be faster in achieving $\varepsilon$-solutiion for some values of $L_1$, $R_0$, and $\varepsilon$. Deriving a tight lower bound and optimal method for convex $(L_0, L_1)$-smooth optimization remains an open problem.

# 6  ADAPTIVE GRADIENT DESCENT

In this section, we consider Adaptive Gradient Descent (AdGD, Algorithm 4) proposed by Malitsky & Mishchenko (2020). In the original paper, the method is analyzed under the assumption that the gradient of $f$ is locally Lipschitz, i.e., for any compact set $\mathcal{C}$ gradient of $f$ is assumed to be bounded. Clearly, $(L_0, L_1)$-smoothness of $f$ implies that $\nabla f$ is locally Lipschitz, e.g., this can be deduced from (6). In particular, Malitsky & Mishchenko (2020) prove $\mathcal{O}(LD^2/N)$ convergence rate for AdGD, where $L$ is smoothness constant on the convex combination of $\{x^*, x^0, x^1, \ldots\}$: this set is bounded since the authors prove that AdGD does not leave ball centered at $x^*$ with radius $D > 0$ such that $D^2 := \|x^1 - x^*\|^2 + \frac{1}{2}\|x^1 - x^0\|^2 + 2\lambda_1 \theta_1(f(x^0) - f(x^*))$. Moreover, they derive $\|x^k - x^{k-1}\|^2 \leqslant 2D^2$ for all $k \geqslant 1$.

---

**Algorithm 4** Adaptive Gradient Descent (Malitsky & Mishchenko, 2020)

---

1: **Input:** $x^0 \in \mathbb{R}^d$, $\lambda_0 > 0$, $\theta_0 = +\infty$, $\gamma \leqslant \frac{1}{2}$
2: $x^1 = x^0 - \lambda_0 \nabla f(x^0)$
3: **for** $k = 1, 2, \ldots$ **do**
4: $\quad \lambda_k = \min\left\{\sqrt{1 + \theta_{k-1}} \lambda_{k-1}, \frac{\gamma \|x^k - x^{k-1}\|}{\|\nabla f(x^k) - \nabla f(x^{k-1})\|}\right\}$
5: $\quad x^{k+1} = x^k - \lambda_k \nabla f(x^k)$
6: $\quad \theta_k = \frac{\lambda_k}{\lambda_{k-1}}$
7: **end for**

---

In the case of $(L_0, L_1)$-smoothness, constant $L$ can be estimated explicitly: in view of the mentioned upper bounds on $\|x^k - x^*\|$ and $\|x^k - x^{k-1}\|$, we have

$$\|\nabla f(x^k)\| \overset{(6)}{\leqslant} L_0 \exp(L_1 \|x^k - x^*\|) \|x^k - x^*\| \leqslant L_0 \exp(L_1 D) D, \tag{24}$$

which allows us to lower-bound $\frac{\gamma\|x^k - x^{k-1}\|}{\|\nabla f(x^k) - \nabla f(x^{k-1})\|}$ and $\lambda_k$ as $\frac{\gamma}{L_0(1 + L_1 D \exp{(L_1 D)}) \exp{(\sqrt{2}L_1 D)}}$ for all $k \geqslant 1$. Then, following the proof by Malitsky & Mishchenko (2020), we get the following rate (for the definition of $\hat{x}^N$ and the detailed statement of the result we refer to Appendix G):

$$f(\hat{x}^N) - f(x^*) \leqslant \frac{L_0(1 + L_1 D \exp{(L_1 D)}) \exp{(\sqrt{2}L_1 D)}D^2}{N}. \tag{25}$$

Although this result shows that AdGD has the same *rate* $1/N$ of convergence for smooth and $(L_0, L_1)$-smooth functions, constant $(1 + L_1 D \exp{(L_1 D)}) \exp{(\sqrt{2}L_1 D)}$ appearing in the upper bound can be huge. To address this issue, we derive a refined convergence result for AdGD.

**Theorem 6.1.** *Let Assumptions 1 with $\mu = 0$ and 3 hold. For all $N \geqslant 1$ we define point $\hat{x}^N := \frac{1}{S_N}\left(\lambda_N(1 + \theta_N) + \sum_{k=1}^{N} w_k x^k\right)$, where $w_k := \lambda_k(1 + \theta_k) - \lambda_{k+1}\theta_{k+1}$, $S_N := \lambda_1\theta_1 + \sum_{k=1}^{N} \lambda_k$, and $\{x^k\}_{k\geqslant 0}$ are the iterates produced by AdGD with $\gamma = 1/4$. Then, for $N > mK - \frac{\sqrt{2N}(m+1)L_1 D}{\nu}$ iterate $\hat{x}^N$ satisfies*

$$f(\hat{x}^N) - f(x^*) \leqslant \frac{2L_0 D^2}{\nu(N - mK) - \sqrt{2N}(m+1)L_1 D}, \tag{26}$$

*where $D > 0$ and $D^2 := \|x^1 - x^*\|^2 + \frac{3}{4}\|x^1 - x^0\|^2 + 2\lambda_1\theta_1(f(x^0) - f(x^*))$, $m := 1 + \log_{\sqrt{2}}\left\lceil \frac{(1 + L_1 D \exp{(2L_1 D)})}{2}\right\rceil$, $K := \frac{2L_1^2 D^2}{\nu^2}$, and $\nu = e^{-\nu}$. In particular, for $N \geqslant \left(2mK + \frac{4(m+1)L_1 D}{\nu}\right)^2$, we have*

$$f(\hat{x}^N) - f(x^*) \leqslant \frac{4L_0 D^2}{\nu N}. \tag{27}$$

The above states that AdGD converges at least $\sim (1 + L_1 D \exp{(L_1 D)}) \exp{(\sqrt{2}L_1 D)}$ faster (for sufficiently large $N$) than the upper bound from (25). This is a noticeable factor: if $L_1 = 1$, $D = 10$, it is of the order $8 \cdot 10^6$. Moreover, in contrast to (25), Theorem 6.1 does not follow its counterpart from Malitsky & Mishchenko (2020). To achieve it, we use $\gamma = 1/4$ to get a new potential function

$$\Phi_k := \|x^k - x^*\|^2 + \frac{1}{4}\|x^k - x^{k-1}\|^2 + 2\lambda_k\theta_k(f(x^{k-1}) - f(x^*)) + \frac{1}{2}\sum_{i=0}^{k-1}\|x^{i+1} - x^i\|^2, \tag{28}$$

and show that $\Phi_{k+1} \leqslant \Phi_k \ \forall k \geqslant 1$. In contrast, the potential function from (Malitsky & Mishchenko, 2020) does not have term $\frac{1}{2}\sum_{i=0}^{k-1}\|x^{i+1} - x^i\|^2$, which is the key for obtaining a better guarantee under $(L_0, L_1)$-smoothness. Finally, we also note that up to the numerical factors bound (27) coincides with the bound from (Malitsky & Mishchenko, 2020) derived for $L$-smooth problems. Moreover, up to the numerical factors and the replacement of $\|x^0 - x^*\|^2$ with $D^2$, which might be larger than $\|x^0 - x^*\|^2$, our bound (27) for AdGD coincides with the ones derived for $(L_0, L_1)$-GD and GD-PS. These facts highlight the adaptivity of AdGD in theory.

## 7 STOCHASTIC EXTENSIONS

In this section, we consider the finite-sum minimization problem, i.e., we assume that $f(x) := \frac{1}{n}\sum_{i=1}^{n} f_i(x)$. Problems of this type are typical for ML applications (Shalev-Shwartz & Ben-David, 2014), where $f_i(x)$ represents the loss function evaluated for $i$-th example in the dataset and $x$ are parameters of the model. Since the size of the dataset $n$ is usually large, stochastic first-order methods such as Stochastic Gradient Descent (Robbins & Monro, 1951) are the methods of choice for this class of problems. However, to proceed, we need to impose some assumptions on functions $\{f_i\}_{i=1}^n$.

**Assumption 4.** *For all $i = 1, \ldots, n$ function $f_i$ is convex and symmetrically $(L_0, L_1)$-smooth, i.e., inequalities (2) with $\mu = 0$ and (4) for function $f_i$ as well. Moreover, we assume that there exists $x^* \in \mathbb{R}^d$ such that $x^* \in \arg\min_{x \in \mathbb{R}^d} f_i(x)$ for all $i = 1, \ldots, n$, i.e., functions $\{f_i\}_{i=1}^n$ have a common minimizer.*

The first part of the assumption (convexity and $(L_0, L_1)$-smoothness of all $\{f_i\}_{i=1}^n$) is a natural generalization of convexity and $(L_0, L_1)$-smoothness of $f$ to the finite-sum case. Next, the existence of common minimizer $x^*$ for all $\{f_i\}_{i=1}^n$ is a typical assumption for over-parameterized models (Belkin et al., 2019; Liang & Rakhlin, 2020; Zhang et al., 2021; Bartlett et al., 1998) and used in several recent works on the analysis of stochastic methods (Vaswani et al., 2019a;b; Loizou et al., 2021; Gower et al., 2021). Although Assumption 4 does not cover all possibly interesting stochastic scenarios, it does allow the variance of the stochastic gradients to depend on $x$ and grow with the growth of $\|x - x^*\|$, which is typical for DL, unlike the standard bounded variance assumption.

For such problems, we consider a direct extension of $(L_0, L_1)$-GD called $(L_0, L_1)$-Stochastic Gradient Descent ($(L_0, L_1)$-SGD, Algorithm 5 in Appendix H.1).

**Theorem 7.1.** *Let Assumption 4 hold. Then, the iterates generated by $(L_0, L_1)$-SGD with $0 < \eta \leqslant \frac{\nu}{2}$, $\nu = e^{-\nu}$ after $N$ iterations satisfy*

$$\min_{k=0,\ldots,N} \mathbb{E}\left[\min\left\{\frac{\nu L_0}{4nL_1^2}, f(x^k) - f(x^*)\right\}\right] \leqslant \frac{2L_0\|x^0 - x^*\|^2}{\eta(N+1)}. \tag{29}$$

As in the deterministic case, the upper bound is proportional to $L_0$ and $1/(N+1)$ and does not depend on a smoothness constant on some ball around the solution. However, one can notice that the convergence criterion in the above result is quite non-standard: typically, the results are given in terms of $\mathbb{E}\left[f(x^k) - f(x^*)\right]$. This happens because although functions $\{f_i\}_{i=1}^n$ have a common minimizer, we cannot guarantee that for some $k_0$ and any $k \geqslant k_0$ we have $\|\nabla f_i(x^k)\| \leqslant L_0/L_1$ with probability 1 for all $i \in \{1, \ldots, n\}$, i.e., the method does not have to converge uniformly for all samples. This implies that with some small probability $f(x^k) - f(x^*)$ can be larger than $\nu L_0/(4nL_1^2)$ for any $k = 0, 1, \ldots, N$ and any $N \geqslant 0$. However, in view of (29), this probability has to be smaller than $\frac{8nL_1^2\|x^0 - x^*\|^2}{\eta\nu(N+1)}$, i.e., with probability at least $1 - \frac{8nL_1^2\|x^0 - x^*\|^2}{\eta\nu(N+1)}$ for $k(N)$ such that $\mathbb{E}\left[\min\left\{\frac{\nu L_0}{4nL_1^2}, f(x^{k(N)}) - f(x^*)\right\}\right] = \min_{k=0,\ldots,N} \mathbb{E}\left[\min\left\{\frac{\nu L_0}{4nL_1^2}, f(x^k) - f(x^*)\right\}\right]$ we have $f(x^{k(N)}) - f(x^*) \leqslant \frac{\nu L_0}{4nL_1^2}$, which is small for large enough $n$.

Next, we consider SGD-PS proposed by Loizou et al. (2021) (Algorithm 6 in Appendix H.2). In contrast to the deterministic case, SGD-PS requires to know $\{f_i(x^*)\}_{i=1}^n$ in advance. Nevertheless, these values equal 0 for some existing over-parameterized models, and thus, the method can be applied in such cases. Under the same assumptions, we also derive a similar result for SGD-PS.

**Theorem 7.2.** *Let Assumption 4 hold. Then, the iterates of SGD-PS after $N$ iterations satisfy*

$$\min_{k=0,\ldots,N} \mathbb{E}\left[\min\left\{\frac{\nu L_0}{4nL_1^2}, f(x^k) - f(x^*)\right\}\right] \leqslant \frac{4L_0\|x^0 - x^*\|^2}{\nu(N+1)}. \tag{30}$$

The result is very similar to the one we derive for $(L_0, L_1)$-SGD. Therefore, the discussion provided after Theorem 7.1 (with $\eta = \nu/2$) is valid for the above result as well.

## 8 CONCLUSION AND FUTURE WORK

In this paper, we derive improved convergence rates for $(L_0, L_1)$-GD and GD-PS, derive convergence guarantees for the new accelerated method called $(L_0, L_1)$-STM, and also derive a new result for AdGD in the case of (strongly) convex $(L_0, L_1)$-smooth optimization. Our results for $(L_0, L_1)$-GD and GD-PS depend neither on $\|\nabla f(x^0)\|$ nor on $f(x^0) - f(x^*)$ nor on exponential functions of $R_0$. We also prove new results for the stochastic extensions of $(L_0, L_1)$-GD and GD-PS in the case of finite sums of functions having a common minimizer. Nevertheless, several important questions remain open. One of these questions is the lower bounds for the class of (strongly) convex $(L_0, L_1)$-smooth functions and optimal methods for this class. Moreover, it would be interesting to develop stochastic extensions of $(L_0, L_1)$-GD and GD-PS with strong theoretical guarantees beyond the case of finite sums with shared minimizer.

ACKNOWLEDGEMENTS

All authors affiliated with the Innopolis University were supported by the Research Center of the Artificial Intelligence Institute of Innopolis University. We thank Konstantin Mishchenko for his useful suggestions for improving the writing. We also thank the anonymous reviewers for their valuable feedback.

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

## CONTENTS

# A    EXTRA RELATED WORK

**On the importance of convex analysis.**    Although many existing problems are not convex, it is useful to understand methods behavior under the convexity assumption as well due to several reasons. First of all, since the class of non-convex functions is too broad, the existing results for this class are quite pessimistic. In particular, among first-order methods, Gradient Descent is the best (in theory) first-order method if only smoothness is assumed (Carmon et al., 2021). In contrast, while accelerated/momentum methods do not have theoretical advantages over Gradient Descent for non-convex problems and shine in theory only under convexity-like assumptions, they work better in practice even when the problems are not convex (Sutskever et al., 2013). Last but not least, several recent works show that some problems appearing in Deep Learning, Optimal Control, and Reinforcement Learning have properties akin to (strongly) convex functions (Liu et al., 2022) and are even hiddenly convex (Fatkhullin et al., 2023).

**Results for stochastic methods.**    In the main part of the paper, we mainly discuss the existing convergence results under $(L_0, L_1)$-smoothness for deterministic methods. However, there exist several stochastic extensions. For non-convex twice differentiable $(L_0, L_1)$-smooth problems with $\sigma$-bounded noise, Zhang et al. (2020b) show $\mathcal{O}\left(\max\left\{(L_0+L_1\sigma)\hat{\Delta}/\varepsilon^2, (L_1\hat{\Delta}/\varepsilon, \hat{\Delta}^2/\varepsilon^4\right\}\right)$ complexity bound with $\hat{\Delta} := f(x) - \inf_{x \in \mathbb{R}^d} f(x) + (5L_0 + 2L_1\sigma)\sigma^2 + 9\sigma L_0^2/L_1$ for Clip-SGD with a clipping level dependent on the noise level. Zhang et al. (2020a) generalize the result from (Zhang et al., 2020b) to the case of $(L_0, L_1)$-smooth functions that are not necessarily twice differentiable and derive improved $\mathcal{O}\left(\Delta L_0\sigma^2/\varepsilon^4\right)$ complexity for sufficiently small $\varepsilon$. Next, Zhao et al. (2021) derive $\mathcal{O}\left(\max\left\{L_1\Delta/\varepsilon, L_0\Delta/\varepsilon^2, L_1\Delta\sigma^2/\varepsilon^3, L_0\Delta\sigma^2/\varepsilon^4\right\}\right)$ complexity bound for Normalized SGD with sufficiently large batchsize under $\sigma$-bounded variance assumption. Under the additional assumption of expected $(L_0, L_1)$-smoothness, Chen et al. (2023) improve the previous results and derive $\mathcal{O}\left((L_1\sigma+L_0)\Delta/\varepsilon^{-3}\right)$ complexity for SPIDER (Fang et al., 2018). Crawshaw et al. (2022) derive a similar bound to the one from (Zhang et al., 2020a) for a Generalized SignSGD under coordinate-wise $(L_0, L_1)$-smoothness. Next, Faw et al. (2023); Wang et al. (2023) derive (high-probability) complexity bounds for AdaGrad-Norm under the affine variance assumption. Similar results are obtained for Adam by Wang et al. (2022), and Li et al. (2024b) analyze Adam for $(L_0, L_1)$-smooth problems under bounded noise assumption. Recently, Koloskova et al. (2023) derive $\mathcal{O}\left(\max\left\{\Delta/\eta\varepsilon^2, \Delta/\eta c\varepsilon\right\}\right)$ complexity bound of finding $\varepsilon$-stationary point for Clip-SGD under $\sigma$-bounded variance assumption, where $c$ is the clipping level, $\eta \leqslant 1/9(L_0+cL_1)$ is the stepsize, and $\varepsilon = \Omega\left(\min\left\{\sigma^2, \sigma^2/c\right\} + \sqrt{\eta(L_0 + cL_1)}\sigma\right)$. Next, Li et al. (2024a) obtain $\mathcal{O}\left(\max\left\{L\tilde{\Delta}/\varepsilon^2, (L_1^2\tilde{\Delta}^4+L_0\tilde{\Delta}^3)/\varepsilon^4\right\}\right)$ high-probability complexity bound[7] for SGD with sufficiently small stepsize, where $\tilde{\Delta} := (\Delta+\sigma)/\delta$, $\delta$ is failure probability, and $L := L_0 + L_1(L_1\tilde{\Delta} + \sqrt{L_0\tilde{\Delta}})$ Finally, Hübler et al. (2024) prove $\mathcal{O}\left((\Delta e^{L_1^2}+\sigma+e^{L_1^2}L_0)^4/\varepsilon^4\right)$ and $\mathcal{O}\left((L_1\Delta+\sigma+\frac{L_0}{L_1})^4/\varepsilon^4\right)$ complexity bounds for parameter-agnostic and parameter-non-agnostic versions of Normalized SGD with momentum. To the best of our knowledge, there are no results in the literature for convex $(L_0, L_1)$-smooth stochastic optimization.

**Gradient clipping.**    As follows from the above discussion, gradient clipping is a useful tool for handling possible non-smoothness of the objective, which is also confirmed in practice (Goodfellow et al., 2016). However, it is worth mentioning that clipping has also other applications. In particular, gradient clipping is used to handle heavy-tailed noise (Zhang et al., 2020c; Gorbunov et al., 2020; Cutkosky & Mehta, 2021), to achieve differentiable privacy (Abadi et al., 2016; Chen et al., 2020), and also to tolerate Byzantine attacks (Karimireddy et al., 2021; Malinovsky et al., 2023).

**Polyak Stepsizes.**    GD with Polyak Stepsizes (GD-PS) is a celebrated approach for making GD parameter-free (under the assumption that $f(x^*)$ is known) (Polyak, 1987). In particular, Hazan & Kakade (2019) show that GD-PS achieves the same rate as GD with optimally chosen constant stepsize (up to a constant factor) for convex Lipschitz functions, convex smooth functions, and strongly convex smooth functions. Moreover, some recent works (Loizou et al., 2021; Galli et al.,

---

[7]The complexity bound from (Li et al., 2024a) for more general notion of smoothness. The complexity bound we provide in the text is the special case of the one from (Li et al., 2024a).

2023; Berrada et al., 2020; Horváth et al., 2022; Gower et al., 2022; Abdukhakimov et al., 2024; Li et al., 2022) also consider different stochastic extensions of GD-PS.

**Other Notions of Generalized Smoothness.** $(L_0, L_1)$-smoothness belongs to the class of assumptions on so-called generalized smoothness. Classical assumptions of this type include Hölder continuity of the gradient (Nemirovski & Yudin, 1983; Nemirovskii & Nesterov, 1985), relative smoothness (Bauschke et al., 2017), and local smoothness, i.e., Lipschitzness of the gradient on any compact (Malitsky & Mishchenko, 2020; Patel & Berahas, 2022; Patel et al., 2022; Gorbunov et al., 2021; Sadiev et al., 2023). Although these assumptions are quite broad (e.g., for local smoothness, it is sufficient to assume just continuity of the gradient), they do not relate the growth of non-smoothness/local Lipschitz constant of the gradient with the growth of the gradient or distance to the solution. From this perspective, assumptions such as polynomial growth of the gradient norm (Mai & Johansson, 2021), $\alpha$-symmetric $(L_0, L_1)$-smoothness (Chen et al., 2023) which also introduces the key technical tools, and $(r, \ell)$-smoothness (Li et al., 2024a) are closer to Assumption 3 than local Lipschitz/Hölder continuity of the gradient and relative smoothness.

## B   EXAMPLES OF $(L_0, L_1)$-SMOOTH FUNCTIONS

**Example B.1** (Power of Norm). *Let $f(x) = \|x\|^{2n}$, where $n$ is a positive integer. Then, $f(x)$ is convex and $(2n, 2n-1)$-smooth. Moreover, $f(x)$ is not $L$-smooth for $n \geqslant 2$.*

*Proof.* Convexity of $f$ follows from convexity and monotonicity of $\varphi(t) = t^{2n}$ for $t \geqslant 0$ and convexity of $h(x) = \|x\|$, since $f(x) = \varphi(h(x))$. To show $(L_0, L_1)$-smoothness, we compute gradient and Hessian of $f(x)$:

$$\nabla f(x) = 2n\|x\|^{2(n-1)}x,$$

$$\nabla^2 f(x) = \begin{cases} 2\mathbf{I}, & \text{if } n = 1 \\ 4n(n-1)\|x\|^{2(n-2)}xx^\top + 2n\|x\|^{2(n-1)}\mathbf{I}, & \text{if } n > 1. \end{cases}$$

Therefore,

$$\|\nabla f(x)\| = 2n\|x\|^{2n-1},$$

$$\|\nabla^2 f(x)\|_2 = \begin{cases} 2, & \text{if } n = 1 \\ 2n(2n-1)\|x\|^{2(n-1)}, & \text{if } n > 1 \end{cases}$$

$$= 2n(2n-1)\|x\|^{2n-2},$$

which implies

$$\|\nabla^2 f(x)\|_2 - (2n-1)\|\nabla f(x)\| = 2n(2n-1)\|x\|^{2n-2}(1 - \|x\|).$$

If $\|x\| \geqslant 1$, then we have $\|\nabla^2 f(x)\|_2 \leqslant (2n-1)\|\nabla f(x)\|$. If $\|x\| \leqslant 1$, then

$$\|\nabla^2 f(x)\|_2 - (2n-1)\|\nabla f(x)\| \leqslant 2n(2n-1)\max_{t \in [0,1]} \psi(t),$$

where $\psi(t) := t^{2n-2}(1-t)$. For $n = 1$ we have $\max_{t \in [0,1]} \psi(t) = 1$ and $\|\nabla^2 f(x)\|_2 - (2n-1)\|\nabla f(x)\| \leqslant 2$. For $n > 1$ we have $\max_{t \in [0,1]} \psi(t) = \left(\frac{2n-2}{2n-1}\right)^{2n-2} \frac{1}{2n-1} \leqslant \frac{1}{2n-1}$, which gives $\|\nabla^2 f(x)\|_2 - (2n-1)\|\nabla f(x)\| \leqslant 2n$. Putting two cases together, we get

$$\|\nabla^2 f(x)\|_2 \leqslant 2n + (2n-1)\|\nabla f(x)\|$$

that is equivalent to $(2n, 2n-1)$-smoothness (Chen et al., 2023, Theorem 1). Non-smoothness of $f$ for $n > 1$ follows from the unboundedness of $\|\nabla^2 f(x)\|_2$ in this case. $\qquad\square$

**Example B.2** (Exponent of the Inner Product). *Function $f(x) = \exp(a^\top x)$ for some $a \in \mathbb{R}^d$ is convex, $(0, \|a\|)$-smooth, but not $L$-smooth for any $L \geqslant 0$.*

*Proof.* Let us compute the gradient and Hessian of $f$:

$$\nabla f(x) = a \exp(a^\top x), \quad \nabla^2 f(x) = aa^\top \exp(a^\top x).$$

Clearly $\nabla^2 f(x) \succcurlyeq 0$, meaning that $f(x)$ is convex. Moreover,

$$\|\nabla^2 f(x)\|_2 = \|a\|^2 \exp(a^\top x) = \|a\| \cdot \|\nabla f(x)\|$$

that is equivalent to $(0, \|a\|)$-smoothness (Chen et al., 2023, Theorem 1). When $a \neq 0$ function $f$ has unbounded Hessian, i.e., $f$ is not $L$-smooth for any $L \geqslant 0$ in this case. $\qquad\square$

**Example B.3** (Logistic Function). *Consider logistic function: $f(x) = \log\left(1 + \exp(-a^\top x)\right)$, where $a \in \mathbb{R}^d$ is some vector. Function $f$ is $(L_0, L_1)$-smooth with $L_0 = 0$ and $L_1 = \|a\|$.*

*Proof.* The gradient and the Hessian of $f(x)$ equal

$$\nabla f(x) = -\frac{a}{1 + \exp(a^\top x)}, \quad \nabla^2 f(x) = \frac{aa^\top}{\left(\exp\left(-\frac{1}{2}a^\top x\right) + \exp\left(\frac{1}{2}a^\top x\right)\right)^2}.$$

Moreover,

$$\|\nabla f(x)\| = \frac{\|a\|}{1 + \exp(a^\top x)}, \quad \|\nabla^2 f(x)\|_2 = \frac{\|a\|^2}{\left(\exp\left(-\frac{1}{2}a^\top x\right) + \exp\left(\frac{1}{2}a^\top x\right)\right)^2}.$$

This leads to

$$
\begin{aligned}
\frac{\|\nabla^2 f(x)\|}{\|\nabla f(x)\|} &= \frac{1 + \exp(a^\top x)}{\left(\exp\left(-\frac{1}{2}a^\top x\right) + \exp\left(\frac{1}{2}a^\top x\right)\right)^2}\|a\| \\
&= \frac{1 + \exp(a^\top x)}{\exp\left(-a^\top x\right)\left(1 + \exp\left(a^\top x\right)\right)^2}\|a\| \\
&= \frac{1}{1 + \exp(-a^\top x)}\|a\| \leqslant \|a\|,
\end{aligned}
$$

implying that $\|\nabla^2 f(x)\| \leqslant \|a\| \cdot \|\nabla f(x)\|$ for all $x \in \mathbb{R}^d$. This condition is equivalent to $(0, \|a\|)$-smoothness (Chen et al., 2023, Theorem 1). $\qquad\square$

## C    Proof of Lemma 2.2

**Lemma C.1** (Lemma 2.2). *Let Assumption 3 hold and $\nu$ satisfy[8] $\nu = e^{-\nu}$. Then, the following statements hold.*

1. *For $f_* := \inf_{x \in \mathbb{R}^d} f(x)$, arbitrary $x \in \mathbb{R}^d$, and $\nu$ such that $\nu \exp(\nu) = 1$, we have*

$$\frac{\nu \|\nabla f(x)\|^2}{2(L_0 + L_1\|\nabla f(x)\|)} \leqslant f(x) - f_*. \tag{31}$$

2. *If additionally Assumption 1 with $\mu = 0$ holds, then for any $x, y \in \mathbb{R}^d$ such that*

$$L_1\|x - y\| \exp\left(L_1\|x - y\|\right) \leqslant 1, \tag{32}$$

*we have*

$$\frac{\nu \|\nabla f(x) - \nabla f(y)\|^2}{2(L_0 + L_1\|\nabla f(y)\|)} \leqslant f(y) - f(x) - \langle \nabla f(x), y - x \rangle, \tag{33}$$

*and*

$$\frac{\nu \|\nabla f(x) - \nabla f(y)\|^2}{2(L_0 + L_1\|\nabla f(y)\|)} + \frac{\nu \|\nabla f(x) - \nabla f(y)\|^2}{2(L_0 + L_1\|\nabla f(x)\|)} \leqslant \langle \nabla f(x) - \nabla f(y), x - y \rangle. \tag{34}$$

*Proof.* To prove (31), we apply (7) with $y = x - \frac{\nu}{L_0 + L_1\|\nabla f(x)\|}\nabla f(x)$ for given $x \in \mathbb{R}^d$ and $\nu$ such that $\nu \exp(\nu) = 1$:

$$
\begin{aligned}
f_* \leqslant f(y) \quad &\overset{(7)}{\leqslant} \quad f(x) + \langle \nabla f(x), y - x \rangle + \frac{L_0 + L_1\|\nabla f(x)\|}{2}\exp(L_1\|x - y\|)\|x - y\|^2 \\
&= \quad f(x) - \frac{\nu\|\nabla f(x)\|^2}{L_0 + L_1\|\nabla f(x)\|} \\
&\quad + \frac{L_0 + L_1\|\nabla f(x)\|}{2} \cdot \exp\left(\frac{L_1\nu\|\nabla f(x)\|}{L_0 + L_1\|\nabla f(x)\|}\right) \cdot \frac{\nu^2\|\nabla f(x)\|^2}{(L_0 + L_1\|\nabla f(x)\|)^2} \\
&\leqslant \quad f(x) - \frac{\nu\|\nabla f(x)\|^2}{L_0 + L_1\|\nabla f(x)\|} + \frac{\nu\|\nabla f(x)\|^2}{2(L_0 + L_1\|\nabla f(x)\|)} \cdot \nu \exp(\nu) \\
&\overset{\nu = e^{-\nu}}{\leqslant} \quad f(x) - \frac{\nu\|\nabla f(x)\|^2}{2(L_0 + L_1\|\nabla f(x)\|)}.
\end{aligned}
$$

Rearranging the terms, we get (31).

Next, we will prove (33) and (34) under Assumptions 1 and 3. The proof follows similar steps to the one that holds for standard $L$-smoothness (i.e., cocoercivity of the gradient) (Nesterov, 2018):

$$\|\nabla f(x) - \nabla f(y)\|^2 \leqslant L\langle \nabla f(x) - \nabla f(y), x - y \rangle.$$

That is, for given $x$ we consider function $\varphi_x(y) := f(y) - \langle \nabla f(x), y \rangle$. This function is differentiable and $\nabla \varphi_x(y) = \nabla f(y) - \nabla f(x)$. Moreover, for any $u, y \in \mathbb{R}^d$ we have

$$\|\nabla \varphi_x(u) - \nabla \varphi_x(y)\| = \|\nabla f(u) - \nabla f(y)\| \overset{(6)}{\leqslant} (L_0 + L_1\|\nabla f(u)\|)\|u - y\|\exp(L_1\|u - y\|), \tag{35}$$

Next, for given $x$ and for any $y, u \in \mathbb{R}^d$ we define function $\psi_{x,y,u}(t) : \mathbb{R} \to \mathbb{R}$ as $\psi_{x,y,u}(t) := \varphi_x(u + t(y - u))$. Then, by definition of $\psi_{x,y,u}$, we have $\varphi_x(u) = \psi_{x,y,u}(0)$, $\varphi_x(y) = \psi_{x,y,u}(1)$,

---

[8]One can check numerically that $0.56 < \nu < 0.57$.

and $\psi'_{x,y,u}(t) = \langle \nabla \varphi_x(u + t(y-u)), y - u \rangle$. Therefore, using Newton-Leibniz formula, we derive

$$
\begin{aligned}
\varphi_x(y) - \varphi_x(u) &= \psi_{x,y,u}(1) - \psi_{x,y,u}(0) = \int_0^1 \psi'_{x,y,u}(t)dt \\
&= \int_0^1 \langle \nabla \varphi_x(u + t(y-u)), y - u \rangle dt \\
&= \langle \nabla \varphi_x(u), y - u \rangle + \int_0^1 \langle \nabla \varphi_x(u + t(y-u)) - \nabla \varphi_x(u), y - u \rangle dt \\
&\leqslant \langle \nabla \varphi_x(u), y - u \rangle + \int_0^1 \|\nabla \varphi_x(u + t(y-u)) - \nabla \varphi_x(u)\| \cdot \|u - y\| dt \\
&\overset{(35)}{\leqslant} \langle \nabla \varphi_x(u), y - u \rangle + \int_0^1 (L_0 + L_1\|\nabla f(u)\|) \exp(tL_1\|u - y\|)\|u - y\|^2 t dt \\
&\leqslant \langle \nabla \varphi_x(u), y - u \rangle + \frac{L_0 + L_1\|\nabla f(u)\|}{2} \exp(L_1\|u - y\|)\|u - y\|^2
\end{aligned}
$$

that implies $\forall u, y \in \mathbb{R}^d$

$$
\varphi_x(y) \leqslant \varphi_x(u) + \langle \nabla \varphi_x(u), y - u \rangle + \frac{L_0 + L_1\|\nabla f(u)\|}{2} \exp(L_1\|u - y\|)\|u - y\|^2. \tag{36}
$$

To proceed, we will need the following inequality:

$$
\begin{aligned}
\nu \exp\left(\nu \frac{L_1\|\nabla \varphi_x(u)\|}{L_0 + L_1\|\nabla f(u)\|}\right) &= \nu \exp\left(\nu \frac{L_1\|\nabla f(u) - \nabla f(x)\|}{L_0 + L_1\|\nabla f(u)\|}\right) \\
&\overset{(6)}{\leqslant} \nu \exp\left(\nu \frac{L_1\|x - u\| \exp(L_1\|x - u\|)(L_0 + L_1\|\nabla f(u)\|)}{L_0 + L_1\|\nabla f(u)\|}\right) \\
&= \nu \exp(\nu L_1\|x - u\| \exp(L_1\|x - u\|)) \\
&\overset{(32)}{\leqslant} \nu \exp(\nu) \overset{\nu = e^{-\nu}}{=} 1. \tag{37}
\end{aligned}
$$

Using the above bound and (36) with $y = u - \frac{\nu}{L_0 + L_1\|\nabla f(u)\|} \nabla \varphi_x(u)$, we derive

$$
\begin{aligned}
\varphi_x &\left(u - \frac{\nu}{L_0 + L_1\|\nabla f(u)\|} \nabla \varphi_x(u)\right) \\
&\overset{(36)}{\leqslant} \varphi_x(u) - \nu \frac{\|\nabla \varphi_x(u)\|^2}{L_0 + L_1\|\nabla f(u)\|} + \frac{\nu^2\|\nabla \varphi_x(u)\|^2}{2(L_0 + L_1\|\nabla f(u)\|)} \exp\left(\nu \frac{L_1\|\nabla \varphi_x(u)\|}{L_0 + L_1\|\nabla f(u)\|}\right) \\
&\overset{(37)}{\leqslant} \varphi_x(u) - \nu \frac{\|\nabla \varphi_x(u)\|^2}{L_0 + L_1\|\nabla f(u)\|} + \frac{\nu\|\nabla \varphi_x(u)\|^2}{2(L_0 + L_1\|\nabla f(u)\|)} \\
&\leqslant \varphi_x(u) - \nu \frac{\|\nabla \varphi_x(u)\|^2}{2(L_0 + L_1\|\nabla f(u)\|)},
\end{aligned}
$$

Taking into account that $x$ is an optimum for $\varphi_x(u)$ ($\nabla \varphi_x(x) = 0$) and the definition of $\varphi_x(u)$, we get the following inequality from the above one:

$$
f(x) - \langle \nabla f(x), x \rangle \leqslant f(u) - \langle \nabla f(x), u \rangle - \frac{\nu\|\nabla f(x) - \nabla f(u)\|^2}{2(L_0 + L_1\|\nabla f(u)\|)}, \quad \forall x, u \in \mathbb{R}^d,
$$

which is equivalent to

$$
\frac{\nu\|\nabla f(x) - \nabla f(y)\|^2}{2(L_0 + L_1\|\nabla f(y)\|)} \leqslant f(y) - f(x) - \langle \nabla f(x), y - x \rangle, \quad \forall x, y \in \mathbb{R}^d.
$$

Therefore, we established (33). Moreover, by swapping $x$ and $y$ in the above inequality, we also get

$$
\frac{\nu\|\nabla f(x) - \nabla f(y)\|^2}{2(L_0 + L_1\|\nabla f(y)\|)} \leqslant f(x) - f(y) - \langle \nabla f(y), x - y \rangle, \quad \forall x, y \in \mathbb{R}^d.
$$

To get (34), it remains to sum the above two inequalities. $\qquad\square$

## D    MISSING PROOFS FOR $(L_0, L_1)$-GD

**Lemma D.1** (Lemma 3.1: monotonicity of function value). *Let Assumption 3 hold. Then, for all $k \geqslant 0$ the iterates generated by $(L_0, L_1)$-GD with $\eta \leqslant \nu$, $\nu = e^{-\nu}$ satisfy*

$$f(x^{k+1}) \leqslant f(x^k) - \frac{\eta \|\nabla f(x^k)\|^2}{2(L_0 + L_1 \|\nabla f(x^k)\|)} \leqslant f(x^k). \tag{38}$$

*Proof.* Applying (7) with $y = x^{k+1}$ and $x = x^k$ and using

$$\exp(L_1 \|x^{k+1} - x^k\|) = \exp\left(\eta \frac{L_1 \|\nabla f(x^k)\|}{L_0 + L_1 \|\nabla f(x^k)\|}\right) \leqslant \exp(\eta) \tag{39}$$

we get

$$
\begin{aligned}
f(x^{k+1}) \quad &\leqslant \quad f(x^k) + \langle \nabla f(x^k), x^{k+1} - x^k \rangle + \frac{L_0 + L_1 \|\nabla f(x^k)\|}{2} \|x^{k+1} - x^k\|^2 \exp(\eta) \\
&= \quad f(x^k) - \frac{\eta \|\nabla f(x^k)\|^2}{L_0 + L_1 \|\nabla f(x^k)\|} + \frac{\eta^2 \exp(\eta) \|\nabla f(x^k)\|^2}{2(L_0 + L_1 \|\nabla f(x^k)\|)} \\
&= \quad f(x^k) - \eta \left(1 - \frac{\eta \exp(\eta)}{2}\right) \frac{\|\nabla f(x^k)\|^2}{L_0 + L_1 \|\nabla f(x^k)\|} \\
&\overset{\eta \leqslant \nu}{\leqslant} \quad f(x^k) - \eta \left(1 - \frac{\nu \exp(\nu)}{2}\right) \frac{\|\nabla f(x^k)\|^2}{L_0 + L_1 \|\nabla f(x^k)\|} \\
&\overset{\nu = e^{-\nu}}{=} \quad f(x^k) - \frac{\eta \|\nabla f(x^k)\|^2}{2(L_0 + L_1 \|\nabla f(x^k)\|)} \leqslant f(x^k),
\end{aligned}
$$

which finishes the proof. $\qquad\square$

**Lemma D.2** (Lemma 3.2: monotonicity of gradient norm). *Let Assumptions 1 with $\mu = 0$ and 3 hold. Then, for all $k \geqslant 0$ the iterates generated by $(L_0, L_1)$-GD with $\eta \leqslant \nu$, $\nu = e^{-\nu}$ satisfy*

$$\|\nabla f(x^{k+1})\| \leqslant \|\nabla f(x^k)\|. \tag{40}$$

*Proof.* For convenience, we introduce the following notation: $\omega_k := L_0 + L_1 \|\nabla f(x^k)\|$ for all $k \geqslant 0$. Since

$$
\begin{aligned}
L_1 \|x^{k+1} - x^k\| \exp\left(L_1 \|x^{k+1} - x^k\|\right) \quad &= \quad \frac{\eta L_1 \|\nabla f(x^k)\|}{L_0 + L_1 \|\nabla f(x^k)\|} \exp\left(\frac{\eta L_1 \|\nabla f(x^k)\|}{L_0 + L_1 \|\nabla f(x^k)\|}\right) \\
&\leqslant \quad \eta \exp(\eta) \overset{\eta \leqslant \nu}{\leqslant} \nu \exp(\nu) \overset{\nu = e^{-\nu}}{=} 1,
\end{aligned}
$$

the assumptions for the second part of Lemma 2.2 are satisfied for $x = x^{k+1}$ and $y = x^k$, and inequality (11) implies

$$
\begin{aligned}
\left(\frac{\nu}{2\omega_k} + \frac{\nu}{2\omega_{k+1}}\right) \|\nabla f(x^{k+1}) - \nabla f(x^k)\|^2 \quad &\leqslant \quad \langle \nabla f(x^{k+1}) - \nabla f(x^k), x^{k+1} - x^k \rangle \\
&= \quad -\frac{\eta}{\omega_k} \langle \nabla f(x^{k+1}) - \nabla f(x^k), \nabla f(x^k) \rangle,
\end{aligned}
$$

where in the second line we use $x^{k+1} = x^k - \frac{\eta}{\omega_k} \nabla f(x^k)$. Multiplying both sides by $2\omega_k / \nu$ and rearranging the terms, we get

$$
\begin{aligned}
\left(1 + \frac{\omega_k}{\omega_{k+1}}\right) &\left(\|\nabla f(x^{k+1})\|^2 + \|\nabla f(x^k)\|^2 - 2\langle \nabla f(x^{k+1}), \nabla f(x^k) \rangle\right) \\
&\leqslant \quad -\frac{2\eta}{\nu} \cdot \langle \nabla f(x^{k+1}), \nabla f(x^k) \rangle + \frac{2\eta}{\nu} \cdot \|\nabla f(x^k)\|^2,
\end{aligned}
$$

which is equivalent to

$$
\begin{aligned}
\left(1 + \frac{\omega_k}{\omega_{k+1}}\right)\|\nabla f(x^{k+1})\|^2 \;\leqslant\; & \left(1 + \frac{\omega_k}{\omega_{k+1}}\right)\|\nabla f(x^k)\|^2 \\
& + 2\left(1 + \frac{\omega_k}{\omega_{k+1}} - \frac{\eta}{\nu}\right)\langle\nabla f(x^{k+1}), \nabla f(x^k)\rangle \\
& - 2\left(1 + \frac{\omega_k}{\omega_{k+1}} - \frac{\eta}{\nu}\right)\|\nabla f(x^k)\|^2 \\
= \; & \left(1 + \frac{\omega_k}{\omega_{k+1}}\right)\|\nabla f(x^k)\|^2 \\
& + 2\left(1 + \frac{\omega_k}{\omega_{k+1}} - \frac{\eta}{\nu}\right)\langle\nabla f(x^{k+1}) - \nabla f(x^k), \nabla f(x^k)\rangle \\
= \; & \left(1 + \frac{\omega_k}{\omega_{k+1}}\right)\|\nabla f(x^k)\|^2 \\
& - \frac{2\omega_k}{\eta}\left(1 + \frac{\omega_k}{\omega_{k+1}} - \frac{\eta}{\nu}\right)\langle\nabla f(x^{k+1}) - \nabla f(x^k), x^{k+1} - x^k\rangle.
\end{aligned}
\tag{41}
$$

We notice that $\frac{2\omega_k}{\eta} > 0$ and $1 + \frac{\omega_k}{\omega_{k+1}} - \frac{\eta}{\nu} \geqslant \frac{\omega_k}{\omega_{k+1}} \geqslant 0$ since $0 < \frac{\eta}{\nu} \leqslant 1$. Moreover, due to the convexity of $f$ we also have $\langle\nabla f(x^{k+1}) - \nabla f(x^k), x^{k+1} - x^k\rangle \geqslant 0$. Therefore, we have

$$
-\frac{2\omega_k}{\eta}\left(1 + \frac{\omega_k}{\omega_{k+1}} - \frac{\eta}{\nu}\right)\langle\nabla f(x^{k+1}) - \nabla f(x^k), x^{k+1} - x^k\rangle \leqslant 0.
$$

Together with (41), the above inequality implies

$$
\left(1 + \frac{\omega_k}{\omega_{k+1}}\right)\|\nabla f(x^{k+1})\|^2 \leqslant \left(1 + \frac{\omega_k}{\omega_{k+1}}\right)\|\nabla f(x^k)\|^2, \quad \forall k \geqslant 0,
$$

which is equivalent to (40). $\qquad\square$

**Theorem D.1** (Theorem 3.1)**.** *Let Assumptions 1 with $\mu = 0$ and 3 hold. Then, the iterates generated by $(L_0, L_1)$-GD with $0 < \eta \leqslant \frac{\nu}{2}$, $\nu = e^{-\nu}$ satisfy the following implication:*

$$
\|\nabla f(x^k)\| \geqslant \frac{L_0}{L_1} \;\Longrightarrow\; k \leqslant \frac{8L_1^2\|x^0 - x^*\|^2}{\nu\eta} - 1 \;\text{ and }\; \|x^{k+1} - x^*\|^2 \leqslant \|x^k - x^*\|^2 - \frac{\nu\eta}{8L_1^2}. \tag{42}
$$

*Moreover, the output after $N > \frac{8L_1^2\|x^0 - x^*\|^2}{\nu\eta} - 1$ iterations satisfies*

$$
f(x^N) - f(x^*) \leqslant \frac{2L_0\|x^0 - x^*\|^2}{\eta(N+1-T)} - \frac{\nu L_0 T}{4L_1^2(N+1-T)} \leqslant \frac{2L_0\|x^0 - x^*\|^2}{\eta(N+1)}, \tag{43}
$$

*where $T := |\mathcal{T}|$ for the set $\mathcal{T} := \{k \in \{0, 1, \ldots N-1\} \mid \|\nabla f(x^k)\| \geqslant \frac{L_0}{L_1}\}$. In addition, if $\mu > 0$ and $N > \frac{8L_1^2\|x^0 - x^*\|^2}{\eta} - 1$, then*

$$
\|x^N - x^*\|^2 \leqslant \left(1 - \frac{\mu\eta}{4L_0}\right)^{N-T}\left(\|x^0 - x^*\|^2 - \frac{\nu\eta T}{8L_1^2}\right) \leqslant \left(1 - \frac{\mu\eta}{4L_0}\right)^{N-T}\|x^0 - x^*\|^2. \tag{44}
$$

**Remark D.1.** *In the strongly convex case, our convergence bound implies that $(L_0, L_1)$-GD with $\eta = \nu/2$ satisfies $\|x^N - x^*\|^2 \leqslant \varepsilon$ after $N = \mathcal{O}\left(\max\left\{L_0\log(R_0^2/\varepsilon)/\mu, L_1^2 R_0^2\right\}\right)$ iterations, while the complexity of Clip-GD derived by Koloskova et al. (2023) is $\mathcal{O}\left(\max\left\{L_0\log(R_0^2/\varepsilon)/\mu, L_0 R_0 \min\left\{\sqrt{L/\mu}, LR_0\right\}\right\}\right)$, which again can be arbitrarily worse than our bound due to the dependence on $L$.*

*Proof of Theorem D.1.* We start by expanding the squared distance to the solution:

$$
\begin{aligned}
\|x^{k+1} - x^*\|^2 &= \left\| x^k - x^* - \frac{\eta}{L_0 + L_1 \|\nabla f(x^k)\|} \nabla f(x^k) \right\|^2 \\
&= \|x^k - x^*\|^2 - \frac{2\eta}{L_0 + L_1 \|\nabla f(x^k)\|} \langle x^k - x^*, \nabla f(x^k) \rangle \\
&\quad + \frac{\eta^2 \|\nabla f(x^k)\|^2}{(L_0 + L_1 \|\nabla f(x^k)\|)^2} \\
&\overset{(2)}{\leqslant} \|x^k - x^*\|^2 - \frac{2\eta}{L_0 + L_1 \|\nabla f(x^k)\|} \left( f(x^k) - f(x^*) \right) \\
&\quad + \frac{\eta^2 \|\nabla f(x^k)\|^2}{(L_0 + L_1 \|\nabla f(x^k)\|)^2} \\
&\overset{(8)}{\leqslant} \|x^k - x^*\|^2 - 2\eta \left( 1 - \frac{\eta}{\nu} \right) \frac{f(x^k) - f(x^*)}{L_0 + L_1 \|\nabla f(x^k)\|} \\
&\overset{\eta \leqslant \frac{\nu}{2}}{\leqslant} \|x^k - x^*\|^2 - \eta \frac{f(x^k) - f(x^*)}{L_0 + L_1 \|\nabla f(x^k)\|}.
\end{aligned}
\tag{45}
$$

To continue the derivation, we consider two possible cases: $\|\nabla f(x^k)\| \geqslant \frac{L_0}{L_1}$ or $\|\nabla f(x^k)\| < \frac{L_0}{L_1}$.

**Case 1:** $\|\nabla f(x^k)\| \geqslant \frac{L_0}{L_1}$**.** In this case, we have

$$
L_0 + L_1 \|\nabla f(x^k)\| \leqslant 2 L_1 \|\nabla f(x^k)\|,
\tag{46}
$$

$$
\frac{\nu \|\nabla f(x^k)\|}{4 L_1} \overset{(46)}{\leqslant} \frac{\nu \|\nabla f(x^k)\|^2}{2(L_0 + L_1 \|\nabla f(x^k)\|)} \overset{(8)}{\leqslant} f(x^k) - f(x^*).
\tag{47}
$$

Plugging the above inequalities in (45), we continue the derivation as follows:

$$
\begin{aligned}
\|x^{k+1} - x^*\|^2 &\overset{(45),(46)}{\leqslant} \|x^k - x^*\|^2 - \eta \frac{f(x^k) - f(x^*)}{2 L_1 \|\nabla f(x^k)\|} \\
&\overset{(47)}{\leqslant} \|x^k - x^*\|^2 - \frac{\nu \eta}{8 L_1^2}.
\end{aligned}
\tag{48}
$$

We notice that if $\|\nabla f(x^k)\| \geqslant \frac{L_0}{L_1}$, then, in view of Lemma 3.2, we also have $\|\nabla f(x^t)\| \geqslant \frac{L_0}{L_1}$ for all $t = 0, 1, \ldots, k$. Therefore, (48) implies

$$
\|x^{k+1} - x^*\|^2 \leqslant \|x^0 - x^*\|^2 - \frac{\nu \eta}{8 L_1^2} (k+1).
\tag{49}
$$

Since $\|x^{k+1} - x^*\|^2 \geqslant 0$, $k$ should be bounded as $k \leqslant \frac{8 L_1^2 \|x^0 - x^*\|^2}{\nu \eta} - 1$, which gives (14). We denote $T := |\mathcal{T}|$ for the set $\mathcal{T} := \{ k \in \{0, 1, \ldots N-1\} \mid \|\nabla f(x^k)\| \geqslant \frac{L_0}{L_1} \}$. Therefore, in view of (49) and non-negativity of the squared distance, $T$ is bounded as $T \leqslant \frac{8 L^2 \|x^0 - x^*\|^2}{\nu \eta}$.

**Case 2:** $\|\nabla f(x^k)\| < \frac{L_0}{L_1}$**.** In this case, we have

$$
L_0 + L_1 \|\nabla f(x^k)\| \leqslant 2 L_0,
\tag{50}
$$

implying that

$$
\|x^{k+1} - x^*\|^2 \overset{(45),(50)}{\leqslant} \|x^k - x^*\|^2 - \frac{\eta}{2 L_0} \left( f(x^k) - f(x^*) \right).
\tag{51}
$$

Moreover, since the norm of the gradient is non-increasing along the trajectory of $(L_0, L_1)$-GD (Lemma 3.2), $\|\nabla f(x^k)\| < \frac{L_0}{L_1}$ implies that $k > T$. Therefore, we can sum up inequalities (51) for

$k = T, T + 1, \ldots, N$, rearrange the terms, and get

$$
\begin{aligned}
\frac{1}{N+1-T} \sum_{k=T}^{N} \left( f(x^k) - f(x^*) \right) &\leqslant \frac{2L_0}{\eta(N+1-T)} \sum_{k=T}^{N} \left( \|x^k - x^*\|^2 - \|x^{k+1} - x^*\|^2 \right) \\
&= \frac{2L_0 \left( \|x^T - x^*\|^2 - \|x^{N+1} - x^*\|^2 \right)}{\eta(N+1-T)} \\
&\leqslant \frac{2L_0 \|x^T - x^*\|^2}{\eta(N-T)}.
\end{aligned}
$$

Finally, we take into account that for $k = 0, 1, \ldots, T-1$, we have $\|\nabla f(x^k)\| \geqslant \frac{L_0}{L_1}$:

$$
\frac{1}{N+1-T} \sum_{k=T}^{N} \left( f(x^k) - f(x^*) \right) \overset{(49)}{\leqslant} \frac{2L_0 \|x^0 - x^*\|^2}{\eta(N+1-T)} - \frac{\nu L_0 T}{4L_1^2(N+1-T)}. \tag{52}
$$

It remains to notice that Lemma 3.1 implies $f(x^N) - f(x^*) \leqslant \frac{1}{N-T} \sum_{k=T+1}^{N} \left( f(x^k) - f(x^*) \right)$. Together with the above inequality, it implies the first part (15). To derive the second part of (15), it remains to notice that for $N > \frac{8L_1^2 \|x^0 - x^*\|^2}{\nu\eta} - 1$ the right-hand side of (52) as a function of $T$ attains its maximum at $T = 0$. Indeed, the derivative of function

$$
\begin{aligned}
\varphi(T) &:= \frac{2L_0 \|x^0 - x^*\|^2}{\eta(N+1-T)} - \frac{\nu L_0 T}{4L_1^2(N+1-T)} \\
&= \frac{2L_0 \|x^0 - x^*\|^2}{\eta(N+1-T)} + \frac{\nu L_0}{4L_1^2} - \frac{\nu L_0(N+1)}{4L_1^2(N-T+1)}
\end{aligned}
$$

equals

$$
\varphi'(T) = \frac{2L_0 \|x^0 - x^*\|^2}{\eta(N-T+1)^2} - \frac{\nu L_0(N+1)}{4L_1^2(N-T+1)^2}.
$$

Since $N > \frac{8L_1^2 \|x^0 - x^*\|^2}{\nu\eta} - 1$, we have $\varphi'(T) < 0$, i.e., $\varphi(T)$ is a decreasing function of $T$, meaning that

$$
\frac{2L_0 \|x^0 - x^*\|^2}{\eta(N+1-T)} - \frac{\nu L_0 T}{4L_1^2(N+1-T)} \leqslant \frac{2L_0 \|x^0 - x^*\|^2}{\eta(N+1)},
$$

which gives (43).

To prove (44), we notice that for $\mu > 0$ we have $f(x^k) - f(x^*) \geqslant \frac{\mu}{2} \|x^k - x^*\|^2$ implying

$$
\|x^{k+1} - x^*\|^2 \overset{(51)}{\leqslant} \left( 1 - \frac{\mu\eta}{4L_0} \right) \|x^k - x^*\|^2, \tag{53}
$$

when $\|\nabla f(x^k)\| < \frac{L_0}{L_1}$. Therefore, for $N > T$ we have

$$
\begin{aligned}
\|x^N - x^*\|^2 &\overset{(53)}{\leqslant} \left( 1 - \frac{\mu\eta}{4L_0} \right)^{N-T} \|x^T - x^*\|^2 \\
&\overset{(49)}{\leqslant} \left( 1 - \frac{\mu\eta}{4L_0} \right)^{N-T} \left( \|x^0 - x^*\|^2 - \frac{\nu\eta T}{8L_1^2} \right) \\
&\leqslant \left( 1 - \frac{\mu\eta}{4L_0} \right)^{N-T} \|x^0 - x^*\|^2,
\end{aligned}
$$

which gives (44) and concludes the proof. $\qquad \square$

### D.1 COMPARISON WITH THE PROOFS FROM (KOLOSKOVA ET AL., 2023; TAKEZAWA ET AL., 2024; CHEN ET AL., 2023)

**Comparison with (Koloskova et al., 2023; Takezawa et al., 2024).** In this paragraph, we further elaborate on the difference between our proof and the ones given by (Koloskova et al., 2023; Takezawa

et al., 2024). As we explain in the main text, our proof follows the one from (Koloskova et al., 2023; Takezawa et al., 2024), which follows the analysis of standard GD. Then, similarly to (Koloskova et al., 2023), we consider two possible situations: either $\|\nabla f(x^k)\| \geqslant L_0/L_1$ or $\|\nabla f(x^k)\| < L_0/L_1$. In the second case, the gradient is small, and $L_1$-term in $(L_0, L_1)$-smoothness is dominated by $L_0$-term, i.e., in inequality (8), the denominator of the left-hand side is $\mathcal{O}(L_0)$. In this case, the method behaves as standard GD for $L$-smooth problems with $L = \mathcal{O}(L_0)$. However, when $\|\nabla f(x^k)\| \geqslant L_0/L_1$, the $L_1$-term in $(L_0, L_1)$-smoothness is the leading one and this is the crucial difference between our proof and the one obtained by Koloskova et al. (2023): to handle this case, we use (8) and show that such situations lead to the decrease of $\|x^{k+1} - x^*\|^2$ by some positive constant $\nu\eta/(8L_1^2)$. In contrast, Koloskova et al. (2023) use traditional smoothness, which leads to worse complexity, as we explained in our first response. Moreover, Lemma 3.2 shows that the gradient norm is non-increasing along the trajectory of the method – similar to standard GD (e.g., see Lemma C.3 from (Gorbunov et al., 2022)).

**Comparison with (Chen et al., 2023).** Although Chen et al. (2023) analyze $\beta$-Normalized GD without assuming convexity, their results can be used to derive convergence bounds in the convex case as well. In particular, from convexity, we have

$$f(x^k) - f(x^*) \leqslant \langle \nabla f(x^k), x^k - x^* \rangle \leqslant \|\nabla f(x^k)\| \cdot \|x^k - x^*\|.$$

Although the proof from Chen et al. (2023) does not give a bound for $\|x^k - x^*\|$, one can assume for simplicity that it is bounded by $D > 0$ (e.g., we derive the boundedness of $\|x^k - x^*\|$ for $(L_0, L_1)$-GD in (45)). Then, using the bound for $\frac{1}{N}\sum_{k=0}^{N-1} \|\nabla f(x^k)\|$ from (Chen et al., 2023, Theorem 2), we get

$$\frac{1}{N}\sum_{k=0}^{N-1}(f(x^k) - f(x^*)) \leqslant \frac{2L_0 D(f(x^0) - f(x^*))}{N\varepsilon} + \frac{D\varepsilon}{2},$$

where $\varepsilon > 0$. In the original paper, $\varepsilon$ is the target accuracy. However, one can optimize it in the above bound and get $\varepsilon_{\text{opt}} = 2\sqrt{\frac{L_0(f(x_0) - f(x^*))}{N}}$ and

$$\frac{1}{N}\sum_{k=0}^{N-1}(f(x^k) - f(x^*)) \leqslant \frac{4D\sqrt{L_0(f(x^0) - f(x^*))}}{\sqrt{N}}.$$

The above rate is $\sqrt{N}$-worse than the leading term in the results we obtained for $(L_0, L_1)$-GD. This fact illustrate non-triviality of the extension of the known results from non-convex case to the convex case with the right dependence on $N$.

# E    MISSING PROOFS FOR GRADIENT DESCENT WITH POLYAK STEPSIZES

**Theorem E.1** (Theorem 4.1). *Let Assumptions 1 with $\mu = 0$ and 3 hold. Then, the iterates generated by* GD-PS *satisfy the following implication:*

$$\|\nabla f(x^k)\| \geqslant \frac{L_0}{L_1} \implies \|x^{k+1} - x^*\|^2 \leqslant \|x^k - x^*\|^2 - \frac{\nu^2}{16L_1^2}. \tag{54}$$

*Moreover, the output after $N$ steps the iterates satisfy*

$$\frac{4L_0}{\nu}\|x^{N+1} - x^*\|^2 + \sum_{k \in \{0,1,\ldots,N\} \setminus \mathcal{T}} \left( f(x^k) - f(x^*) \right) \leqslant \frac{4L_0}{\nu}\|x^0 - x^*\|^2 - \frac{\nu L_0 T}{4L_1^2}, \tag{55}$$

*where $\mathcal{T} := \{k \in \{0, 1, \ldots, N\} \mid \|\nabla f(x^k)\| \geqslant \frac{L_0}{L_1}\}$, $T := |\mathcal{T}|$, and if $N > T - 1$, it holds that*

$$f(\hat{x}^N) - f(x^*) \leqslant \frac{4L_0\|x^0 - x^*\|^2}{\nu(N - T + 1)} - \frac{\nu L_0 T}{4L_1^2(N - T + 1)} \tag{56}$$

*where $\hat{x}^N \in \{x^0, x^1, \ldots, x^N\}$ is such that $f(\hat{x}^N) = \min_{x \in \{x^0, x^1, \ldots, x^N\}} f(x)$. In particular, for $N > \frac{16L_1^2\|x^0 - x^*\|^2}{\nu^2} - 1$ inequality $N > T - 1$ is guaranteed and*

$$f(\hat{x}^N) - f(x^*) \leqslant \frac{4L_0\|x^0 - x^*\|^2}{\nu(N + 1)}. \tag{57}$$

*In addition, if $\mu > 0$ and $N > \frac{16L_1^2\|x^0 - x^*\|^2}{\nu^2} - 1$, then*

$$\|x^N - x^*\|^2 \leqslant \left(1 - \frac{\mu\nu}{8L_0}\right)^{N-T}\left(\|x^0 - x^*\|^2 - \frac{\nu^2 T}{16L_1^2}\right) \leqslant \left(1 - \frac{\mu\nu}{8L_0}\right)^{N-T}\|x^0 - x^*\|^2. \tag{58}$$

*Proof.* As for $(L_0, L_1)$-GD, we start by expanding the squared distance to the solution:

$$
\begin{aligned}
\|x^{k+1} - x^*\|^2 &= \left\| x^k - x^* - \frac{f(x^k) - f(x^*)}{\|\nabla f(x^k)\|^2}\nabla f(x^k) \right\|^2 \\
&= \|x^k - x^*\|^2 - \frac{2\left(f(x^k) - f(x^*)\right)}{\|\nabla f(x^k)\|^2}\langle x^k - x^*, \nabla f(x^k) \rangle + \frac{\left(f(x^k) - f(x^*)\right)^2}{\|\nabla f(x^k)\|^2} \\
&\overset{(2)}{\leqslant} \|x^k - x^*\|^2 - \frac{\left(f(x^k) - f(x^*)\right)^2}{\|\nabla f(x^k)\|^2} \\
&\overset{(8)}{\leqslant} \|x^k - x^*\|^2 - \frac{\nu}{2} \cdot \frac{f(x^k) - f(x^*)}{L_0 + L_1\|\nabla f(x^k)\|}.
\end{aligned} \tag{59}
$$

To continue the derivation, we consider two possible cases: $\|\nabla f(x^k)\| \geqslant \frac{L_0}{L_1}$ or $\|\nabla f(x^k)\| < \frac{L_0}{L_1}$.

**Case 1:** $\|\nabla f(x^k)\| \geqslant \frac{L_0}{L_1}$. In this case, inequalities (46) and (47) hold and the derivation from (59) can be continued as follows:

$$
\begin{aligned}
\|x^{k+1} - x^*\|^2 &\overset{(59),(46)}{\leqslant} \|x^k - x^*\|^2 - \frac{\nu}{2} \cdot \frac{f(x^k) - f(x^*)}{2L_1\|\nabla f(x^k)\|} \\
&\overset{(47)}{\leqslant} \|x^k - x^*\|^2 - \frac{\nu^2}{16L_1^2},
\end{aligned} \tag{60}
$$

which gives (54).

**Case 2:** $\|\nabla f(x^k)\| < \frac{L_0}{L_1}$. In this case, inequality (50) holds and we have

$$\|x^{k+1} - x^*\|^2 \overset{(59),(50)}{\leqslant} \|x^k - x^*\|^2 - \frac{\nu}{2} \cdot \frac{f(x^k) - f(x^*)}{2L_0}. \tag{61}$$

Next, we introduce the set of indices $\mathcal{T} := \{k \in \{0, 1, \dots, N\} \mid \|\nabla f(x^k)\| \geqslant \frac{L_0}{L_1}\}$ of size $T := |\mathcal{T}|$. In view of the above derivations, if $k \in \mathcal{T}$, inequality (60) holds, and if $k \in \{0, 1, \dots, N\} \setminus \mathcal{T}$, inequality (61) is satisfied. Therefore, unrolling the pair of inequalities (60) and (61), we get

$$\|x^{N+1} - x^*\|^2 \quad \leqslant \quad \|x^0 - x^*\|^2 - \frac{\nu^2 T}{16 L_1^2} - \frac{\nu}{4 L_0} \sum_{k \in \{0,1,\dots,N\} \setminus \mathcal{T}} \left( f(x^k) - f(x^*) \right),$$

which is equivalent to (55). Therefore, if $N > T - 1$, set $\{0, 1, \dots, N\} \setminus \mathcal{T}$ is non-empty and the above inequality implies

$$f(\hat{x}^N) - f(x^*) \quad \leqslant \quad \frac{1}{N - T + 1} \sum_{k \in \{0,1,\dots,N\} \setminus \mathcal{T}} \left( f(x^k) - f(x^*) \right)$$

$$\leqslant \quad \frac{4 L_0 \|x^0 - x^*\|^2}{\nu(N - T + 1)} - \frac{\nu L_0 T}{4 L_1^2 (N - T + 1)},$$

where $\hat{x}^N \in \{x^0, x^1, \dots, x^N\}$ is such that $f(\hat{x}^N) = \min_{x \in \{x^0, x^1, \dots, x^N\}} f(x)$. Moreover, since the left-hand side of (55) is non-negative, we have $T \leqslant \frac{16 L_1^2 \|x^0 - x^*\|^2}{\nu^2}$. Therefore, for $N > \frac{16 L_1^2 \|x^0 - x^*\|^2}{\nu^2} - 1$ inequality $N > T - 1$ is guaranteed as well as (56). Finally, to derive (57), we consider the right-hand side of (56) as a function of $T$:

$$\varphi(T) \quad := \quad \frac{4 L_0 \|x^0 - x^*\|^2}{\nu(N - T + 1)} - \frac{\nu L_0 T}{4 L_1^2 (N - T + 1)}$$

$$= \quad \frac{4 L_0 \|x^0 - x^*\|^2}{\nu(N - T + 1)} + \frac{\nu L_0}{4 L_1^2} - \frac{\nu L_0 (N + 1)}{4 L_1^2 (N - T + 1)}.$$

The derivative of this function equals

$$\varphi'(T) \quad = \quad \frac{4 L_0 \|x^0 - x^*\|^2}{\nu(N - T + 1)^2} - \frac{\nu L_0 (N + 1)}{4 L_1^2 (N - T + 1)^2}.$$

Since $N > \frac{16 L_1^2 \|x^0 - x^*\|^2}{\nu^2} - 1$, we have $\varphi'(T) < 0$, i.e., $\varphi(T)$ is a decreasing function of $T$. Therefore, since $T \geqslant 0$, we have that

$$\varphi(T) \leqslant \varphi(0) \iff \frac{4 L_0 \|x^0 - x^*\|^2}{\nu(N - T + 1)} - \frac{\nu L_0 T}{4 L_1^2 (N - T + 1)} \leqslant \frac{4 L_0 \|x^0 - x^*\|^2}{\nu(N + 1)}.$$

Combining the above inequality with (56), we obtain (57).

To prove (58), we notice that for $\mu > 0$ we have $f(x^k) - f(x^*) \geqslant \frac{\mu}{2} \|x^k - x^*\|^2$ implying

$$\|x^{k+1} - x^*\|^2 \overset{(61)}{\leqslant} \left( 1 - \frac{\mu \nu}{8 L_0} \right) \|x^k - x^*\|^2, \tag{62}$$

when $\|\nabla f(x^k)\| < \frac{L_0}{L_1}$. Unrolling the pair of inequalities (60) and (62), we get for $N > T$

$$\|x^N - x^*\|^2 \leqslant \left( 1 - \frac{\mu \nu}{8 L_0} \right)^{N-T} \|x^0 - x^*\|^2 - \frac{\nu^2}{16 L_1^2} \sum_{k \in \mathcal{T}} \left( 1 - \frac{\mu \nu}{8 L_0} \right)^{t_k},$$

where $t_k$ is the cardinality of $\{k + 1, \dots, N\} \setminus \mathcal{T}$. Since $|\mathcal{T}| = T$ we have that $t_k \leqslant N - T$ for all $k \in \mathcal{T}$. Therefore, we can continue the derivation as follows:

$$\|x^N - x^*\|^2 \quad \leqslant \quad \left( 1 - \frac{\mu \nu}{8 L_0} \right)^{N-T} \left( \|x^0 - x^*\|^2 - \frac{\nu^2 T}{16 L_1^2} \right)$$

$$\leqslant \quad \left( 1 - \frac{\mu \nu}{8 L_0} \right)^{N-T} \|x^0 - x^*\|^2,$$

which gives (58) and concludes the proof. $\qquad \square$

# F  MISSING PROOFS FOR $(L_0, L_1)$-SIMILAR TRIANGLES METHOD

**Lemma F.1** (Lemma E.1 from (Gorbunov et al., 2020))**.** *Let sequences $\{\alpha_k\}_{k\geqslant 0}$ and $\{A_k\}_{k\geqslant 0}$ be defined as follows:*

$$\alpha_0 = A_0 = 0, \quad \alpha_{k+1} = \frac{\eta(k+2)}{2}, \quad A_{k+1} = A_k + \alpha_{k+1}, \quad \forall k \geqslant 0.$$

*Then, for all $k \geqslant 0$*

$$A_{k+1} \;\geqslant\; \frac{\eta(k+1)(k+4)}{4}, \tag{63}$$

$$A_{k+1} \;\geqslant\; \frac{\alpha_{k+1}^2}{\eta}. \tag{64}$$

**Lemma F.2** (Lemma 5.1)**.** *Let $f$ satisfy Assumptions 1 with $\mu = 0$ and 3. Then, the iterates generated by $(L_0, L_1)$-STM with $0 < \eta \leqslant \frac{\nu}{2}$, $\nu = e^{-\nu}$ satisfy for all $N \geqslant 0$*

$$A_N \left(f(y^N) - f(x^*)\right) + \frac{G_N}{2} R_N^2 \;\leqslant\; \frac{G_1}{2} R_0^2 + \sum_{k=1}^{N-1} \frac{G_{k+1} - G_k}{2} R_k^2 \tag{65}$$

$$-\sum_{k=0}^{N-1} \frac{\alpha_{k+1}^2}{4G_{k+1}} \|\nabla f(x^{k+1})\|^2, \tag{66}$$

*where $R_k := \|z^k - x^*\|$ for all $k \geqslant 0$.*

*Proof.* The proof follows the one of Lemma F.4 from (Gorbunov et al., 2020). From the update rule, we have $z^{k+1} = z^k - \frac{\alpha_{k+1}}{G_{k+1}} \nabla f(x^{k+1})$ and

$$
\begin{aligned}
\alpha_{k+1}\langle \nabla f(x^{k+1}), z^k - x^* \rangle &= \alpha_{k+1}\langle \nabla f(x^{k+1}), z^k - z^{k+1} \rangle + \alpha_{k+1}\langle \nabla f(x^{k+1}), z^{k+1} - x^* \rangle \\
&= \alpha_{k+1}\langle \nabla f(x^{k+1}), z^k - z^{k+1} \rangle + G_{k+1}\langle z^{k+1} - z^k, x^* - z^{k+1} \rangle \\
&= \alpha_{k+1}\langle \nabla f(x^{k+1}), z^k - z^{k+1} \rangle - \frac{G_{k+1}}{2}\|z^k - z^{k+1}\|^2 \\
&\quad + \frac{G_{k+1}}{2}\|z^k - x^*\|^2 - \frac{G_{k+1}}{2}\|z^{k+1} - x^*\|^2.
\end{aligned}
$$

The update rules for $y^{k+1}$ and $x^{k+1}$ imply

$$A_{k+1}(y^{k+1} - x^{k+1}) = \alpha_{k+1}(z^{k+1} - z^k). \tag{67}$$

Moreover, to proceed, we will need the following upper-bound:

$$
\begin{aligned}
\exp\left(L_1 \|x^{k+1} - y^{k+1}\|\right) &\overset{(67)}{=} \exp\left(\frac{L_1 \alpha_{k+1} \|z^{k+1} - z^k\|}{A_{k+1}}\right) \\
&= \exp\left(\frac{\alpha_{k+1}^2 L_1 \|\nabla f(x^{k+1})\|}{A_{k+1}(L_0 + L_1\|\nabla f(x^{k+1})\|)}\right) \leqslant \exp\left(\frac{\alpha_{k+1}^2}{A_{k+1}}\right) \\
&\overset{(64)}{\leqslant} \exp(\eta) \overset{\eta \leqslant \nu}{\leqslant} \exp(\nu) \overset{\nu = e^{-\nu}}{=} \nu.
\end{aligned}
\tag{68}
$$

Using these formulas, we continue the derivation as follows:

$$
\begin{aligned}
\alpha_{k+1}\langle \nabla f(x^{k+1}), z^k - x^* \rangle \quad = \quad & A_{k+1}\langle \nabla f(x^{k+1}), x^{k+1} - y^{k+1} \rangle - \frac{G_{k+1}}{2}\|z^k - z^{k+1}\|^2 \\
& + \frac{G_{k+1}}{2}\|z^k - x^*\|^2 - \frac{G_{k+1}}{2}\|z^{k+1} - x^*\|^2 \\
\overset{(7)}{\leqslant} \quad & A_{k+1}\left( f(x^{k+1}) - f(y^{k+1}) \right) \\
& + \frac{A_{k+1}G_{k+1}\exp\left( L_1\|x^{k+1} - y^{k+1}\| \right)}{2}\|x^{k+1} - y^{k+1}\|^2 \\
& - \frac{G_{k+1}}{2}\|z^k - z^{k+1}\|^2 + \frac{G_{k+1}}{2}\|z^k - x^*\|^2 \\
& - \frac{G_{k+1}}{2}\|z^{k+1} - x^*\|^2 \\
\overset{(67),(68)}{\leqslant} \quad & A_{k+1}\left( f(x^{k+1}) - f(y^{k+1}) \right) + \frac{G_{k+1}}{2}\cdot\frac{\nu\alpha_{k+1}^2}{A_{k+1}}\|z^k - z^{k+1}\|^2 \\
& - \frac{G_{k+1}}{2}\|z^k - z^{k+1}\|^2 + \frac{G_{k+1}}{2}\|z^k - x^*\|^2 \\
& - \frac{G_{k+1}}{2}\|z^{k+1} - x^*\|^2 \\
= \quad & A_{k+1}\left( f(x^{k+1}) - f(y^{k+1}) \right) \\
& + \frac{G_{k+1}}{2}\left( \frac{\nu\alpha_{k+1}^2}{A_{k+1}} - 1 \right)\|z^k - z^{k+1}\|^2 \\
& + \frac{G_{k+1}}{2}\|z^k - x^*\|^2 - \frac{G_{k+1}}{2}\|z^{k+1} - x^*\|^2 \\
\overset{(64),\eta\leqslant\frac{\nu}{2}}{\leqslant} \quad & A_{k+1}\left( f(x^{k+1}) - f(y^{k+1}) \right) - \frac{G_{k+1}}{4}\|z^k - z^{k+1}\|^2 \\
& + \frac{G_{k+1}}{2}\|z^k - x^*\|^2 - \frac{G_{k+1}}{2}\|z^{k+1} - x^*\|^2. \quad (69)
\end{aligned}
$$

Next, using the definition of $x^{k+1}$ and $A_{k+1} = A_k + \alpha_{k+1}$, we get

$$
\alpha_{k+1}(x^{k+1} - z^k) = A_k(y^k - x^{k+1}). \quad (70)
$$

Combining the established inequalities, we obtain

$$
\begin{aligned}
\alpha_{k+1}\langle \nabla f(x^{k+1}), x^{k+1} - x^* \rangle \quad = \quad & \alpha_{k+1}\langle \nabla f(x^{k+1}), x^{k+1} - z^k \rangle \\
& + \alpha_{k+1}\langle \nabla f(x^{k+1}), z^k - x^* \rangle \\
\overset{(69),(70)}{\leqslant} \quad & A_k\langle \nabla f(x^{k+1}), y^k - x^{k+1} \rangle \\
& + A_{k+1}\left( f(x^{k+1}) - f(y^{k+1}) \right) - \frac{G_{k+1}}{4}\|z^k - z^{k+1}\|^2 \\
& + \frac{G_{k+1}}{2}\|z^k - x^*\|^2 - \frac{G_{k+1}}{2}\|z^{k+1} - x^*\|^2 \\
\overset{(2)}{\leqslant} \quad & A_k\left( f(y^k) - f(x^{k+1}) \right) + A_{k+1}\left( f(x^{k+1}) - f(y^{k+1}) \right) \\
& - \frac{G_{k+1}}{4}\|z^k - z^{k+1}\|^2 + \frac{G_{k+1}}{2}\|z^k - x^*\|^2 \\
& - \frac{G_{k+1}}{2}\|z^{k+1} - x^*\|^2,
\end{aligned}
$$

which can be rewritten as

$$
\begin{aligned}
A_{k+1} f(y^{k+1}) - A_k f(y^k) \ \leqslant\ & \alpha_{k+1}\left(f(x^{k+1}) + \langle \nabla f(x^{k+1}), x^* - x^{k+1}\rangle\right) \\
& + \frac{G_{k+1}}{2}\|z^k - x^*\|^2 - \frac{G_{k+1}}{2}\|z^{k+1} - x^*\|^2 - \frac{\alpha_{k+1}^2}{4G_{k+1}}\|\nabla f(x^{k+1})\|^2 \\
\overset{(2)}{\leqslant}\ & \alpha_{k+1} f(x^*) \\
& + \frac{G_{k+1}}{2}\|z^k - x^*\|^2 - \frac{G_{k+1}}{2}\|z^{k+1} - x^*\|^2 - \frac{\alpha_{k+1}^2}{4G_{k+1}}\|\nabla f(x^{k+1})\|^2 .
\end{aligned}
$$

Summing up the above inequality for $k = 0, 1, \ldots, N-1$ and using $A_0 = \alpha_0 = 0$, $\sum_{k=0}^{N-1}\alpha_{k+1} = A_N$, and new notation $R_k := \|z^k - x^*\|$, we derive

$$
A_N\left(f(y^N) - f(x^*)\right) + \frac{G_N}{2}R_N^2 \ \leqslant\ \frac{G_1}{2}R_0^2 + \sum_{k=1}^{N-1}\frac{G_{k+1} - G_k}{2}R_k^2 - \sum_{k=0}^{N-1}\frac{\alpha_{k+1}^2}{4G_{k+1}}\|\nabla f(x^{k+1})\|^2 ,
$$

which finishes the proof. $\qquad\square$

**Theorem F.1** (Theorem 5.1). *Let $f$ satisfy Assumptions 1 with $\mu = 0$ and 3. Then, the iterates generated by $(L_0, L_1)$-STM with $0 < \eta \leqslant \frac{\nu}{2}$, $\nu = e^{-\nu}$, $G_1 = L_0 + L_1\|\nabla f(x^0)\|$, and*

$$
G_{k+1} = \max\{G_k, L_0 + L_1\|\nabla f(x^{k+1})\|\}, \quad k \geqslant 0, \tag{71}
$$

*satisfy*

$$
f(y^N) - f(x^*) \leqslant \frac{2L_0(1 + L_1\|x^0 - x^*\|\exp(L_1\|x^0 - x^*\|))\|x^0 - x^*\|^2}{\eta N(N+3)}. \tag{72}
$$

*Proof.* Let us prove by induction that $R_k \leqslant R_0$ for all $k \geqslant 0$. For $k = 0$, the statement is trivial. Next, we assume that the statement holds for $k = N$ and derive that it also holds for $k = N + 1$. Indeed, from Lemma 5.1 we have

$$
\begin{aligned}
\frac{G_{N+1}}{2}R_{N+1}^2 \ \leqslant\ & A_{N+1}\left(f(y^{N+1}) - f(x^*)\right) + \frac{G_{N+1}}{2}R_{N+1}^2 \\
\overset{(21)}{\leqslant}\ & \frac{G_1}{2}R_0^2 + \sum_{k=1}^{N}\frac{G_{k+1} - G_k}{2}R_k^2 \\
\leqslant\ & \frac{G_1}{2}R_0^2 + \sum_{k=1}^{N}\frac{G_{k+1} - G_k}{2}R_0^2 = \frac{G_{N+1}}{2}R_0^2,
\end{aligned} \tag{73}
$$

implying that $R_{N+1} \leqslant R_0$. That is, we proved that $R_k \leqslant R_0$ for all $k \geqslant 0$, i.e., the sequence $\{z^k\}_{k\geqslant 0}$ stays in $B_{R_0}(x^*) := \{x \in \mathbb{R}^d \mid \|x - x^*\| \leqslant R_0\}$. Since $x^0 = y^0 = z^0$, $x^{k+1}$ is a convex combination of $y^k$ and $z^k$, $y^{k+1}$ is a convex combination of $y^k$ and $z^{k+1}$, we also have that sequences $\{x^k\}_{k\geqslant 0}$ and $\{y^k\}_{k\geqslant 0}$ stay in $B_{R_0}(x^*)$, which can be formally shown using an induction argument. Therefore, we can upper-bound $G_k$ for all $k \geqslant 0$ as follows

$$
\begin{aligned}
G_k \ =\ & L_0 + L_1\max_{t=0,\ldots,k}\|\nabla f(x^t)\| \overset{(6)}{\leqslant} L_0 + L_1 L_0\max_{t=0,\ldots,k}\exp(L_1\|x^t - x^*\|)\|x^t - x^*\| \\
\leqslant\ & L_0\left(1 + L_1 R_0\exp(L_1 R_0)\right).
\end{aligned} \tag{74}
$$

Moreover, from (73) we also have

$$
f(y^N) - f(x^*) \ \leqslant\ \frac{G_N R_0^2}{2A_N} \overset{(74),(63)}{\leqslant} \frac{2L_0(1 + L_1 R_0\exp(L_1 R_0))R_0^2}{\eta N(N+3)},
$$

which finishes the proof. $\qquad\square$

# G  MISSING PROOFS FOR ADAPTIVE GRADIENT DESCENT

## G.1  DERIVATION OF (25)

The key lemma about the convergence of AdGD holds for any convex function regardless of the smoothness properties.

**Lemma G.1** (Lemma 1 from Malitsky & Mishchenko (2020))**.** *Let Assumption 1 with $\mu = 0$ hold, and $x^*$ be any minimizer of $f$. Then, the iterates generated by Algorithm 4 with $\gamma = \frac{1}{2}$ satisfy*

$$\|x^{k+1} - x^*\|^2 + \frac{1}{2}\|x^{k+1} - x^k\|^2 + 2\lambda_k(1 + \theta_k)(f(x^k) - f(x^*))$$
$$\leqslant \|x^k - x^*\|^2 + \frac{1}{2}\|x^k - x^{k-1}\|^2 + 2\lambda_k\theta_k(f(x^{k-1}) - f(x^*)). \quad (75)$$

In particular, the above lemma implies boundedness of $\|x^k - x^*\|$ and $\|x^k - x^{k-1}\|$, which allows us to get the upper bound on the gradient norm (24) and a lower bound for $\lambda_k$ as stated in the paragraph before (25). For completeness, we provide a detailed statement of the result and its proof below.

**Theorem G.1.** *Let Assumptions 1 with $\mu = 0$ and 3 hold. For all $N \geqslant 1$ we define point $\hat{x}^N := \frac{1}{S_N}\left(\lambda_N(1 + \theta_N) + \sum_{k=1}^{N} w_k x^k\right)$, where $w_k := \lambda_k(1 + \theta_k) - \lambda_{k+1}\theta_{k+1}$, $S_N := \lambda_1\theta_1 + \sum_{k=1}^{N}\lambda_k$, and $\{x^k\}_{k \geqslant 0}$ are the iterates produced by AdGD with $\gamma = 1/2$. Then, $\hat{x}^N$ satisfies*

$$f(\hat{x}^N) - f(x^*) \leqslant \frac{L_0(1 + L_1 D \exp(L_1 D)) \exp\left(\sqrt{2}L_1 D\right) D^2}{N}, \quad (76)$$

*where $D > 0$ and $D^2 := \|x^1 - x^*\|^2 + \frac{1}{2}\|x^1 - x^0\|^2 + 2\lambda_1\theta_1(f(x^0) - f(x^*))$.*

*Proof.* The proof follows almost the same lines as the proof from (Malitsky & Mishchenko, 2020). Telescoping inequality (75), we get

$$\|x^{k+1} - x^*\|^2 + \frac{1}{2}\|x^{k+1} - x^k\|^2 + 2\lambda_k(1 + \theta_k)(f(x^k) - f(x^*))$$
$$+ 2\sum_{i=1}^{k-1}\left[\lambda_i(1 + \theta_i) - \lambda_{i+1}\theta_{i+1}\right](f(x^i) - f(x^*))$$
$$\leqslant \|x^1 - x^*\|^2 + \frac{1}{2}\|x^1 - x^0\|^2 + 2\lambda_1\theta_1(f(x^0) - f(x^*)). \quad (77)$$

Since $\lambda_i(1 + \theta_i) - \lambda_{i+1}\theta_{i+1} \geqslant 0$ by definition of $\lambda_i$, we conclude that the term in the second line of the above inequality is non-negative. Therefore, for any $k \geqslant 1$ we have

$$\|x^k - x^*\|^2 \quad \leqslant \quad D^2, \quad (78)$$
$$\|x^k - x^{k-1}\|^2 \quad \leqslant \quad 2D^2. \quad (79)$$

Using Jensen's inequality in (77), we derive

$$S_k(f(\hat{x}^k) - f(x_*)) \leqslant \frac{D^2}{2},$$

where

$$\hat{x}^k \quad = \quad \frac{\lambda_k(1 + \theta_k)x^k + \sum_{i=1}^{k-1}w_i x^i}{S_k}, \quad (80)$$

$$w_k \quad = \quad \lambda_i(1 + \theta_i) - \lambda_{i+1}\theta_{i+1}, \quad (81)$$

$$S_k \quad = \quad \lambda_1\theta_1 + \sum_{i=1}^{k}\lambda_i. \quad (82)$$

Thus, we have

$$f(\hat{x}^k) - f_* \leqslant \frac{D^2}{2S_k}. \quad (83)$$

Next, we notice that for any $k \geqslant 1$

$$\frac{\|x^k - x^{k-1}\|}{\|\nabla f(x^k) - \nabla f(x^{k-1})\|} \overset{(6)}{\geqslant} \frac{1}{(L_0 + L_1\|\nabla f(x^k)\|)\exp(L_1\|x^k - x^{k-1}\|)}$$

$$\overset{(24)}{\geqslant} \frac{1}{L_0(1 + L_1 D\exp(L_1 D))\exp(\sqrt{2}L_1 D)}.$$

Since $\theta_0 = +\infty$, we have $\lambda_1 = \frac{\|x^1 - x^0\|}{2\|\nabla f(x^1) - \nabla f(x^0)\|}$. Moreover, for $k > 1$ we have either $\lambda_k \geqslant \lambda_{k-1}$ or $\lambda_k = \frac{\|x^k - x^{k-1}\|}{2\|\nabla f(x^k) - \nabla f(x^{k-1})\|}$. Therefore, by induction we can prove that

$$\lambda_k \geqslant \frac{1}{2L_0(1 + L_1 D\exp(L_1 D))\exp(\sqrt{2}L_1 D)} \tag{84}$$

that implies

$$S_k = \lambda_1\theta_1 + \sum_{i=1}^{k}\lambda_i \geqslant \frac{k}{2L_0(1 + L_1 D\exp(L_1 D))\exp(\sqrt{2}L_1 D)}.$$

Therefore, we have

$$f(\hat{x}^k) - f(x^*) \leqslant \frac{D^2}{2S_k} \leqslant \frac{L_0(1 + L_1 D\exp(L_1 D))\exp(\sqrt{2}L_1 D)D^2}{k},$$

which is equivalent to (76) when $k = N$. $\qquad\square$

## G.2 PROOF OF THEOREM 6.1

To show an improved result, we consider Algorithm 4 with $\gamma = \frac{1}{4}$ and refine Lemma G.1 as follows.

**Lemma G.2.** *Let Assumption 1 with $\mu = 0$ hold, and $x^*$ be any minimizer of $f$. Then, the iterates generated by Algorithm 4 with $\gamma = \frac{1}{4}$ satisfy for all $k \geqslant 1$*

$$\|x^{k+1} - x^*\|^2 + \frac{1}{4}\|x^{k+1} - x^k\|^2 + 2\lambda_k(1 + \theta_k)(f(x^k) - f(x^*)) + \frac{1}{2}\sum_{i=0}^{k}\|x^{i+1} - x^i\|^2$$

$$\leqslant \|x^k - x^*\|^2 + \frac{1}{4}\|x^k - x^{k-1}\|^2 + 2\lambda_k\theta_k(f(x^{k-1}) - f(x^*)) + \frac{1}{2}\sum_{i=0}^{k-1}\|x^{i+1} - x^i\|^2. \tag{85}$$

*Proof.* The proof is almost identical to the one from (Malitsky & Mishchenko, 2020) and starts as the standard proof for GD:

$$\|x^{k+1} - x^*\|^2 = \|x^k - x^*\|^2 + 2\langle x^{k+1} - x^k, x^k - x^*\rangle + \|x^{k+1} - x^k\|^2$$

$$= \|x^k - x^*\|^2 + 2\lambda_k\langle\nabla f(x^k), x^* - x^k\rangle + \|x^{k+1} - x^k\|^2.$$

$$\overset{(2)}{\leqslant} \|x^k - x^*\|^2 + 2\lambda_k(f(x^*) - f(x^k)) + \|x^{k+1} - x^k\|^2.$$

Introducing $\Sigma_{k+1} = \frac{1}{2}\sum_{i=0}^{k}\|x^{i+1} - x^i\|^2$, we rewrite the above inequality as

$$\|x^{k+1} - x^*\|^2 + \Sigma_{k+1} \leqslant \|x^k - x^*\|^2 - 2\lambda_k(f(x^k) - f(x^*)) + \|x^{k+1} - x^k\|^2$$

$$+\Sigma_k + \frac{1}{2}\|x^{k+1} - x^k\|^2. \tag{86}$$

Next, we transform $\|x^{k+1} - x^k\|^2$ similarly to the original proof:

$$\|x^{k+1} - x^k\|^2 = 2\|x^{k+1} - x^k\|^2 - \|x^{k+1} - x^k\|^2$$

$$= -2\lambda_k\langle\nabla f(x^k), x^{k+1} - x^k\rangle - \|x^{k+1} - x^k\|^2$$

$$= 2\lambda_k\langle\nabla f(x^k) - \nabla f(x^{k-1}), x^k - x^{k+1}\rangle$$

$$+2\lambda_k\langle\nabla f(x^{k-1}), x^k - x^{k+1}\rangle - \|x^{k+1} - x^k\|^2. \tag{87}$$

Next, we apply Cauchy-Schwarz inequality, the definition of $\lambda_k$ with $\gamma = \frac{1}{4}$, and Young's inequality to estimate the first inner-product in the right-hand side:

$$
\begin{aligned}
2\lambda_k \langle \nabla f(x^k) - \nabla f(x^{k-1}), x^k - x^{k+1} \rangle &\leqslant 2\lambda_k \|\nabla f(x^k) - \nabla f(x^{k-1})\| \|x^k - x^{k+1}\| \\
&\leqslant \frac{1}{2}\|x^k - x^{k-1}\| \|x^k - x^{k+1}\| \\
&\leqslant \frac{1}{4}\|x^k - x^{k-1}\|^2 + \frac{1}{4}\|x^{k+1} - x^k\|^2.
\end{aligned} \tag{88}
$$

Then, using the convexity of $f$, we handle the second inner product from the right-hand side of (87):

$$
\begin{aligned}
2\lambda_k \langle \nabla f(x^{k-1}), x^k - x^{k+1} \rangle &= \frac{2\lambda_k}{\lambda_{k-1}} \langle x^{k-1} - x^k, x^k - x^{k+1} \rangle \\
&= 2\lambda_k \theta_k \langle x^{k-1} - x^k, \nabla f(x^k) \rangle \\
&\leqslant 2\lambda_k \theta_k (f(x^{k-1}) - f(x^k)).
\end{aligned} \tag{89}
$$

Plugging (88) and (89) in (87), we get

$$
\|x^{k+1} - x^k\|^2 \leqslant \frac{1}{4}\|x^k - x^{k-1}\|^2 - \frac{3}{4}\|x^{k+1} - x^k\|^2 + 2\lambda_k \theta_k (f(x^{k-1}) - f(x^k)).
$$

Finally, using the above upper bound for $\|x^{k+1} - x^k\|^2$ in (86), we obtain

$$
\begin{aligned}
\|x^{k+1} - x^*\|^2 + \Sigma_{k+1} &\leqslant \|x^k - x^*\|^2 - 2\lambda_k(f(x^k) - f(x^*)) \\
&\quad + \frac{1}{4}\|x^k - x^{k-1}\|^2 - \frac{3}{4}\|x^{k+1} - x^k\|^2 + 2\lambda_k \theta_k (f(x^{k-1}) - f(x^k)) \\
&\quad + \Sigma_k + \frac{1}{2}\|x^{k+1} - x^k\|^2 \\
&= \|x^k - x^*\|^2 + \frac{1}{4}\|x^k - x^{k-1}\|^2 + 2\lambda_k \theta_k (f(x^{k-1}) - f(x^*)) + \Sigma_k \\
&\quad - \frac{1}{4}\|x^{k+1} - x^k\|^2 - 2\lambda_k(1 + \theta_k)(f(x^k) - f(x^*)).
\end{aligned}
$$

Rearranging the terms, we derive (85). $\qquad\square$

The above lemma implies not only the boundedness of the iterates but also the boundedness of $\sum_{i=0}^{k-1}\|x^{i+1} - x^i\|^2$ for $k \geqslant 1$.

**Corollary 1.** *Let Assumption 1 with $\mu = 0$ hold, and $x^*$ be any minimizer of $f$. Then, the iterates generated by Algorithm 4 with $\gamma = \frac{1}{4}$ satisfy for all $k \geqslant 1$*

$$
\|x^{k+1} - x^*\|^2 \leqslant D^2, \tag{90}
$$

$$
\|x^{k+1} - x^k\|^2 \leqslant 4D^2, \tag{91}
$$

$$
\sum_{i=0}^{k-1}\|x^{i+1} - x^i\|^2 \leqslant 2D^2, \tag{92}
$$

*where $D > 0$ and $D^2 := \|x^1 - x^*\|^2 + \frac{3}{4}\|x^1 - x^0\|^2 + 2\lambda_1\theta_1(f(x^0) - f(x^*))$.*

Using the above results, we derive the following theorem.

**Theorem G.2** (Theorem 6.1)**.** *Let Assumptions 1 with $\mu = 0$ and 3 hold. For all $N \geqslant 1$ we define point $\hat{x}^N := \frac{1}{S_N}\left(\lambda_N(1 + \theta_N) + \sum_{k=1}^N w_k x^k\right)$, where $w_k := \lambda_k(1 + \theta_k) - \lambda_{k+1}\theta_{k+1}$, $S_N := \lambda_1\theta_1 + \sum_{k=1}^N \lambda_k$, and $\{x^k\}_{k\geqslant 0}$ are the iterates produced by AdGD with $\gamma = 1/4$. Then, for $N > mK - \frac{\sqrt{2N}(m+1)L_1 D}{\nu}$ iterate $\hat{x}^N$ satisfies*

$$
f(\hat{x}^N) - f(x^*) \leqslant \frac{2L_0 D^2}{\nu(N - mK) - \sqrt{2N}(m+1)L_1 D}, \tag{93}
$$

where $D > 0$ and $D^2 := \|x^1 - x^*\|^2 + \frac{3}{4}\|x^1 - x^0\|^2 + 2\lambda_1\theta_1(f(x^0) - f(x^*))$, $m := 1 + \log_{\sqrt{2}}\left\lceil \frac{(1+L_1 D \exp(2L_1 D))}{2}\right\rceil$, $K := \frac{2L_1^2 D^2}{\nu^2}$, and $\nu = e^{-\nu}$. In particular, for $N \geqslant \left(2mK + \frac{4(m+1)L_1 D}{\nu}\right)^2$, we have

$$f(\hat{x}^N) - f(x^*) \leqslant \frac{4L_0 D^2}{\nu N}. \tag{94}$$

*Proof.* Using Lemma 2.1, we obtain

$$\left\|\nabla f(x^k) - \nabla f(x^{k-1})\right\| \leqslant \|x^k - x^{k-1}\| \left(L_0 + L_1 \left\|\nabla f(x^k)\right\|\right) \exp\left(L_1\|x^k - x^{k-1}\|\right). \tag{95}$$

Moreover, since[9] $\theta_0 = +\infty$, we have $\lambda_1 = \frac{\|x^1 - x^0\|}{4\|\nabla f(x^1) - \nabla f(x^0)\|}$.

Next, for $k > 1$ we have either $\lambda_k = \sqrt{1 + \theta_{k-1}}\lambda_{k-1}$ or $\lambda_k = \frac{\|x^k - x^{k-1}\|}{4\|\nabla f(x^k) - \nabla f(x^{k-1})\|}$. For convenience of the analysis of these two options, we let $\mathcal{K}$ be the set of indices $k > 1$ such that $\lambda_k = \sqrt{1 + \theta_{k-1}}\lambda_{k-1}$ and $\lambda_{k-1} = \frac{\|x^{k-1} - x^{k-2}\|}{4\|\nabla f(x^{k-1}) - \nabla f(x^{k-2})\|}$.

**Option 1:** $\lambda_k = \sqrt{1 + \theta_{k-1}}\lambda_{k-1}$. Then, by definition of $\mathcal{K}$, there exists $\tau \geqslant 1$ and index $t$ such that $t \in \mathcal{K}$, $\lambda_l = \sqrt{1 + \theta_{l-1}}\lambda_{l-1}$ for all $l \in \{t, t+1, \ldots, t+\tau - 1\}$, $k \in \{t, t+1, \ldots, t+\tau - 1\}$, and $\lambda_{t+\tau} = \frac{\|x^{t+\tau} - x^{t+\tau-1}\|}{4\|\nabla f(x^{t+\tau}) - \nabla f(x^{t+\tau-1})\|}$, i.e., $k$ belongs to some sub-sequence of indices such that Option 1 holds. Following exactly the same steps as in the derivation of (84), we conclude that

$$\lambda_k \geqslant \frac{1}{2L_0 \exp(L_1 D)(1 + DL_1 \exp(L_1 D))} =: \lambda_{\min}$$

for any $k \geqslant 1$. Since $\theta_l \geqslant 1$ for all $l \in \{t, t+1, \ldots, t+\tau-1\}$, we get that $\lambda_l \geqslant \sqrt{2}\lambda_{l-1} \geqslant 2^{\frac{l-t}{2}}\lambda_t \geqslant 2^{\frac{l-t}{2}}\lambda_{t-1}$ for $l \in \{t+1, \ldots, t+\tau-1\}$, meaning that for $l - t$ larger than $1 + \log_{\sqrt{2}}\left\lceil \frac{\nu}{4L_0\lambda_{\min}}\right\rceil \leqslant 1 + \log_{\sqrt{2}}\left\lceil \frac{\nu \exp(L_1 D)(1 + DL_1 \exp(L_1 D))}{2}\right\rceil =: m$ we have $\lambda_l \geqslant \frac{\nu}{4L_0}$, where $\nu = e^{-\nu}$. Putting all together, we conclude that

$$\lambda_l \geqslant \begin{cases} \lambda_{t-1}, & \text{for } l \in \{t, t+1, \ldots, t+m\}, \\ \frac{\nu}{4L_0}, & \text{for } l \in \{t+m+1, t+m+2, \ldots, t+\tau-1\}. \end{cases} \tag{96}$$

**Option 2:** $\lambda_k = \frac{\|x^k - x^{k-1}\|}{4\|\nabla f(x^k) - \nabla f(x^{k-1})\|}$. Then, using (95), we get

$$\lambda_k = \frac{\|x^k - x^{k-1}\|}{4\|\nabla f(x^k) - \nabla f(x^{k-1})\|} \geqslant \frac{\exp(-L_1\|x^k - x^{k-1}\|)}{L_0 + L_1\|\nabla f(x^k)\|} = \frac{\lambda_k \exp(-L_1\|x^k - x^{k-1}\|)}{4(\lambda_k L_0 + L_1\|x^{k+1} - x^k\|)},$$

implying that

$$\lambda_k \geqslant \frac{\exp(-L_1\|x^k - x^{k-1}\|)}{4L_0} - \frac{L_1}{4L_0}\|x^{k+1} - x^k\|. \tag{97}$$

To continue the proof, we split the set of indices $\{1, 2, \ldots, N\}$ into three disjoint sets $\mathcal{T}_1, \mathcal{T}_2, \mathcal{T}_3$ defined as follows: $\mathcal{T}_2 := \left\{k \in \{1, 2, \ldots, N\} \mid \lambda_k = \frac{\|x^k - x^{k-1}\|}{4\|\nabla f(x^k) - \nabla f(x^{k-1})\|}\right\}$, $\mathcal{T}_1 := \left\{k \in \{1, 2, \ldots, N\} \mid \lambda_k = \sqrt{1 + \theta_{k-1}}\lambda_{k-1} \text{ and } \exists t \in \mathcal{K} \text{ such that } t \leqslant k \leqslant t + m\right\}$, $\mathcal{T}_3 := \{1, 2, \ldots, N\} \setminus (\mathcal{T}_1 \cup \mathcal{T}_2)$. Then, taking into account the lower bounds (96) and (97), we have $\forall k \in \{1, 2, \ldots, N\}$

$$\lambda_k \geqslant \begin{cases} \lambda_{t-1}, & \text{if } k \in \mathcal{T}_1, \text{ where } t \in \mathcal{K} \text{ and } 0 \leqslant k - t \leqslant m, \\ \frac{\exp(-L_1\|x^k - x^{k-1}\|)}{4L_0} - \frac{L_1}{4L_0}\|x^{k+1} - x^k\|, & \text{if } k \in \mathcal{T}_2, \\ \frac{\nu}{4L_0}, & \text{if } k \in \mathcal{T}_3, \end{cases}$$

$$\underset{t-1\in\mathcal{T}_2,\,(97)}{\geqslant} \begin{cases} \frac{\exp(-L_1\|x^{t-1} - x^{t-2}\|)}{4L_0} - \frac{L_1}{4L_0}\|x^t - x^{t-1}\|, & \text{if } k \in \mathcal{T}_1, \text{ where } t \in \mathcal{K} \text{ and } 0 \leqslant k - t \leqslant m, \\ \frac{\exp(-L_1\|x^k - x^{k-1}\|)}{4L_0} - \frac{L_1}{4L_0}\|x^{k+1} - x^k\|, & \text{if } k \in \mathcal{T}_2, \\ \frac{\nu}{4L_0}, & \text{if } k \in \mathcal{T}_3. \end{cases}$$

---

[9]In practice $\theta_0$ it is sufficient to take $\theta_0 \geqslant \frac{\|x^1 - x^0\|^2}{16\lambda_0^2\|\nabla f(x^1) - \nabla f(x^0)\|^2} - 1$.

Also the number of steps when $L_1\|x^k - x^{k-1}\| \geqslant \nu$ holds is bounded by

$$K := \frac{2L_1^2 D^2}{\nu^2},$$

since $\sum_{k=0}^{N} \|x^{k+1} - x^k\|^2 \leqslant 2D^2$. For convenience, we introduce a new set of indices $\mathcal{M} := \left\{ k \in \{1, 2, \ldots, N\} \mid L_1\|x^k - x^{k-1}\| \leqslant \nu \right\}$. Then, for $k \in \mathcal{M}$ we have

$$\exp\left(-L_1\|x^k - x^{k-1}\|\right) \geqslant \exp(-\nu) = \nu. \tag{98}$$

Therefore, we can lower bound the sum of stepsizes as follows:

$$
\begin{aligned}
\sum_{k=1}^{N} \lambda_k \;\geqslant\;& \sum_{k \in \mathcal{T}_1, t \in \mathcal{K}, 0 \leqslant k-t \leqslant m} \frac{\exp\left(-L_1\|x^t - x^{t-1}\|\right)}{4L_0} + \sum_{k \in \mathcal{T}_2} \frac{\exp\left(-L_1\|x^k - x^{k-1}\|\right)}{4L_0} \\
&+ \sum_{k \in \mathcal{T}_3} \frac{\nu}{4L_0} - \frac{L_1}{4L_0} \sum_{k \in \mathcal{T}_2} \|x^{k+1} - x^k\| - \frac{mL_1}{4L_0} \sum_{t \in \mathcal{T}_2 : t+1 \in \mathcal{K}} \|x^{t+1} - x^t\| \\
\geqslant\;& \sum_{k \in \mathcal{T}_1 \cap \mathcal{M}, t \in \mathcal{K}, 0 \leqslant k-t \leqslant m} \frac{\exp\left(-L_1\|x^t - x^{t-1}\|\right)}{4L_0} + \sum_{k \in \mathcal{T}_2 \cap \mathcal{M}} \frac{\exp\left(-L_1\|x^k - x^{k-1}\|\right)}{4L_0} \\
&+ \sum_{k \in \mathcal{T}_3} \frac{\nu}{4L_0} - \frac{(m+1)L_1}{4L_0} \sum_{k \in \mathcal{T}_2} \|x^{k+1} - x^k\| \\
\overset{(98)}{\geqslant}\;& \frac{\nu(N - mK)}{4L_0} - \frac{(m+1)L_1}{4L_0} \sum_{k=0}^{N} \|x^{k+1} - x^k\| \\
\geqslant\;& \frac{\nu(N - mK)}{4L_0} - \frac{(m+1)L_1}{4L_0} \sqrt{N \sum_{k=0}^{N} \|x^{k+1} - x^k\|^2} \\
\overset{(92)}{\geqslant}\;& \frac{\nu(N - mK)}{4L_0} - \frac{\sqrt{2N}(m+1)L_1 D}{4L_0}. 
\end{aligned}
\tag{99}
$$

Since $S_N \geqslant \sum_{k=1}^{N} \lambda_k$ (see the definition in (82)) we have from (83) and the above lower bound on $\sum_{k=1}^{N} \lambda_k$ that

$$f(\hat{x}^N) - f(x^*) \leqslant \frac{D^2}{2S_N} \overset{(99)}{\leqslant} \frac{2L_0 D^2}{\nu(N - mK) - \sqrt{2N}(m+1)L_1 D}. \tag{100}$$

In particular, we derived (93) under Assumption 3 holds, and when $N \geqslant \left(2mK + \frac{4(m+1)L_1 D}{\nu}\right)^2$, we have $\nu(N - mK) - \sqrt{2N}(m+1)L_1 D \geqslant \frac{\nu N}{2}$, which in combination with (100) implies (94). $\qquad\square$

### G.3 CONVERGENCE IN THE STRONGLY CONVEX CASE

To show an improved result in the strongly convex case ($\mu > 0$ in Assumptions 1), we consider Algorithm 4 with more a more conservative stepsize selection rule:

$$\lambda_k = \min\left\{ \sqrt{1 + \frac{3\theta_{k-1}}{4}} \lambda_{k-1}, \frac{\|x^k - x^{k-1}\|}{4\|\nabla f(x^k) - \nabla f(x^{k-1})\|} \right\}. \tag{101}$$

For these stepsizes, Lemma G.2 holds as well. However, in contrast to the convex case, we will use Assumption 2 instead of Assumption 3. The key reason for this is that we need to use (10) for $x = x^k$ and $y = x^*$ that not necessarily satisfy (9). In contrast, inequality (10) holds for any $x, y \in \mathbb{R}^d$ under the Assumption 2 and convexity.

**Theorem G.3.** *Let Assumptions 1 with $\mu > 0$ and 2 hold. For all $N \geqslant 1$ we define the Lyapunov function*

$$\Psi_k = \left(1 - \frac{\lambda_k \mu}{4}\right)\left\|x^k - x^*\right\|^2 + \frac{1}{4}\left(1 + (1 - \alpha^*)\frac{8\mu}{L_0}\right)\left\|x^k - x^{k-1}\right\|^2$$
$$+ 2\lambda_k \theta_k (f(x^{k-1}) - f_*),$$

*where $\{x^k\}_{k \geqslant 0}$ are the iterates produced by AdGD with $\lambda_k$ defined in (101), and $\alpha^* = \frac{73 - \sqrt{3281}}{16} \approx 0.98$. Then, for $N > \sqrt{2N}(m+1)L_1 D$ Lyapunov function $\Psi_{N+1}$ satisfies*

$$\Psi_{N+1} \leq \left(1 - \frac{\alpha^* \mu}{8L_0} + \frac{\alpha^* \mu (m+1)L_1 D}{4\sqrt{2N}L_0}\right)^N \Psi_1, \tag{102}$$

*where $D > 0$ and $D^2 := \|x^1 - x^*\|^2 + \frac{3}{4}\|x^1 - x^0\|^2 + 2\lambda_1 \theta_1 (f(x^0) - f(x^*))$, and $m := 1 + \log_{\sqrt{\frac{7}{4}}}\left\lceil \frac{1 + L_1 D}{2} \right\rceil$. In particular, for $N \geqslant 8(m+1)^2 L_1^2 D^2$, we have*

$$\Psi_{N+1} \leq \left(1 - \frac{\alpha^* \mu}{16L_0}\right)^N \Psi_1. \tag{103}$$

*Proof.* The proof follows the one from (Malitsky & Mishchenko, 2020). First of all, we note that the stricter inequality $\lambda_k \leqslant \sqrt{1 + \frac{3\theta_{k-1}}{4}}\lambda_{k-1}$ is not used in the derivation of Lemma G.2. Therefore, Lemma G.2 holds as well as Corollary 1. Next, we make certain steps in the analysis tighter to use the fact that $\mu > 0$. Strong convexity implies

$$\lambda_k \langle \nabla f(x^k), x^* - x^k \rangle \leqslant \lambda_k (f(x^*) - f(x^k)) - \lambda_k \frac{\mu}{2}\|x^* - x^k\|^2, \tag{104}$$

and $\left\|\nabla f(x^k) - \nabla f(x^{k-1})\right\| \geqslant \mu \left\|x^k - x^{k-1}\right\|$. The latter implies that $\lambda_k \leqslant \frac{1}{4\mu}$ for $k \geqslant 1$. Since Lemma 2.2 holds under Assumption 2 with $\nu = 1$ and without condition (9), and bound $\lambda_k \leq \frac{1}{4\mu}$ holds, we have

$$\lambda_k \langle \nabla f(x^k), x^* - x^k \rangle \overset{(10)}{\leqslant} \lambda_k (f(x^*) - f(x^k)) - \frac{\lambda_k}{2(L_0 + L_1\|\nabla f(x^*)\|)}\|\nabla f(x^k)\|^2$$
$$= \lambda_k (f_* - f(x^k)) - \frac{1}{2L_0 \lambda_k}\|x^{k+1} - x^k\|^2$$
$$\overset{\lambda_k \leq \frac{1}{4\mu}}{\leq} \lambda_k (f_* - f(x^k)) - \frac{2\mu}{L_0}\|x^{k+1} - x^k\|^2. \tag{105}$$

Convex combination of (104) and (105) with $\alpha \in (0,1)$, which will be specified latter, gives

$$\lambda_k \langle \nabla f(x^k), x^* - x^k \rangle \leqslant \lambda_k (f_* - f(x^k)) - \alpha \frac{\lambda_k \mu}{2}\|x^k - x^*\|^2 - (1 - \alpha)\frac{2\mu}{L_0}\|x^{k+1} - x^k\|^2.$$

Using the above inequality instead of convexity and keeping the rest of the proof of Lemma G.2 as is with omitted $\Sigma_i$ terms, we get an analog of (75):

$$\left\|x^{k+1} - x^*\right\|^2 + \frac{1}{4}\left(1 + (1 - \alpha)\frac{8\mu}{L_0}\right)\left\|x^{k+1} - x^k\right\|^2 + \frac{1 + \theta_k}{1 + 3\theta_k/4} \cdot 2\lambda_{k+1}\theta_{k+1}(f(x^k) - f_*)$$
$$\leqslant \left\|x^{k+1} - x^*\right\|^2 + \frac{1}{4}\left(1 + (1 - \alpha)\frac{8\mu}{L_0}\right)\left\|x^{k+1} - x^k\right\|^2 + 2\lambda_k (1 + \theta_k)(f(x^k) - f_*)$$
$$\leqslant \left(1 - \alpha \frac{\lambda_k \mu}{2}\right)\left\|x^k - x^*\right\|^2 + \frac{1}{4}\left\|x^k - x^{k-1}\right\|^2 + 2\lambda_k \theta_k (f(x^{k-1}) - f_*),$$

where the first inequality follows from $\frac{1 + \theta_k}{1 + 3\theta_k/4}\lambda_{k+1}\theta_{k+1} \leqslant \lambda_k (1 + \theta_k)$ provided by the new condition on $\lambda_k$. Thus, we have contraction in every term: $1 - \alpha\frac{\lambda_k \mu}{2}$ in the first, $\frac{1}{1 + (1-\alpha)\frac{8\mu}{L_0}} = 1 - \frac{(1-\alpha)\frac{8\mu}{L_0}}{1 + (1-\alpha)\frac{8\mu}{L_0}}$ in the second and $\frac{1 + 3\theta_k/4}{1 + \theta_k} = 1 - \frac{\theta_k}{4(1 + \theta_k)}$ in the last one. We bound the third term

as $\frac{\theta_k}{4(1+\theta_k)} = \frac{1}{4} \cdot \frac{\lambda_k}{(\lambda_k + \lambda_{k-1})} \geqslant \frac{\mu \lambda_k}{2}$ using $\lambda_k \leqslant \frac{1}{4\mu}$ for both terms in the denominator. Taking $\alpha = \alpha^* := \frac{73 - \sqrt{3281}}{16} \approx 0.98$, which is the root of $\alpha^* = \frac{64(1-\alpha^*)}{1+8(1-\alpha^*)}$, we bound the second term as $\frac{(1-\alpha)\frac{8\mu}{L_0}}{\left(1+(1-\alpha)\frac{8\mu}{L_0}\right)} \geqslant \frac{\mu}{4L_0} \cdot \frac{32(1-\alpha)}{1+8(1-\alpha)} \stackrel{\alpha=\alpha^*}{=} \alpha^* \frac{\mu}{8L_0}$. Therefore, for $\Psi_k = \left(1 - \frac{\lambda_k \mu}{4}\right)\left\|x^k - x^*\right\|^2 + \frac{1}{4}\left(1 + (1-\alpha^*)\frac{8\mu}{L_0}\right)\left\|x^k - x^{k-1}\right\|^2 + 2\lambda_k \theta_k (f(x^{k-1}) - f_*)$ we have

$$\Psi_{k+1} \leq \left(1 - \frac{\alpha^* \mu}{2}\min\left\{\lambda_k, \frac{1}{4L_0}\right\}\right)\Psi_k. \tag{106}$$

The final step of the proof is unrolling the recursion for Lyapunov function $\Psi_k$. Following the proof of Theorem G.2, we have that

$$\min\left\{\lambda_k, \frac{1}{4L_0}\right\} \geqslant \begin{cases} \frac{1}{4L_0} - \frac{L_1}{4L_0}\|x^t - x^{t-1}\|, & \text{if } k \in \mathcal{T}_1, \text{ where } t \in \mathcal{K} \text{ and } 0 \leqslant k - t \leqslant m, \\ \frac{1}{4L_0} - \frac{L_1}{4L_0}\|x^{k+1} - x^k\|, & \text{if } k \in \mathcal{T}_2, \\ \frac{1}{4L_0}, & \text{if } k \in \mathcal{T}_3. \end{cases} \tag{107}$$

with $m := 1 + \log_{\sqrt{\frac{7}{4}}}\left\lceil\frac{(1+DL_1)}{2}\right\rceil$, which differs from $m$ defined in the convex case due to the new condition on $\lambda_k$. Therefore, by repeating all the steps from the proof of (99), we obtain

$$\sum_{k=1}^{N}\min\left\{\lambda_k, \frac{1}{4L_0}\right\} \geqslant \frac{N}{4L_0} - \frac{\sqrt{2N}(m+1)L_1 D}{4L_0}. \tag{108}$$

Next, we bound the product that arises during recursion unrolling by using the relation between the geometric mean and the arithmetic mean:

$$\prod_{k=1}^{N}\left(1 - \frac{\alpha^*\mu}{2}\min\left\{\lambda_k, \frac{1}{4L_0}\right\}\right) \leqslant \left(1 - \frac{\alpha^*\mu}{2}\frac{1}{N}\sum_{k=1}^{N}\min\left\{\lambda_k, \frac{1}{4L_0}\right\}\right)^N$$

$$\stackrel{(108)}{\leqslant} \left(1 - \frac{\alpha^*\mu}{8L_0} + \frac{\alpha^*\mu(m+1)L_1 D}{4\sqrt{2N}L_0}\right)^N. \tag{109}$$

Finally, we combine (106) and (109) and get

$$\Psi_{N+1} \stackrel{(106)}{\leq} \prod_{k=1}^{N}\left(1 - \frac{\alpha^*\mu}{2}\min\left\{\lambda_k, \frac{1}{4L_0}\right\}\right)\Psi_1$$

$$\stackrel{(109)}{\leq} \left(1 - \frac{\alpha^*\mu}{8L_0} + \frac{\alpha^*\mu(m+1)L_1 D}{4\sqrt{2N}L_0}\right)^N \Psi_1.$$

Moreover, when $N \geqslant 8(m+1)^2 L_1^2 D^2$, we have $\frac{\alpha^*\mu(m+1)L_1 D}{4\sqrt{2N}L_0} \leqslant \frac{\alpha^*\mu}{16L_0}$, which in combination with the above inequality implies (103).

$\square$

# H    STOCHASTIC EXTENSIONS: MISSING PROOFS AND DETAILS

## H.1    $(L_0, L_1)$-STOCHASTIC GRADIENT DESCENT

---

**Algorithm 5** $(L_0, L_1)$-Stochastic Gradient Descent ($(L_0, L_1)$-SGD)

---

**Input:** starting point $x^0$, number of iterations $N$, stepsize parameter $\eta > 0$, $L_0 > 0$, $L_1 \geqslant 0$
 1: **for** $k = 0, 1, \ldots, N - 1$ **do**
 2:    Sample $\xi^k \sim \{1, \ldots, n\}$ uniformly at random
 3:    $x^{k+1} = x^k - \frac{\eta}{L_0 + L_1 \|\nabla f_{\xi^k}(x^k)\|} \nabla f_{\xi^k}(x^k)$
 4: **end for**
**Output:** $x^N$

---

**Theorem H.1** (Theorem 7.1). *Let Assumption 4 hold. Then, the iterates generated by $(L_0, L_1)$-SGD with $0 < \eta \leqslant \frac{\nu}{2}$, $\nu = e^{-\nu}$ after $N$ iterations satisfy*

$$\min_{k=0,\ldots,N} \mathbb{E}\left[ \min\left\{ \frac{\nu L_0}{4nL_1^2}, f(x^k) - f(x^*) \right\} \right] \leqslant \frac{2L_0 \|x^0 - x^*\|^2}{\eta(N+1)}. \tag{110}$$

*Proof.* Similarly to the deterministic case, we start by expanding the squared distance $x^*$, which is a common minimizer for all $\{f_i\}_{i=1}^n$:

$$
\begin{aligned}
\|x^{k+1} - x^*\|^2 &= \left\| x^k - x^* - \frac{\eta}{L_0 + L_1 \|\nabla f_{\xi^k}(x^k)\|} \nabla f_{\xi^k}(x^k) \right\|^2 \\
&= \|x^k - x^*\|^2 - \frac{2\eta}{L_0 + L_1 \|\nabla f_{\xi^k}(x^k)\|} \langle x^k - x^*, \nabla f_{\xi^k}(x^k) \rangle \\
&\quad + \frac{\eta^2 \|\nabla f_{\xi^k}(x^k)\|^2}{(L_0 + L_1 \|\nabla f_{\xi^k}(x^k)\|)^2} \\
&\overset{(2)}{\leqslant} \|x^k - x^*\|^2 - \frac{2\eta}{L_0 + L_1 \|\nabla f_{\xi^k}(x^k)\|} \left( f_{\xi^k}(x^k) - f_{\xi^k}(x^*) \right) \\
&\quad + \frac{\eta^2 \|\nabla f_{\xi^k}(x^k)\|^2}{(L_0 + L_1 \|\nabla f_{\xi^k}(x^k)\|)^2} \\
&\overset{(8)}{\leqslant} \|x^k - x^*\|^2 - 2\eta\left(1 - \frac{\eta}{\nu}\right) \frac{f_{\xi^k}(x^k) - f_{\xi^k}(x^*)}{L_0 + L_1 \|\nabla f_{\xi^k}(x^k)\|} \\
&\overset{\eta \leqslant \frac{\nu}{2}}{\leqslant} \|x^k - x^*\|^2 - \eta \frac{f_{\xi^k}(x^k) - f_{\xi^k}(x^*)}{L_0 + L_1 \|\nabla f_{\xi^k}(x^k)\|}. \tag{111}
\end{aligned}
$$

As before, we consider two possible cases: $\|\nabla f_{\xi^k}(x^k)\| \geqslant \frac{L_0}{L_1}$ or $\|\nabla f_{\xi^k}(x^k)\| < \frac{L_0}{L_1}$.

**Case 1:** $\|\nabla f_{\xi^k}(x^k)\| \geqslant \frac{L_0}{L_1}$. In this case, we have

$$L_0 + L_1 \|\nabla f_{\xi^k}(x^k)\| \leqslant 2L_1 \|\nabla f_{\xi^k}(x^k)\|, \tag{112}$$

$$\frac{\nu \|\nabla f_{\xi^k}(x^k)\|}{4L_1} \overset{(112)}{\leqslant} \frac{\nu \|\nabla f_{\xi^k}(x^k)\|^2}{2(L_0 + L_1 \|\nabla f_{\xi^k}(x^k)\|)} \overset{(8)}{\leqslant} f_{\xi^k}(x^k) - f_{\xi^k}(x^*). \tag{113}$$

Plugging the above inequalities in (111), we continue the derivation as follows:

$$
\begin{aligned}
\|x^{k+1} - x^*\|^2 &\overset{(111),(112)}{\leqslant} \|x^k - x^*\|^2 - \eta \frac{f_{\xi^k}(x^k) - f_{\xi^k}(x^*)}{2L_1 \|\nabla f_{\xi^k}(x^k)\|} \\
&\overset{(113)}{\leqslant} \|x^k - x^*\|^2 - \frac{\nu\eta}{8L_1^2}. \tag{114}
\end{aligned}
$$

**Case 2:** $\|\nabla f_{\xi^k}(x^k)\| < \frac{L_0}{L_1}$. In this case, we have

$$L_0 + L_1 \|\nabla f_{\xi^k}(x^k)\| \leqslant 2L_0, \tag{115}$$

implying that

$$\|x^{k+1} - x^*\|^2 \overset{(45),(115)}{\leqslant} \|x^k - x^*\|^2 - \frac{\eta}{2L_0} \left( f_{\xi^k}(x^k) - f_{\xi^k}(x^*) \right). \tag{116}$$

To combine (114) and (116), we introduce event $E(x^k) := \left\{ \|\nabla f_{\xi^k}(x^k)\| \geqslant \frac{L_0}{L_1} \mid x^k \right\}$ for given $x^k$ and indicator of event $E(x^k)$ as $\mathbb{1}_{E(x^k)}$, i.e., for given $x^k$, we have $\mathbb{1}_{E(x^k)} = 1$ if $\|\nabla f_{\xi^k}(x^k)\| \geqslant \frac{L_0}{L_1}$, and $\mathbb{1}_{E(x^k)} = 0$ if $\|\nabla f_{\xi^k}(x^k)\| < \frac{L_0}{L_1}$. Then, inequalities (114) and (116) can be unified as follows:

$$\|x^{k+1} - x^*\|^2 \leqslant \|x^k - x^*\|^2 - \mathbb{1}_{E(x^k)} \cdot \frac{\nu\eta}{8L_1^2} - (1 - \mathbb{1}_{E(x^k)}) \cdot \frac{\eta}{2L_0} \left( f_{\xi^k}(x^k) - f_{\xi^k}(x^*) \right).$$

Let us denote the expectation conditioned on $x^k$ as $\mathbb{E}_k[\cdot] := \mathbb{E}[\cdot \mid x^k]$. Taking $\mathbb{E}_k[\cdot]$ from the both sides of the above inequality, we derive

$$
\begin{aligned}
\mathbb{E}_k \left[ \|x^{k+1} - x^*\|^2 \right] &\leqslant \|x^k - x^*\|^2 - p_k \cdot \frac{\nu\eta}{8L_1^2} \\
&\quad - \mathbb{E}_k \left[ (1 - \mathbb{1}_{E(x^k)}) \cdot \frac{\eta}{2L_0} \left( f_{\xi^k}(x^k) - f_{\xi^k}(x^*) \right) \right],
\end{aligned}
$$

where $p_k := \mathbb{P} \left\{ \|\nabla f_{\xi^k}(x^k)\| \geqslant \frac{L_0}{L_1} \mid x^k \right\} = \mathbb{P}\{E(x^k)\} = \mathbb{E}_k[\mathbb{1}_{E(x^k)}]$. Note that $p_k$ is a random variable itself. Nevertheless, if $p_k > 0$, it means that for at least one $\xi^k \in \{1, \ldots, n\}$ we have $\|\nabla f_{\xi^k}(x^k)\| \geqslant \frac{L_0}{L_1}$ for given $x^k$. Therefore, either $p_k \geqslant \frac{1}{n}$ or $p_k = 0$. Moreover, when $p_k = 0$, we have $1 - \mathbb{1}_{E(x^k)} := 1$ for given $x^k$. Putting all together, we continue as follows:

$$
\begin{aligned}
\mathbb{E}_k \left[ \|x^{k+1} - x^*\|^2 \right] &\leqslant \|x^k - x^*\|^2 - \mathbb{1}_{\{p_k>0\}} \cdot p_k \cdot \frac{\nu\eta}{8L_1^2} \\
&\quad - \mathbb{1}_{\{p_k=0\}} \cdot \mathbb{E}_k \left[ (1 - \mathbb{1}_{E(x^k)}) \cdot \frac{\eta}{2L_0} \left( f_{\xi^k}(x^k) - f_{\xi^k}(x^*) \right) \right], \\
&= \|x^k - x^*\|^2 - \mathbb{1}_{\{p_k>0\}} \cdot p_k \cdot \frac{\nu\eta}{8L_1^2} - \mathbb{1}_{\{p_k=0\}} \cdot \frac{\eta}{2L_0} \left( f(x^k) - f(x^*) \right) \\
&\leqslant \|x^k - x^*\|^2 - \mathbb{1}_{\{p_k>0\}} \cdot \frac{\nu\eta}{8nL_1^2} - \mathbb{1}_{\{p_k=0\}} \cdot \frac{\eta}{2L_0} \left( f(x^k) - f(x^*) \right) \\
&\leqslant \|x^k - x^*\|^2 - \min \left\{ \frac{\nu\eta}{8nL_1^2}, \frac{\eta}{2L_0} \left( f(x^k) - f(x^*) \right) \right\}.
\end{aligned}
$$

Taking full expectation from the above inequality and telescoping the result, we get

$$
\begin{aligned}
\sum_{k=0}^{N} \mathbb{E} \left[ \min \left\{ \frac{\nu\eta}{8nL_1^2}, \frac{\eta}{2L_0} \left( f(x^k) - f(x^*) \right) \right\} \right] &\leqslant \sum_{k=0}^{N+1} (\mathbb{E}[\|x^{k+1} - x^*\|^2] - \mathbb{E}[\|x^k - x^*\|^2]) \\
&\leqslant \|x^0 - x^*\|^2.
\end{aligned}
$$

Since $\frac{\eta(N+1)}{2L_0} \min\limits_{k=0,\ldots,N} \mathbb{E} \left[ \min \left\{ \frac{\nu L_0}{4nL_1^2}, f(x^k) - f(x^*) \right\} \right]$ is not greater than $\sum\limits_{k=0}^{N} \mathbb{E} \left[ \min \left\{ \frac{\nu\eta}{8nL_1^2}, \frac{\eta}{2L_0} \left( f(x^k) - f(x^*) \right) \right\} \right]$, we also have

$$\frac{\eta(N+1)}{2L_0} \min_{k=0,\ldots,N} \mathbb{E} \left[ \min \left\{ \frac{\nu L_0}{4nL_1^2}, f(x^k) - f(x^*) \right\} \right] \leqslant \|x^0 - x^*\|^2.$$

Dividing both sides by $\frac{\eta(N+1)}{2L_0}$, we obtain (110). $\qquad \square$

## H.2 STOCHASTIC GRADIENT DESCENT WITH POLYAK STEPSIZES

---

**Algorithm 6** Stochastic Gradient Descent with Polyak Stepsizes (SGD-PS)

---

**Input:** starting point $x^0$, number of iterations $N$, minimal values $f_i(x^*) := \min_{x \in \mathbb{R}^d} f_i(x)$ for all $i \in \{1, \dots, n\}$

1: **for** $k = 0, 1, \dots, N - 1$ **do**
2:      Sample $\xi^k \sim \{1, \dots, n\}$ uniformly at random
3:      $x^{k+1} = x^k - \frac{f_{\xi^k}(x^k) - f_{\xi^k}(x^*)}{\|\nabla f_{\xi^k}(x^k)\|^2} \nabla f_{\xi^k}(x^k)$
4: **end for**
**Output:** $x^N$

---

**Theorem H.2** (Theorem 7.2)*. Let Assumption 4 hold. Then, the iterates generated by* SGD-PS *after $N$ iterations satisfy*

$$\min_{k=0,\dots,N} \mathbb{E}\left[\min\left\{\frac{\nu L_0}{4nL_1^2}, f(x^k) - f(x^*)\right\}\right] \leqslant \frac{4L_0\|x^0 - x^*\|^2}{\nu(N+1)}. \tag{117}$$

*Proof.* Similarly to the deterministic case, we start by expanding the squared distance $x^*$, which is a common minimizer for all $\{f_i\}_{i=1}^n$:

$$
\begin{aligned}
\|x^{k+1} - x^*\|^2 &= \left\|x^k - x^* - \frac{f_{\xi^k}(x^k) - f_{\xi^k}(x^*)}{\|\nabla f_{\xi^k}(x^k)\|^2} \nabla f_{\xi^k}(x^k)\right\|^2 \\
&= \|x^k - x^*\|^2 - \frac{2\left(f_{\xi^k}(x^k) - f_{\xi^k}(x^*)\right)}{\|\nabla f_{\xi^k}(x^k)\|^2}\langle x^k - x^*, \nabla f_{\xi^k}(x^k)\rangle \\
&\quad + \frac{\left(f_{\xi^k}(x^k) - f_{\xi^k}(x^*)\right)^2}{\|\nabla f_{\xi^k}(x^k)\|^2} \\
&\overset{(2)}{\leqslant} \|x^k - x^*\|^2 - \frac{\left(f_{\xi^k}(x^k) - f_{\xi^k}(x^*)\right)^2}{\|\nabla f_{\xi^k}(x^k)\|^2} \\
&\overset{(8)}{\leqslant} \|x^k - x^*\|^2 - \frac{\nu}{2} \cdot \frac{f_{\xi^k}(x^k) - f_{\xi^k}(x^*)}{L_0 + L_1\|\nabla f_{\xi^k}(x^k)\|}.
\end{aligned} \tag{118}
$$

As before, we consider two possible cases: $\|\nabla f_{\xi^k}(x^k)\| \geqslant \frac{L_0}{L_1}$ or $\|\nabla f_{\xi^k}(x^k)\| < \frac{L_0}{L_1}$.

**Case 1:** $\|\nabla f_{\xi^k}(x^k)\| \geqslant \frac{L_0}{L_1}$. In this case, inequalities (112) and (113) hold and the derivation from (118) can be continued as follows:

$$
\begin{aligned}
\|x^{k+1} - x^*\|^2 &\overset{(118),(112)}{\leqslant} \|x^k - x^*\|^2 - \frac{\nu}{2} \cdot \frac{f_{\xi^k}(x^k) - f_{\xi^k}(x^*)}{2L_1\|\nabla f_{\xi^k}(x^k)\|} \\
&\overset{(113)}{\leqslant} \|x^k - x^*\|^2 - \frac{\nu^2}{16L_1^2},
\end{aligned} \tag{119}
$$

which gives (54).

**Case 2:** $\|\nabla f_{\xi^k}(x^k)\| < \frac{L_0}{L_1}$. In this case, inequality (115) holds and we have

$$\|x^{k+1} - x^*\|^2 \overset{(118),(115)}{\leqslant} \|x^k - x^*\|^2 - \frac{\nu}{2} \cdot \frac{f_{\xi^k}(x^k) - f_{\xi^k}(x^*)}{2L_0}. \tag{120}$$

To combine (119) and (120), we introduce event $E(x^k) := \left\{\|\nabla f_{\xi^k}(x^k)\| \geqslant \frac{L_0}{L_1} \mid x^k\right\}$ for given $x^k$ and indicator of event $E(x^k)$ as $\mathbb{1}_{E(x^k)}$, i.e., for given $x^k$, we have $\mathbb{1}_{E(x^k)} = 1$ if $\|\nabla f_{\xi^k}(x^k)\| \geqslant \frac{L_0}{L_1}$, and $\mathbb{1}_{E(x^k)} = 0$ if $\|\nabla f_{\xi^k}(x^k)\| < \frac{L_0}{L_1}$. Then, inequalities (119) and (120) can be unified as follows:

$$\|x^{k+1} - x^*\|^2 \leqslant \|x^k - x^*\|^2 - \mathbb{1}_{E(x^k)} \cdot \frac{\nu^2}{16L_1^2} - (1 - \mathbb{1}_{E(x^k)}) \cdot \frac{\nu}{4L_0}\left(f_{\xi^k}(x^k) - f_{\xi^k}(x^*)\right).$$

Let us denote the expectation conditioned on $x^k$ as $\mathbb{E}_k[\cdot] := \mathbb{E}[\cdot \mid x^k]$. Taking $\mathbb{E}_k[\cdot]$ from the both sides of the above inequality, we derive

$$
\begin{aligned}
\mathbb{E}_k\left[\|x^{k+1} - x^*\|^2\right] \quad \leqslant \quad & \|x^k - x^*\|^2 - p_k \cdot \frac{\nu^2}{16L_1^2} \\
& -\mathbb{E}_k\left[\left(1 - \mathbb{1}_{E(x^k)}\right) \cdot \frac{\nu}{4L_0}\left(f_{\xi^k}(x^k) - f_{\xi^k}(x^*)\right)\right],
\end{aligned}
$$

where $p_k := \mathbb{P}\left\{\|\nabla f_{\xi^k}(x^k)\| \geqslant \frac{L_0}{L_1} \mid x^k\right\} = \mathbb{P}\{E(x^k)\} = \mathbb{E}_k[\mathbb{1}_{E(x^k)}]$. Note that $p_k$ is a random variable itself. Nevertheless, if $p_k > 0$, it means that for at least one $\xi^k \in \{1, \ldots, n\}$ we have $\|\nabla f_{\xi^k}(x^k)\| \geqslant \frac{L_0}{L_1}$ for given $x^k$. Therefore, either $p_k \geqslant \frac{1}{n}$ or $p_k = 0$. Moreover, when $p_k = 0$, we have $1 - \mathbb{1}_{E(x^k)} := 1$ for given $x^k$. Putting all together, we continue as follows:

$$
\begin{aligned}
\mathbb{E}_k\left[\|x^{k+1} - x^*\|^2\right] \quad \leqslant \quad & \|x^k - x^*\|^2 - \mathbb{1}_{\{p_k>0\}} \cdot p_k \cdot \frac{\nu^2}{16L_1^2} \\
& - \mathbb{1}_{\{p_k=0\}} \cdot \mathbb{E}_k\left[\left(1 - \mathbb{1}_{E(x^k)}\right) \cdot \frac{\nu}{4L_0}\left(f_{\xi^k}(x^k) - f_{\xi^k}(x^*)\right)\right], \\
= \quad & \|x^k - x^*\|^2 - \mathbb{1}_{\{p_k>0\}} \cdot p_k \cdot \frac{\nu^2}{16L_1^2} - \mathbb{1}_{\{p_k=0\}} \cdot \frac{\nu}{4L_0}\left(f(x^k) - f(x^*)\right) \\
\leqslant \quad & \|x^k - x^*\|^2 - \mathbb{1}_{\{p_k>0\}} \cdot \frac{\nu^2}{16nL_1^2} - \mathbb{1}_{\{p_k=0\}} \cdot \frac{\nu}{4L_0}\left(f(x^k) - f(x^*)\right) \\
\leqslant \quad & \|x^k - x^*\|^2 - \min\left\{\frac{\nu^2}{16nL_1^2}, \frac{\nu}{4L_0}\left(f(x^k) - f(x^*)\right)\right\}.
\end{aligned}
$$

Taking full expectation from the above inequality and telescoping the result, we get

$$
\begin{aligned}
\sum_{k=0}^N \mathbb{E}\left[\min\left\{\frac{\nu^2}{16nL_1^2}, \frac{\nu}{4L_0}\left(f(x^k) - f(x^*)\right)\right\}\right] \quad \leqslant \quad & \sum_{k=0}^{N+1}\left(\mathbb{E}[\|x^{k+1} - x^*\|^2] - \mathbb{E}[\|x^k - x^*\|^2]\right) \\
\leqslant \quad & \|x^0 - x^*\|^2.
\end{aligned}
$$

Since $\quad \frac{\nu(N+1)}{4L_0} \min\limits_{k=0,\ldots,N} \mathbb{E}\left[\min\left\{\frac{\nu L_0}{4nL_1^2}, f(x^k) - f(x^*)\right\}\right] \quad$ is not greater than $\sum\limits_{k=0}^N \mathbb{E}\left[\min\left\{\frac{\nu^2}{16nL_1^2}, \frac{\nu}{4L_0}\left(f(x^k) - f(x^*)\right)\right\}\right]$, we also have

$$
\frac{\nu(N+1)}{4L_0} \min_{k=0,\ldots,N} \mathbb{E}\left[\min\left\{\frac{\nu L_0}{4nL_1^2}, f(x^k) - f(x^*)\right\}\right] \quad \leqslant \quad \|x^0 - x^*\|^2.
$$

Dividing both sides by $\frac{\nu(N+1)}{4L_0}$, we obtain (117). $\qquad\square$

**Remark H.1.** *Note that the LHS of equation* (29) *can be lower bounded as follows:*

$$
\begin{aligned}
\frac{\nu(N+1)}{4L_0} \min_{k=0,\ldots,N} \mathbb{E}\Bigg[ & \min\left\{\frac{\nu L_0}{4nL_1^2}, f(x^k) - f(x^*)\right\} \mathbf{1}_{f(x^k)-f(x^*)\geqslant \frac{\nu L_0}{4nL_1^2}} \\
& + \min\left\{\frac{\nu L_0}{4nL_1^2}, f(x^k) - f(x^*)\right\} \mathbf{1}_{f(x^k)-f(x^*)\leqslant \frac{\nu L_0}{4nL_1^2}}\Bigg] \\
\geqslant \quad & \min_{k=0,\ldots,N} \mathbb{E}\left[\min\left\{\frac{\nu L_0}{4nL_1^2}, f(x^k) - f(x^*)\right\} \mathbf{1}_{f(x^k)-f(x^*)\geqslant \frac{\nu L_0}{4nL_1^2}}\right] \\
= \quad & \min_{k=0,\cdots,N} \frac{\eta L_0}{4nL_1^2} \mathbf{P}\left(f(x^k) - f(x^*) \geqslant \frac{\nu L_0}{4nL^2}\right)
\end{aligned}
$$

*Therefore, from equation equation (29) we have that* $\min_{k=0,\cdots,N} \mathbf{P}\left(f(x^k) - f(x^*) \geqslant \frac{\nu L_0}{4nL^2}\right) \leqslant \frac{8nL_1^2\|x^k-x^*\|^2}{\eta\nu(L+1)}$.

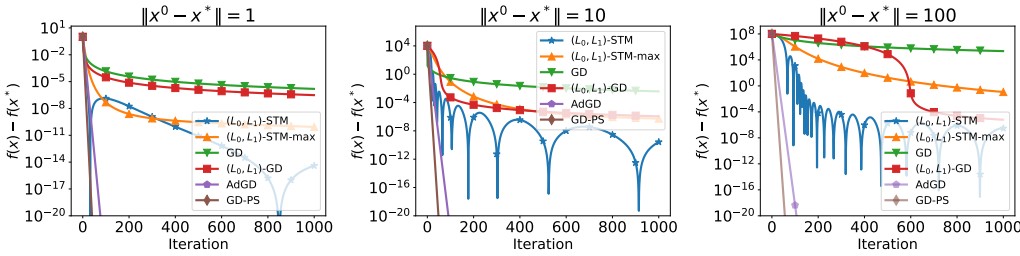

Figure 1: The last iterate discrepancy of algorithms on the one-variable polynomial function $f(x) = x^4$.

## I  NUMERICAL EXPERIMENTS

**Synthetic experiment.** The existing numerical studies already illustrate the benefits of many methods considered in this paper in solving $(L_0, L_1)$-smooth problems. In particular, the results of numerical experiments with Clip-GD, which is closely related to $(L_0, L_1)$-GD, GD-PS, and AdGD on training LSTM (Merity et al., 2018) and/or ResNet (He et al., 2016) models are provided in (Zhang et al., 2020b; Loizou et al., 2021; Malitsky & Mishchenko, 2020). Therefore, in our numerical experiments, we focus on a simple 1-dimensional problem that is convex, $(L_0, L_1)$-smooth, and provides additional insights to the ones presented in the literature. In particular, we consider function $f(x) = x^4$, which is convex, $(4, 3)$-smooth, but not $L$-smooth as illustrated in Example 1.1. We run (i) GD with stepsize $1/L$, $L = 12|x^0|^2$ (which corresponds to the worst-case smoothness constant on the interval $|x| \leqslant |x^0|$), (ii) $(L_0, L_1)$-GD with $L_0 = 4$, $L_1 = 3$, $\eta = \nu/2$, (iii) $(L_0, L_1)$-STM with $G_{k+1} = L_0 + L_1\|\nabla f(x^{k+1})\|$ (not supported by our theory) and (iv) with $G_{k+1} = \max\{G_k, L_0 + L_1\|\nabla f(x^{k+1})\|\}$ (called $(L_0, L_1)$-STM-max on the plots), (v) GD-PS, and (vi) AdGD for starting points $x^0 \in \{1, 10, 100\}$. The results are reported in Figure 1. In all tests, GD-PS and AdGD show the best results among other methods (which is expected since these methods are the only parameter-free methods). Next, standard GD is the slowest among other methods and slow-downs once we move the starting point further from the optimum, which is also expected since $L$ increases and we have to use smaller stepsizes for GD. Finally, let us discuss the behavior of $(L_0, L_1)$-GD, $(L_0, L_1)$-STM-max, and $(L_0, L_1)$-STM. Clearly, it depends on the distance from the starting point to the solution. In particular, when $x^0 = 1$ we have $\|\nabla f(x^0)\| = 4$, meaning that $L = 16$. In this case, GD and $(L_0, L_1)$-GD behave similarly to each other, and $(L_0, L_1)$-STM-max significantly outperforms both of them, which is well-aligned with the derived bounds. However, for $x^0 = 10$ and $x^0 = 100$ we have $\|\nabla f(x^0)\| = 4 \cdot 10^3$ and $\|\nabla f(x^0)\| = 4 \cdot 10^6$ leading to a significant slow down in the convergence of GD and $(L_0, L_1)$-STM-max. In particular, $(L_0, L_1)$-GD achieves a similar optimization error to $(L_0, L_1)$-STM-max for $x^0 = 10$ and much better optimization error for $x^0 = 100$. This is also aligned with our theoretical results: when $R_0$ is large and number of iterations is not too large, bound (15) derived for $(L_0, L_1)$-GD can be better than bound (23) derived for $(L_0, L_1)$-STM-max. Moreover, for $x^0 = 100$, Figure 1 illustrates well the two-stages convergence behavior of $(L_0, L_1)$-GD described in Theorem 3.1. Finally, although our theory does not provide any guarantees for $(L_0, L_1)$-STM with $G_{k+1} = L_0 + L_1\|\nabla f(x^{k+1})\|$, this method converges faster than $(L_0, L_1)$-GD for the considered problem but exhibits highly non-monotone behavior.

**Logistic regression.** We also study the behavior of the algorithms on the Logistic Regression problem of the form

$$f(x) = \frac{1}{n} \sum_{i=1}^{n} f_i(x), \quad \text{where} \quad f_i(x) = \log\left(1 + \exp(-y_i a_i^\top x)\right), \ a_i \in \mathbb{R}^d, \ y_i \in \{-1, 1\}.$$

As Example 1.3 shows, each individual function $f_i$ is $(L_0, L_1)$-smooth. Moreover, function $f$ is also $L$-smooth. This implies that $f(x)$ is $(L_0, L_1)$-smooth, but the derivation of the exact constants $L_0$ and $L_1$ for $f$ is problematic and highly depends on the relation between $\{a_i\}_{i=1}^n$. Nevertheless, one can still compare the methods considered above on this problem and numerically estimate the dependency of the Hessian norm on the norm of the gradient. In particular, we observe linear-like gradient norm dependency on the hessian norm in the toy scenario, where all vectors $a_i$ are close to each other, i.e., we generated $a_i \in \mathbb{R}^{50}$ as $a_i = (1, 2, \ldots, 50)^\top + \xi_i^\top$, where $\xi_i \sim \mathcal{N}(0, \mathbf{I})$ are

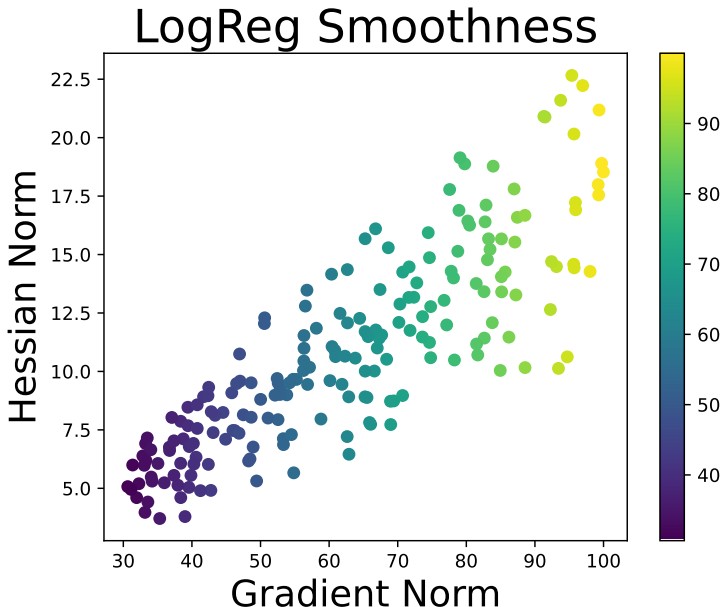

Figure 2: Smoothness dependency on the gradient norm, toy scenario logistic regression.

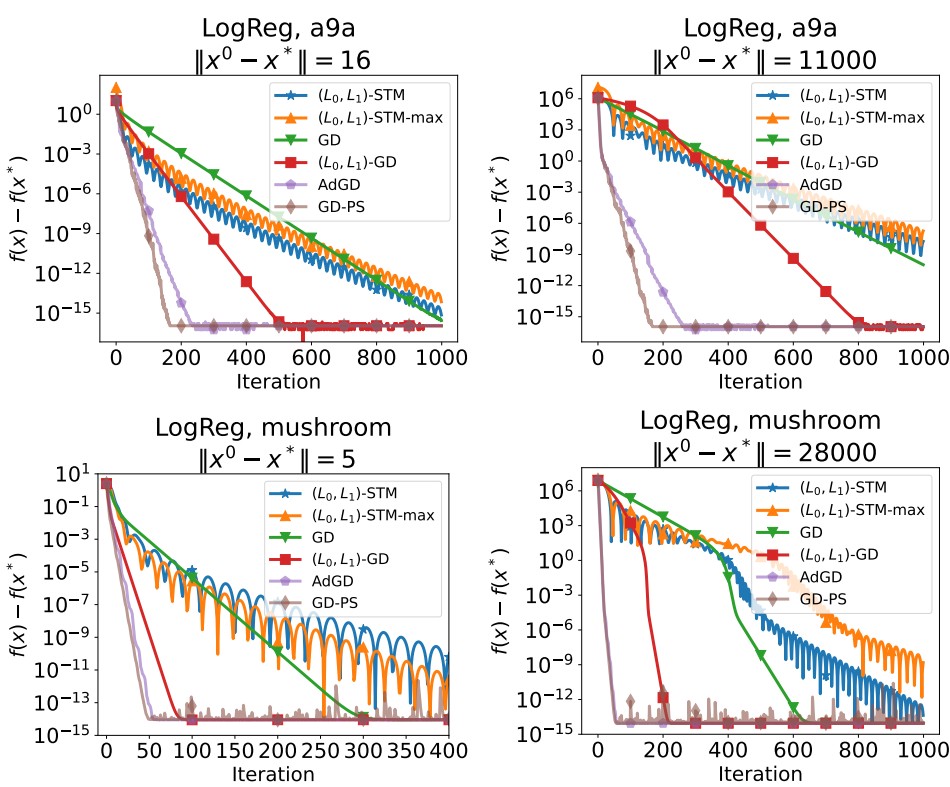

Figure 3: The last iterate discrepancy of algorithms on the logistic regression problem.

i.i.d. standard Gaussian vectors, and all $y_i = 1$ except of one $y_j = -1$ for randomly selected $j$ from $\{1, \ldots, 50\}$ (Figure 1).

We also run the considered methods for real datasets from LIBSVM (Chang & Lin, 2011) – a9a and mushrooms – for different starting points. The results are presented in Figure 3. Despite the fact that for these datasets, $f$ does not have a clear linear dependence of the norm of the Hessian w.r.t. the norm of the gradient, the methods that are better suited for $(L_0, L_1)$-smooth problems (like $(L_0, L_1)$-GD, GD-PS, and AdGD) converge significantly faster than other methods. Moreover, we also emphasize that accelerated variants – $(L_0, L_1)$-STM and $(L_0, L_1)$-STM-max – work not better than standard GD in this case.

