# OpenReview forum: "Methods for Convex $(L_0,L_1)$-Smooth Optimization: Clipping, Acceleration, and Adaptivity"
_ICLR.cc/2025/Conference — ICLR 2025 Poster_

### Official Review · Reviewer_4rMB · 2024-10-27

**Soundness:** 2
**Presentation:** 3
**Contribution:** 3
**Rating:** 6
**Confidence:** 3

**Summary:**

The paper present a study for gradient method for solving optimization problems involving (L0,L1)-smooth objective function, which was first introduced in (Zhang et al, 2020b). The analysis in the current paper is devoted fully for the convex and strongly convex case, where the convergence analysis of several methods is proopsed, both in the deterministic and stochastic setting. However, there are some major issues needed to be addressed.

**Strengths:**

The paper is well-writen, and thus easy to follow. The paper has some signification theoretical contributions for the existing algorithms. There are also new algorithms being adapated from the classical smoothness to the new (L0,L1)-smoothness.

**Weaknesses:**

1. The analysis of the current paper is mostly based on Assymetric and Symmetric (L0,L1) smoothness, which can be hold for functions that are not twice differentiable. This is good since this class cover the original class of function in (Zhang et al, 2020b), where twice differentiability is a must. However, which meaningful classes of functions satisfy Assymetric/Symmetric (L0,L1) smoothness while not C2? I found that all the examples presented are C2, and thus it seems that the use of  Assymetric and Symmetric (L0,L1) smoothness is not neccesary, which reduces the importance of the current paper much. Note that examples for L-smooth functions that are not C2 are diverse, so it is reasonable to use the gradient Lipschitz condition instead of the stronger one, bounded Hessian. I do not think it is the case for (L0,L1) smoothness.

2. Now assume that the function is C2. Lemma 2.1 shows that (L0,L1) smoothness implies (not equivalent) equation (7). In comparison with Lemma 2.2 (Vankov et al., 2024) below, their equation (2.2) seems to be a shaper inequality and in fact is a equivalent condition. Based on this, I suspect that the results obtained by the paper under review is not as tight as claimed by the authors, especially when compared with (Vankov et al., 2024).

3. The convergence theory of algorithms for (L0,L1)-smooth function does not explain why they are better than standard gradient descent. For example, can the authors explain why standard GD performs worse than designated algorithms for (L0,L1)-smooth function in solving logistic regression?

Reference.
D Vankov, A Rodomanov, A Nedich, L Sankar, SU Stich, Optimizing (L0,L1) - Smooth Functions by Gradient Methods, https://arxiv.org/abs/2410.10800

**Questions:**

See weakness

---

> ### Author Response · Authors · 2024-11-20
> **Respose to Reviewer 4rMB**
>
> >**The analysis of the current paper is mostly based on Assymetric and Symmetric (L0,L1) smoothness, which can be hold for functions that are not twice differentiable. This is good since this class cover the original class of function in (Zhang et al, 2020b), where twice differentiability is a must. However, which meaningful classes of functions satisfy Assymetric/Symmetric (L0,L1) smoothness while not C2? I found that all the examples presented are C2, and thus it seems that the use of Assymetric and Symmetric (L0,L1) smoothness is not neccesary, which reduces the importance of the current paper much. Note that examples for L-smooth functions that are not C2 are diverse, so it is reasonable to use the gradient Lipschitz condition instead of the stronger one, bounded Hessian. I do not think it is the case for (L0,L1) smoothness.**
>
>
> Indeed, we consider a more general assumption, including Hessian free given by (Zhang et al. 2020b) [Corollary A.4] which is equivalent to the standard assumption with Hessian assumption (inequality (5) of our work or C2).
> The lack of examples does not necessarily reduce the importance. [1] provides various plots which indicate complicated dependence of Hessian norm on Gradient norms. Having a more complicated model may lead to a more general assumption on smoothness. So we believe that our results also create value for a community, by analyzing more general assumption, adaptive methods, and introducing stochasticity.
>
> [1] Zhang, Jingzhao, et al. "Why gradient clipping accelerates training: A theoretical justification for adaptivity." arXiv preprint arXiv:1905.11881 (2019).
>
> >**Now assume that the function is C2. Lemma 2.1 shows that (L0,L1) smoothness implies (not equivalent) equation (7). In comparison with Lemma 2.2 (Vankov et al., 2024) below, their equation (2.2) seems to be a shaper inequality and in fact is an equivalent condition. Based on this, I suspect that the results obtained by the paper under review is not as tight as claimed by the authors, especially when compared with (Vankov et al., 2024).**
>
> First of all it is worth mentioning that (Vankov et al., 2024) appeared on arXiv after the submission deadline, so it is not appropriate to compare.
>
> But as we stated above, we consider a more general assumption, which includes the assumption made in (Vankov et al., 2024). Moreover, to the best of our knowledge, the lower bounds are not obtained yet for the set of considered assumptions. So, it is challenging for us to say that our results or the results of (Vankov et al., 2024) are optimal for the corresponding set of assumptions. Moreover, for sufficiently large number of iterations even constant factors are almost identical. For  $(L_0,L_1)$-GD we obtain $2/\nu$ constant which is almost the same as 4 for GD of (Vankov et al., 2024), since $\nu = e^{-\nu}$ and $0.56 < \nu < 0.57$, so we argue, that the assumption we made is not too restrictive.
>
> >**The convergence theory of algorithms for $(L0,L1)$-smooth function does not explain why they are better than standard gradient descent. For example, can the authors explain why standard GD performs worse than designated algorithms for $(L0,L1)$-smooth function in solving logistic regression?**
>
> Appendix B considers some examples of functions satisfying the Hessian-based assumption (5), which is covered by the symmetric assumption we used in our paper. $(L_0,L_1)$-smoothness can be reduced to the case of the standard smoothness by bounding the gradient norm by (24), which leads to a pessimistic constant. The standard gradient descent performs constant step-size steps, and the first one is inversely proportional to that pessimistic constant, so it is very small.
> $(L_0,L_1)$-GD allows the stepsize to increase as the gradient norm vanishes. The method performs large stepsizes, thus achieving faster convergence.
>
> Regarding logistic regression, we consider the logistic loss function in example B.3, which is expected to share properties with the average of logistic functions. Although we are not aware of an explicit formula for $L_0$ and $L_1$ for the average of logistic functions, one can note that in practice, the Hessian norm can indeed depend on the gradient norm, as shown in Figure 2. Thus, GD performs worse than other methods since GD uses smaller step sizes corresponding to the worst-case smoothness constant.

---

> > ### Comment · Reviewer_4rMB · 2024-11-21
> >
> > Thanks for your clarifications. I will adjust my score, leaning more towards acceptance.

---

### Official Review · Reviewer_iTbo · 2024-10-31

**Soundness:** 3
**Presentation:** 4
**Contribution:** 3
**Rating:** 6
**Confidence:** 5

**Summary:**

This paper focuses on analyzing optimization methods for convex problems under $(L_0,L_1)$-smoothness settings. It provides improved convergence rates for the Gradient Descent (GD) method with Gradient Clipping and the Gradient Descent method with Polyak Stepsizes. It introduces a new accelerated method based on the Similar Triangles Method and provides new convergence rates for the Adaptive Gradient Descent Method. Finally, it extends the analysis to the stochastic case in overparametrized settings.

**Strengths:**

1. The analysis in this work does not rely on the L-smooth assumption, which was required in previous works. In this way, the work proves a convergence rate for the Gradient Descent (GD) method with Gradient Clipping and the Gradient Descent method with Polyak Stepsizes, with the dominant part having a smaller constant depending on $L_0$.

2. This work proposes a new variant of the Similar Triangles Method that accelerates GD.

3. This work provides a faster convergence rate for the Adaptive Gradient Descent Method in $(L_0,L_1)$-smoothness settings compared to the locally L-smooth setting.

**Weaknesses:**

1. A new acceleration of GD is proposed. It would be supportive to add more remarks to highlight the theoretical merits of this acceleration compared with the STM in (Gasnikov & Nesterov, 2016). Additionally, it would be more supportive to numerically compare its performance with STM in (Gasnikov & Nesterov, 2016).

2. Example 1.3 considers a logistic function with L2 regularization. However, $f(x)$ is not related to L2. It would be better to specify where the L2 regularization is.

3. The discussion after Theorem 7.1 (line 521) claims that the probability must be smaller than $\frac{8nL_1^2\|x^0-x^*\|^2}{\eta\nu(N+1)}$. It would be clearer to explain why the probability should be smaller than this value.

**Questions:**

Please refer to the weakness.

---

> ### Author Response · Authors · 2024-11-20
> **Respose to Reviewer iTbo**
>
> >**A new acceleration of GD is proposed. It would be supportive to add more remarks to highlight the theoretical merits of this acceleration compared with the STM in (Gasnikov & Nesterov, 2016). Additionally, it would be more supportive to numerically compare its performance with STM in (Gasnikov & Nesterov, 2016).**
>
> We have a detailed comparison of our accelerated GD method with the STM from [Gasnikov & Nesterov, 2016]. As highlighted in the line 362-364, our algorithm differs from the original STM in line 6 of Algorithm 3. Moreover, as discussed in lines 408-409, we recover the convergence guarantees of the original STM from Theorem 5.1 with $L_1 = 0$.
>
> However,  we agree that it would be valuable to include experiments to evaluate the empirical performance of the proposed method. We will add these experiments in the camera-ready version.
>
>
> >**Example 1.3 considers a logistic function with L2 regularization. However, f(x) is not related to L2. It would be better to specify where the L2 regularization is.**
>
> Thank you for pointing this out. This is indeed a typo. The problem in example 1.3 is a simple logistic function (without L2 regularization). We fixed this typo in the revised version of our manuscript.
>
> >**The discussion after Theorem 7.1 (line 521) claims that the probability must be smaller than ……. It would be clearer to explain why the probability should be smaller than this value.**
>
> Thank you for your question. We agree that due to space constraints, we were unable to include all the necessary details. For clarity, we added Remark H.1 in line 2146.
>
> We welcome the reviewer to check the revised version of our manuscript for the clarifications.

---

> > ### Comment · Area_Chair_MLFq · 2024-11-24
> >
> > Dear Reviewer iTbo,
> >
> > The author discussion phase will be ending soon. The authors have provided detailed responses. Could you please reply to the authors whether they have addressed your concern and whether you will keep or modify your assessment of this submission?
> >
> > Thanks.
> >
> > Area Chair

---

> > ### Comment · Reviewer_iTbo · 2024-11-25
> >
> > Thanks for your clarifications. My concerns have been addressed except the experimental part. I will keep my initial rating.

---

### Official Review · Reviewer_jtK8 · 2024-11-03

**Soundness:** 3
**Presentation:** 2
**Contribution:** 2
**Rating:** 6
**Confidence:** 4

**Summary:**

This paper takes a closer look at $(L_0,L_1)$-smoothness for the setting of convex optimization. There, it derives more fine-grained convergence rate guarantees than existing work, while discussing extensions to accelerated, stochastic, and certain adaptive settings.

**Strengths:**

**UPDATE:** Given the authors willingness (mentioned in their final comment) to make the paper's claims more transparent and to perform the pedagogic and scholarly improvements, I have raised my score.

---
The paper makes a technical contribution to the convergence analysis of gradient descent(like) methods, under the general $(L_0,L_1)$ smoothness for differentiable functions. In the reviewer's opinion, the main contribution is a tighter bound on Clip-GD under the generalized smoothness assumption (e.g., The Polyak stepsize is chosen to minimize the upper bound in (59), so the result for GD-PS is not hard to get after having proved (45) for Clip-GD, while STM and AdGD have exponential terms in their upper bounds, and hence only of marginal interest).

Strengths: This article fills a gap in the analysis of convex L0-L1 problems that are differentiable, and claims to do that without having to resort to small stepsizes as in the recent work of Li et al. The work is reasonably clearly written, and is easy to follow. Some aspects that could do with more discussion are the two-phase nature of GD (with the $>, < L_0/L_1$), and what that may mean when compared with other work (does the same happen in the nonconvex case for instance? how does it relate to bounds from the work of Li et al for instance?)

**Weaknesses:**

*  From a quick skim of the literature, it seems that Zhang et al noted in passing that twice differentiability could be dropped at the expense of some more analysis, but the full details of the differentiable case were worked out in subsequent work. This aspect is not clear from how the current work cites related work, and should be fixed.

* While the related work section is overall fairly good, a more precise statement about the results of Chen et al is needed, especially because that work introduced some of the key technical tools too, and studied the nonconvex case; in particular, it would be worth noting what happens if one trivially takes their nonconvex results, and tries to adapt them to the convex case (by boosting stationarity to function suboptimality using the current assumptions). Also, their slightly more general $\alpha$-version of Assumption 3 could be noted.

* Section 5 on acceleration could be deferred to the appendix or noted in passing, and more discussion given to the exponential terms in the bound, which are tantamount to saying that essentially no practical acceleration happens, even though the technical result itself is interesting to note. Similarly, the bounds arising in Section 6 should be discussed a bit more, because due to the central assumption of the paper, ultimately the pessimistic exponential terms in D arise.

* The authors's motivation for studying AdGD should be as a result further enhanced: it seems inherently unsuitable to use under the generalized smoothness assumptions since AdGD does not utilize clipping.  And the result for the stochastic case (Section 7) requires a common minimizer for all the stochastic components (Assumption 4). Although such an assumption is also used in some works, this assumption is not that weak, and renders the results less applicable.

**Questions:**

* Can a version of the method that is somewhat more agnostic of the knowledge of L0 and L1 be designed, since right now this knowledge is quite necessary for selecting the correct step sizes?

* Ultimately, a high-level takeaway of the work (which is likely also a takeaway that follows from the Chen paper if one could quickly bridge their stationarity to function suboptimality here) seems to be that the results are ultimately of a *local nature*. What I mean by that is that due to the unavoidable dependence of the type $\exp(\|x-y\|)$, the terms involving that in the bound will be large, unless $\|x-y\|$ is sufficiently small, and hence, in spirit the results can be viewed as showing that the GD methods studied in the paper are good but only locally. This aspect is not a criticism of the paper, but just a comment on how one can typically interpret bounds that involve exponentials.

* Do the results really offer a significant improvement over Li et al. 2024a (_Convex and Non-convex Optimization Under Generalized Smoothness_). The results provided by Li et al. 2024a for GD and NAG also do not rely on the assumption of L-smoothness, and their acceleration results for NAG  **do not** have a dependence on exp(L1). The authors claim that the advantage of their results is that they do not require dependencies on $\Vert \nabla f(x_0) \Vert$ and $f(x_0) - f^*$, as in Li et al. 2024a, and they argue that these quantities could potentially be exponentially large in the worst case (Line 408-420)---But in the reviewer's opinion, assuming these initial quantities to be constants is not a very strong assumption, whereas in comparison, the authors' acceleration result depends on exp(L1), which seems to be less favorable.

---

> ### Author Response · Authors · 2024-11-20
> **Respose to Reviewer jtK8: Part 1**
>
> >**From a quick skim of the literature, it seems that Zhang et al noted in passing that twice differentiability could be dropped at the expense of some more analysis, but the full details of the differentiable case were worked out in subsequent work. This aspect is not clear from how the current work cites related work, and should be fixed.**
>
> Indeed, (Zhang et al, 2020b)  provides an equivalent assumption for $(L_0, L_1)$ smoothness, not involving Hessian (avoiding twice differentiability). This assumption is very similar to the one we use in our work which originates from (Chen et al., 2023. We explicitly refer this in Lemma 2.1, see line 188. We also added a remark in line 197.  See also discussion in lines 826-837 about other notions of generalized smoothness.
> Our assumption covers the Hessian one (so the assumption we use is more general), but,to the best of our knowledge, the assumption equivalent to the Hessian one can improve our results only up to a constant factor, preserving exponentials at their form, see line 197.
>
> We kindly ask the reviewers to let us know whether our revised version addresses their concern.
>
> >**While the related work section is overall fairly good, a more precise statement about the results of Chen et al is needed, especially because that work introduced some of the key technical tools too, and studied the nonconvex case; in particular, it would be worth noting what happens if one trivially takes their nonconvex results, and tries to adapt them to the convex case (by boosting stationarity to function suboptimality using the current assumptions). Also, their slightly more general $\alpha$-version of Assumption 3 could be noted.**
>
> We respectfully and explicitly refer to (Chen et al., 2023) contribution in Lemma 2.1 (see line  185-187), related work section and extra related work in the appendix. Indeed, our results utilize their assumption. We also discuss the results of the $\alpha$-version of the assumption in Appendix A.
>
> We are not aware of any technique allowing trivially adapts the results of the nonconvex case to the convex case, but we believe that a trivial adaptation does not lead to tight results.
>
> We welcome the reviewer to check the revised version of our manuscript for the improved statement of (Chen et al., 2023) results in appendix A. All the changes are highlighted in red.
>
>
> >**Section 5 on acceleration could be deferred to the appendix or noted in passing, and more discussion given to the exponential terms in the bound, which are tantamount to saying that essentially no practical acceleration happens, even though the technical result itself is interesting to note. Similarly, the bounds arising in Section 6 should be discussed a bit more, because due to the central assumption of the paper, ultimately the pessimistic exponential terms in D arise.**
>
> Regarding Section 5, our bound for the accelerated method indeed follows from the pessimistic bound on the gradient norm, which leads to the effective smoothness given by the inequality (24). It actually means an accelerated convergence with a pessimistic bound on smoothness constant. Despite a challenge proving better bound due to the term (20), in experiments, we see $(L_0, L_1)$-STM shows a certain acceleration. We also conjecture that the term (20) can be bounded better. We believe this is an interesting open question, and we want to highlight it in the main part.
>
> Regarding Section 6, we also present a pessimistic bound (25) based on (24). Then, in lines 432-435, we point out that the constant in the bound can be huge and then provide the improved result, avoiding additional exponent. We believe that further discussion in lines 450-463 is quite comprehensive since it provides an example that the improved analysis can lead to a convergence rate up to 10^5 times faster; see line 451. However, we would greatly appreciate any suggestions from the reviewer to enhance this discussion.

---

> > ### Comment · Reviewer_jtK8 · 2024-11-25
> >
> > The reason I raised the flag about (Zhang et al 2020b) is to make sure that the narrative of the current paper is more accurate. In its current version (and even from the authors' reply here), the paper seems to claim greater novelty for dropping twice differentiability. The authors should be a bit more careful in citing related work precisely, otherwise, anybody else who reads this work (or people who read Chen et al) will believe that the possibility of handling only once differentiable functions was first considered by Chen et al, which seems to be not precise, even though, the credit for properly working things out, as well as introducing more refined notions of smoothness belongs to them.

---

> > ### Comment · Reviewer_jtK8 · 2024-11-25
> >
> > > We are not aware of any technique allowing trivially adapts the results of the nonconvex case to the convex case, but we believe that a trivial adaptation does not lead to tight results.
> >
> > The question is not about tight results, but for stronger positioning of the current paper it would be **valuable** to answer this question with mathematical precision. In particular, the zeroth order adaptation (that requires no work), is simply the stationarity result of the nonconvex case also applies to the convex one. The next step of adaptation would be good to put into the motivation part / related work part of the paper, so that is is clear how big a jump from those results is the present work (including, some simple, non-tight results that arise).

---

> ### Author Response · Authors · 2024-11-20
> **Respose to Reviewer jtK8: Part 2**
>
> >**The authors's motivation for studying AdGD should be as a result further enhanced: it seems inherently unsuitable to use under the generalized smoothness assumptions since AdGD does not utilize clipping.**
>
> Indeed,  Zhang et al. (2020) noticed that $(L_0, L_1)$-smoothness is sufficient for the study of clipping algorithms. Stepsize selection rule from Malitsky and Mishchenko (2019) states that
> $$\lambda_{k-1}\sqrt{1 + \frac{\lambda_{k-1}}{\lambda_{k-2}}} \geq \lambda_k \geq \frac{\gamma\|\|x^k-x^{k-1}\|\|}{\|\|\nabla f(x^{k})-\nabla f(x^{k-1})\|\|} \geq \frac{\gamma}{L_0 + L_1 \|\|\nabla f(x)\|\|},
> $$
> where the latter expression represents the form of a stepsize for Normalized Gradient Descent (NGD). Given that NGD is a very close method to Clipped-SGD (Zhang et al. 2019)[Section 3.1 GRADIENT DESCENT ALGORITHMS], the stepsize from Malitsky and Mishchenko (2019) can be interpreted as a form of adaptive clipping.
>
> Zhang, Jingzhao, et al. "Why gradient clipping accelerates training: A theoretical justification for adaptivity." arXiv preprint arXiv:1905.11881 (2019).
>
> Zhang, Bohang, et al. "Improved analysis of clipping algorithms for non-convex optimization." Advances in Neural Information Processing Systems 33 (2020): 15511-15521.
>
> >**And the result for the stochastic case (Section 7) requires a common minimizer for all the stochastic components (Assumption 4). Although such an assumption is also used in some works, this assumption is not that weak, and renders the results less applicable.**
>
> We partially agree with the reviewer that this assumption may be restrictive and give a proper discussion about this in lines 477-484. In line 479, we state that it is a typical assumption for over-parameterized models and provide a number of references. We would like to note that over-parameterized models have become increasingly common in modern applications, driven by the emergence of larger models like GPT, BERT, and their derivatives year after year, making overparameterization the norm. Supporting this, we highlight that over-parameterized networks are widely studied, as evidenced by the references provided below. However, we agree that the generalization that does not require the over-parameterization assumptions is an interesting direction for future research, but out of scope of our current work.
>
> Zhang, Guodong, James Martens, and Roger B. Grosse. "Fast convergence of natural gradient descent for over-parameterized neural networks." Advances in Neural Information Processing Systems 32 (2019).
>
> Kong, Zhifeng. "Convergence Analysis of Training Two-Hidden-Layer Partially Over-Parameterized ReLU Networks via Gradient Descent." International Journal of Computer and Information Engineering 14.6 (2020): 166-177.
>
> Li, Yuanzhi, and Yingyu Liang. "Learning overparameterized neural networks via stochastic gradient descent on structured data." Advances in neural information processing systems 31 (2018).
>
> Arora, Sanjeev, et al. "Fine-grained analysis of optimization and generalization for overparameterized two-layer neural networks." International Conference on Machine Learning. PMLR, 2019.
>
> Allen-Zhu, Zeyuan, Yuanzhi Li, and Yingyu Liang. "Learning and generalization in overparameterized neural networks, going beyond two layers." Advances in neural information processing systems 32 (2019).
>
>
> >**Can a version of the method that is somewhat more agnostic of the knowledge of L0 and L1 be designed, since right now this knowledge is quite necessary for selecting the correct step sizes?**
>
> In fact, the problem of knowing $L_0$ and $L_1$ is identical to the standard L-smooth case. It is enough to tune only one parameter $\hat L$, s.t. $\hat L \geq  L_0$ and $\hat L \geq  L_1$; in the standard case, we tune $\hat L \geq  L$. In other words, in practice, it is enough to take a sufficiently large constant $\hat L$. However, in the case of $(L_0, L_1)$- smoothness, the constants itself can be much smaller.
>
> Furthermore, we wish to highlight that GD-PS is agnostic to  $L_0$ and $L_1$, the only required parameter $f(x^\ast)$ is often known in many applications, such as overparameterized models. Even when $f(x^\ast)$ is unknown, there are techniques that can adapt to its value, e.g., see (Elan and Kakade, 2019).
>
> Hazan, Elad, and Sham Kakade. "Revisiting the Polyak step size." arXiv preprint arXiv:1905.00313 (2019).
>
> Finally, our result for AdGD of Malitsky and Mishchenko (2019) does not require any knowledge of any parameters while recovering the standard GD rates.

---

> ### Author Response · Authors · 2024-11-20
> **Respose to Reviewer jtK8: Part 3**
>
> >**Ultimately, a high-level takeaway of the work (which is likely also a takeaway that follows from the Chen paper if one could quickly bridge their stationarity to function suboptimality here) seems to be that the results are ultimately of a local nature. What I mean by that is that due to the unavoidable dependence of the type , the terms involving that in the bound will be large, unless is sufficiently small, and hence, in spirit the results can be viewed as showing that the GD methods studied in the paper are good but only locally. This aspect is not a criticism of the paper, but just a comment on how one can typically interpret bounds that involve exponentials.**
>
> We respectfully disagree with the reviewer that our result is of local nature. For $(L_0, L_1)$-GD we have a clear characterization of two-stage convergence and the leading term in the rate is independent of $L_1$ and any exponentially large terms (inequality (15)). Furthermore, for GD-PS we also show better results without requiring any change of the method. Finally, our analysis for AdGD does not involve these stages at all, and does not show local nature.
>
> >**Do the results really offer a significant improvement over Li et al. 2024a (Convex and Non-convex Optimization Under Generalized Smoothness). The results provided by Li et al. 2024a for GD and NAG also do not rely on the assumption of L-smoothness, and their acceleration results for NAG do not have a dependence on exp(L1). The authors claim that the advantage of their results is that they do not require dependencies on and , as in Li et al. 2024a, and they argue that these quantities could potentially be exponentially large in the worst case (Line 408-420)---But in the reviewer's opinion, assuming these initial quantities to be constants is not a very strong assumption, whereas in comparison, the authors' acceleration result depends on exp(L1), which seems to be less favorable.**
>
>
> We thoroughly discuss and explain in detail the improvements on the results of (Li et al., 2024a) in lines 135-154, 157-165, 286-290, 326-329, 386-398, 836-837. We added lines 221-224 for additional discussion.
> We would like to briefly summarize it here.
>
> - constant $\ell$ can be much larger than $L_0$ and $L_1$,
> - $\ell \sim L_0(1 + 2L_1R_0 \exp(L_1 R_0))$ in the worst case, and the derived complexity is not optimal
> - our bound does not depend on $f(x^0) - f(x^*)$ and on any bound for $\|\nabla f(x^k)\|$ including $\|\nabla f(x^0)\|$ which is of order $L_0R_0\exp(L_1 R_0)$
>
> Assuming initial quantities to be constants does not lead to a good bound. For example, trivial bound (24) reduces $(L_0, L_1)$-smoothness to the standard smoothness with a pessimistic constant, having double exponential of $L_1D$.
> We further improve the results of  (Li et al., 2024a) by removing the $L_1$ dependent factors from $\ell$ in the worst case, e.g.
> our bound for GD is $\mathcal O (\max [ \frac{L_0 R_0^2}{\varepsilon}, L_1^2R_0^2])$ vs. (Li et al., 2024a) which is $ \mathcal O (\frac{\ell R_0^2}{\varepsilon})$  where $\ell \sim L_0(1 + 2L_1R_0 \exp(L_1 R_0))$  in the worst-case. We improve the result by a factor of $1 + 2L_1R_0 \exp(L_1 R_0)$ which is significantly larger in the worst case.
>
> Regarding our results for the accelerated method, the reviewer fairly noticed an extra term $\exp(L_1D)$ in contrast to others' results. In fact (Li et al., 2024a) have the same issue; we have the discussion in lines 386-398. Our analysis faced a challenge bounding the term (20), discussed in Lines 371-377.
> However, in experiments we see that $(L_0, L_1)$ STM shows better performance than $(L_0, L_1)$-GD. And we believe that the term (20) can be bounded. We think that it would be an important contribution to bound the term, and decide to put this challenge into the main part.

---

> > ### Comment · Reviewer_jtK8 · 2024-11-25
> >
> > As I noted in the review, any bounds involving exponentials can be viewed **in spirit** as local results; the authors are free to disagree and interpret the bounds whichever way they want, I am just expressing here what I believe would be a **valid takeaway for anybody who wants to use the bound** for either understanding the algorithm, or for developing subsequent theory of their own.

---

> ### Comment · Area_Chair_MLFq · 2024-11-24
>
> Dear Reviewer jtK8,
>
> The author discussion phase will be ending soon. The authors have provided detailed responses. Could you please reply to the authors whether they have addressed your concern and whether you will keep or modify your assessment of this submission?
>
> Thanks.
>
> Area Chair

---

> ### Author Response · Authors · 2024-11-25
>
> We thank the reviewer for participating in the discussion with us. Below, we address the reviewer's remaining concerns/comments.
>
> >As I noted in the review, any bounds involving exponentials can be viewed in spirit as local results; the authors are free to disagree and interpret the bounds whichever way they want, I am just expressing here what I believe would be a valid takeaway for anybody who wants to use the bound for either understanding the algorithm, or for developing subsequent theory of their own.
>
> We respectfully disagree with the takeaway suggested by the reviewer. In particular, the results presented in our Theorems 3.1, 4.1, and 7.1 are **independent of any exponential terms** and **hold globally**, i.e., the starting point can be arbitrarily far from a solution. Therefore, it is incorrect to say that our results are local.
>
> >The reason I raised the flag about (Zhang et al 2020b) is to make sure that the narrative of the current paper is more accurate.  The authors should be a bit more careful in citing related work precisely, otherwise, anybody else who reads this work (or people who read Chen et al) will believe that the possibility of handling only once differentiable functions was first considered by Chen et al, which seems to be not precise, even though, the credit for properly working things out, as well as introducing more refined notions of smoothness belongs to them.
>
> We added the proper reference to the introduction. Please, see our revised version (the end of the second paragraph and footnote 3 on page 3).
>
> >In its current version (and even from the authors' reply here), the paper seems to claim greater novelty for dropping twice differentiability.
>
> We never claimed any novelty in dropping twice differentiability.
>
> >The question is not about tight results, but for stronger positioning of the current paper it would be valuable to answer this question with mathematical precision. In particular, the zeroth order adaptation (that requires no work), is simply the stationarity result of the nonconvex case also applies to the convex one. The next step of adaptation would be good to put into the motivation part / related work part of the paper, so that is is clear how big a jump from those results is the present work (including, some simple, non-tight results that arise).
>
> In the non-convex case, the convergence is characterized by the norm of the gradient, while in the convex case, the convergence is measured by a stronger criterion - function value sub-optimality, i.e., $f(x^N) - f(x^\ast)$. These two performance criteria are not directly comparable for convex problems, and a small gradient norm does not necessarily imply small function sub-optimality. To illustrate this, consider a Huber loss function $f:\mathbb{R} \to \mathbb{R}$ with parameter $\varepsilon$ defined as follows:
>
> $$
> f(x) = \begin{cases} \frac{1}{2}x^2,& \text{if } x\in [-\varepsilon, \varepsilon],\\\\ \varepsilon\left(|x| - \frac{1}{2}\varepsilon\right),& \text{otherwise}. \end{cases}
> $$
>
> This function is convex and $L$-smooth with $L=1$. Moreover, for any $x\in\mathbb{R}$ we have $|f'(x)| \leq \varepsilon$. However, the function sub-optimality $f(x) - f(x^\ast)$ is unbounded. Therefore, in general, analysis in the non-convex case cannot be easily adapted to the convex case. More precisely, the proofs from Chen et al. (2023) use inequality (7) as a starting point, while our proofs are based on the careful analysis of $\|\|x^k - x^\ast \|\|$. Moreover, Chen et al. (2023) consider normalized versions of GD and SPIDER, while we do not study normalization. Next, the analysis from Chen et al. (2023) does not consider cases when $\|\| \nabla f(x^k) \|\| \geq L_0 / L_1$ and $\|\| \nabla f(x^k) \|\| < L_0 / L_1$ separately and also their analysis does not show two-stage behavior of the considered methods. Therefore, our analysis cannot be seen as an extension of the analysis from Chen et al. (2023).
>
> **We believe that we addressed all the reviewer's concerns. Therefore, we kindly ask the reviewer to reconsider their score.**

---

> > ### Comment · Reviewer_jtK8 · 2024-11-26
> >
> > >> As I noted in the review, any bounds involving exponentials can be viewed in spirit as local results; the authors are free to disagree and interpret the bounds whichever way they want, I am just expressing here what I believe would be a valid takeaway for anybody who wants to use the bound for either understanding the algorithm, or for developing subsequent theory of their own.
> > > We respectfully disagree with the takeaway suggested by the reviewer. In particular, the results presented in our Theorems 3.1, 4.1, and 7.1 are independent of any exponential terms and hold globally, i.e., the starting point can be arbitrarily far from a solution. Therefore, it is incorrect to say that our results are local.
> >
> > I am confused as to why the authors are getting so defensive and refusing to read the review comment literally. Nowhere did I say that **all your results are local** --- all that I said was, *wherever there's an exponential dependence in the bound, that particular result can be viewed as being local in spirit* (obviously, because in a small enough region, the contribution of the exponential term will be small). Yet, the authors keep insisting on their not being local, which I find baffling.
> >
> > >In the non-convex case, the convergence is characterized by the norm of the gradient, while in the convex case, the convergence is measured by a stronger criterion - function value sub-optimality, i.e., $f(x^N) - f(x^\ast)$. These two performance criteria are directly comparable for convex problems, and a small gradient norm does not necessarily imply small function sub-optimality.
> >
> > Obviously, that was the whole question. Also, I'm not speaking of literally out of the box. But just consider the trivialmost setting, where one has from convexity: $f_k - f^* \le \langle f_k', x_k-x^*\rangle$, to which one can apply Cauchy-Schwarz, and bound in terms of $\|\nabla f(x_k)\|$ and $\|x_k-x^*\|$. Of course, getting a bound on the latter distance is not easy, otoh, it shows already that upto a *diameter term*, convexity translates a small gradient into a function suboptimality. Now, after working a bit harder, one should be able to make more thorough connections to Chen et al's work. For example, we know that the gradient of an L-smooth convex function is cocoercive, i.e., $\langle f'(x)-f'(y), x-y\rangle \ge \frac{1}{L}\|x-y\|^2$. Using this we can even directly control $\|x_k-x^*\|$ in terms of $\|f'(x_k)\|$, etc.
> >
> > All the points I made in the review were supposed to be helpful and constructive, it's a pity that the authors viewed them not as such, and doubled-down on repeating their claims. It is possible, I'm not seeing something the authors are seeing, but I feel they do need to address all the points carefully.

---

> > > ### Author Response · Authors · 2024-11-26
> > >
> > > >I am confused as to why the authors are getting so defensive and refusing to read the review comment literally. Nowhere did I say that all your results are local --- all that I said was, wherever there's an exponential dependence in the bound, that particular result can be viewed as being local in spirit (obviously, because in a small enough region, the contribution of the exponential term will be small). Yet, the authors keep insisting on their not being local, which I find baffling.
> > >
> > > We are afraid that there is a misunderstanding between us and the reviewer. To resolve this misunderstanding, we kindly ask the reviewer to provide a definition of "local result". In our responses, we relied on the standard definition: a result is called local if it holds only for initialization that is sufficiently close to the optimum. **According to this definition, all our results are not local**. Moreover, we kindly ask the reviewer to give precise references to the results the reviewer is referring to. We emphasize that in the original review and in the subsequent responses, the reviewer did not specify the results that are discussed. We also highlight that we are explicit in the results and all the discussions: whenever the result contains exponential factors, we show it explicitly (see Theorems 5.1 and 6.1 and discussions after them).
> > >
> > > >Obviously, that was the whole question. Also, I'm not speaking of literally out of the box. But just consider the trivialmost setting, where one has from convexity: $f_k - f^\ast \leq \langle f_k', x_k - x^\ast \rangle$, to which one can apply Cauchy-Schwarz, and bound in terms of $\|\| \nabla f(x_k) \|\|$ and $\|\| x_k - x^\ast \|\|$. Of course, getting a bound on the latter distance is not easy, otoh, it shows already that upto a diameter term, convexity translates a small gradient into a function suboptimality. Now, after working a bit harder, one should be able to make more thorough connections to Chen et al's work.
> > >
> > > If we continue the derivation as the reviewer suggested, assume that $\|\| x_k - x^\ast \|\| \leq D$, and apply Theorem 2 from Chen et al (2023) with $\alpha = 1$ and $\beta = 1$ (note that it is the only possible choice for $\beta$ when $\alpha=1$), then we get
> > > $$
> > > \frac{1}{N}\sum\limits_{k=0}^{N-1}(f(x_k) - f^\ast) \leq \frac{2L_0 D(f(x_0) - f^\ast)}{N\varepsilon} + \frac{D\varepsilon}{2}
> > > $$
> > > for normalized GD, where $\varepsilon > 0$. In the original paper, $\varepsilon$ is the target accuracy. However, one can optimize it in the above bound and get $\varepsilon_{\text{opt}} = 2\sqrt{\frac{L_0(f(x_0) - f^\ast)}{N}}$ and
> > > $$
> > > \frac{1}{N}\sum\limits_{k=0}^{N-1}(f(x_k) - f^\ast) \leq \frac{4D\sqrt{L_0(f(x_0) - f^\ast)}}{\sqrt{N}}.
> > > $$
> > > The above rate is $\sqrt{N}$-worse than the leading term in the results we obtained for $(L_0,L_1)$-GD and GD-PS. We believe that $1/\sqrt{N}$ differs significantly from $1/N$ rate.
> > >
> > > >For example, we know that the gradient of an L-smooth convex function is cocoercive, i.e.,  $\langle f'(x) - f'(y), x-y \rangle \geq \frac{1}{L}\|\| x-y\|\|^2$. Using this we can even directly control $\|\| x-y \|\|$ in terms of $\|\| \nabla f(x_k) \|\|$, etc.
> > >
> > > The inequality the reviewer is referring to is **coercivity**, and it is equivalent to **$\frac{1}{L}$-strong convexity** (see Theorem 2.1.9 from [1]). Coercivity does not follow from convexity and smoothness. **Cocoercivity** is $\langle \nabla f(x) - \nabla f(y), x-y \rangle \geq \frac{1}{L}\|\| \nabla f(x) - \nabla f(y) \|\|^2$, and it is indeed equivalent to convexity and smoothness (see Theorem 2.1.5 from [1]). Although it can be used to bound $\|\| x_k - x^\ast \|\|$ for smooth problems, it does not hold for $(L_0,L_1)$-smooth problems. Therefore, it cannot be directly applied in our case (though a variant of this inequality holds in our case as well -- see inequality (11) in our paper).
> > >
> > > >All the points I made in the review were supposed to be helpful and constructive, it's a pity that the authors viewed them not as such, and doubled-down on repeating their claims. It is possible, I'm not seeing something the authors are seeing, but I feel they do need to address all the points carefully.
> > >
> > > We never claimed that we do not see the reviewer's comment as helpful and constructive. In our responses, we carefully and respectfully addressed all the reviewer's comments: we highly value the reviewer's feedback and their time, and we are committed to applying all necessary changes that will improve our work according to the reviewer. **We kindly ask the reviewer to let us know about the comments that remain unaddressed.**
> > >
> > > ---
> > > References:
> > >
> > > [1] Yurii Nesterov. Introductory lectures on convex optimization: A basic course. 2004

---

> > > > ### Comment · Reviewer_jtK8 · 2024-11-26
> > > >
> > > > >We are afraid that there is a misunderstanding between us and the reviewer. To resolve this misunderstanding, we kindly ask the reviewer to provide a definition of "local result"
> > > >
> > > > I have repeated again and again that "in spirit" the can be interpreted as, and the reason I called it "in spirit" is that it is not literally, formally like a usual local result. For completeness, let me reiterate what I've been saying in each response, but with a concrete pointer. Lines 174 and 179 of the paper note in summary upper bounds that have the terms $\exp(L_1 R_0)$, and $\exp(L_1D)$, respectively. These terms are exponentially large, and thus unless $R_0$ is small, or $D$ is small, these terms will dominate. Having the need to have small $R_0$ and/or $D$ are what I am calling **local** in spirit. Of course, the authors can say `these are just constants', hence I did not formally call out the dependence to be local.
> > > >
> > > > **Side remark:** If we as an optimization community are a bit more critical about these constants, then one will see that actually these such style of bounds can be even more problematic, especially say if one is trying to make a strong statement in terms of bit complexity, but that is not the topic of this paper, so I did not raise it.
> > > >
> > > > >If we continue the derivation as the reviewer suggested, assume that $|| x_k - x^\ast || \leq D$, and apply Theorem 2 from Chen et al (2023) with $\alpha = 1$ and $\beta = 1$ (note that it is the only possible choice for $\beta$ when $\alpha=1$), then we get $$ \frac{1}{N}\sum\limits_{k=0}^{N-1}(f(x_k) - f^\ast) \leq \frac{2L_0 D(f(x_0) - f^\ast)}{N\varepsilon} + \frac{D\varepsilon}{2} $$ for normalized GD, where $\varepsilon > 0$.......
> > > >
> > > > Thank you, this precisely proves the point I was trying to make. Namely, as noted in my previous message, the **trivialmost** approach already gives some bound already, albeit not tight. But noting such a thing makes the paper's positioning stronger, and not weaker. Perhaps a less trivial approach may already give something a bit stronger, but perhaps to reach the final result of the present paper one has to work a bit harder. So my point was (in all comments so far), that the authors should comment precisely on how far does one get with Chen et al's nonconvex approach, and then use that to justify the need for the present paper to exist.
> > > >
> > > > >The inequality the reviewer is referring to is coercivity..
> > > >
> > > > Sorry, I made a typo when I wrote that, and you are right about the switched terminology. And yes, your (11) is a version of the **co-coercivity** -- and it itself is also **local in spirit** (unavoidably so, as you noted yourself in L198). However, the whole point of what I was trying to express there was that on top of mere convexity, another **simple approach** would be to use co-coercivity, to get a somewhat tighter bound. But I made a typo, which derailed the discussion a bit (and added some junk in my comment). Nevertheless, one tries to follow this route of extending gradient norm bounds using something beyond just convexity, one lands at co-coercivity as one choice, but to use co-coercivity one needs to then develop an analog of co-coercivity, which brings us to  your bound (11), and fits in with enhancing the narrative by connecting to Chen et al's work more directly.

---

> > > > > ### Author Response · Authors · 2024-11-28
> > > > >
> > > > > We thank the reviewer for the feedback.
> > > > >
> > > > > **Comparison with other works.** Following the reviewer's request, we updated the paper: in the revised version, we discuss the difference between our proofs and the ones from (Koloskova et al., 2023; Takezawa et al., 2024; Chen et al., 2023) in Appendix D.1.
> > > > >
> > > > > **Exponential factors.** We thank the reviewer for the clarification. However, in the current version, we explicitly show all exponential factors. We are afraid that mentioning locality in the non-conventional sense may confuse the reader. Therefore, we decided to keep the discussions related to the exponential terms as they are now, but we are open to suggestions from the reviewer regarding the improvement of these discussions.
> > > > >
> > > > > **If the reviewer has further comments/questions/concerns, we are committed to addressing them promptly. If all concerns are resolved, we kindly ask the reviewer to reconsider their score.**

---

> > > > > > ### Comment · Reviewer_jtK8 · 2024-12-02
> > > > > >
> > > > > > I hope the authors have also made their presentation of related work *more precise* as per the comments from my review and other reviews.
> > > > > >
> > > > > > And, I do not think that **acknowledging** that bounds involving $\exp(D)$ factors are *local in spirit* would confuse anybody. At least, that's the most obvious takeaway for me from those bounds. Pedagogically, such a statement could find home in a "Limitations / Discussion" section without hurting exposition, and while enhancing transparency.
> > > > > >
> > > > > > Thanks for plowing through the intense discussion this paper ended up raising (across all reviews).

---

> > > > > > > ### Author Response · Authors · 2024-12-03
> > > > > > >
> > > > > > > We thank the reviewer for the reply and for the suggestions regarding the improvement of the presentation. We promise to address these comments in the final version if our paper gets accepted. In particular:
> > > > > > >
> > > > > > > - In addition to the current clarifications in the introduction, we will elaborate on the contribution of Zhang et al. (2020a) in Section 1.3, i.e., we will add that Zhang et al. (2020a) provide a generalized version of $(L_0,L_1)$-smoothness.
> > > > > > >
> > > > > > > - We will add explicitly to the Conclusion section (Section 8) that our rates for $(L_0,L_1)$-STM and AdGD have exponential factors of $\exp(L_1\|\| x^0 - x^\ast \|\|)$ and $\exp(2L_1D)$ respectively, meaning that their influence is significant unless $L_1 = 0$ or the starting point is close enough to the solution, i.e., the results are local in spirit.
> > > > > > >
> > > > > > > These modifications will require a few lines of additional space to fit the main part into ten pages. We plan to get this extra space by merging the statements of Theorems 7.1 and 7.2 and making formulas (1) and (22) inline formulas.
> > > > > > >
> > > > > > > Overall, we believe that **the mentioned modifications are minor**: they change neither the flow of the paper nor the results. We believe we also addressed all other concerns raised by the reviewer. **Therefore, we kindly ask the reviewer to reconsider their score.**

---

### Official Review · Reviewer_pFMj · 2024-11-03

**Soundness:** 4
**Presentation:** 4
**Contribution:** 4
**Rating:** 8
**Confidence:** 4

**Summary:**

The paper focuses on convex $(L_0, L_1)$-smooth optimization and answers important open questions in this domain. In particular, new convergence rates are derived for the gradient method with smoothed clipping and Polyak stepsizes, improving existing results. The best-known convergence rate is derived for $(L_0, L_1)$-STM. Also, new results proved for AdGD and a stochastic case under the additional assumption on a shared minimizer. The statements in the paper are clear, and compared with existing results in the literature, the proofs are correct.

**Strengths:**

1. Paper improved existing convergence results for gradient methods with (smoothed) clipping and Polyak stepsizes.
2. New convergence results for AdGD under $(L_0, L_1)$-smooth assumption.
3. Proposed $(L_0, L_1)$-STM recovers the best-know convergence rate for accelerated methods without additional knowledge on $R_0$ and $f(x^0) - f^*$.

**Weaknesses:**

1. The paper is missing the conclusion and experiments sections. However, the experiments are provided in the appendix.
2. The assumptions for stochastic problem is restrictive; however, the derived results are new and better than previous ones.

**Questions:**

- In the last part of the equation (15), should it be N+1 - T instead of N+1?
- Why is it so crucial to choose $\gamma= 1/4$? In Theorem 6.1, to achieve a rate better than in (25)? Can it be relaxed?

---

> ### Author Response · Authors · 2024-11-20
> **Respose to Reviewer pFMj**
>
> We thank the reviewer for a very positive evaluation of our work. Below, we address the reviewer’s questions and comments.
>
> >**The paper is missing the conclusion and experiments sections. However, the experiments are provided in the appendix.**
>
> We thank the reviewer for the comment. Following the request, we moved the discussion of the related works on gradient clipping and Polyak stepsizes to Appendix A, moved Figure 1 to the main part, and added a conclusion to the paper. All changes are highlighted in red.
>
> >**The assumptions for stochastic problem is restrictive; however, the derived results are new and better than previous ones**
>
> Тhank you for the comment, but the result is indeed non-trivial. A generalization of deterministic results is challenging, since the analysis involves cases when either $\|\| \nabla f(x^k) \|\| \geq L_0 / L_1$ or $\|\| \nabla f(x^k) \|\| < L_0 / L_1$; for the stochastic case we need to consider the norm of stochastic gradient instead. Dealing with the norm of stochastic gradient instead is challenging due to the need to take the expectation (see lines 2025-2035). But we think our analysis of this challenge is already interesting and brings value. The generalization that does not require the over-parameterization assumptions is an interesting direction for future research.
>
> >**In the last part of the equation (15), should it be N+1 - T instead of N+1?**
>
> We believe that the bound is correct. We derive the final bound in (15) by maximizing it with respect to $T$; please, see the further details in Appendix D (lines 1258-1275).
>
> >**Why is it so crucial to choose $\gamma = 1/4$? In Theorem 6.1, to achieve a rate better than in (25)? Can it be relaxed?**
>
> We chose $\gamma = 1/2$ for the sake of simplicity and ease of presentation, but the original paper  (Malitsky & Mishchenko, 2020) also provides more complicated analysis allowing larger $\gamma$. We follow the simplicity motivation while setting $\gamma = 1/4$. We reduce $\gamma$ to get an extra term \frac{1}{2}\|\|x^{k+1}-x^k\|\|^2 appearing while introducing $\Sigma_{k+1} = \frac{1}{2} \sum_{i=0}^{k}\| \|x^{i+1} - x^{i}\|\|^2$ to the potential (lines 1663-1667).
> Having $\Sigma_{k+1}$ to be bounded for all $k$ is crucial for obtaining convergence. But any $\gamma <= 1/2 - \delta$, for any  $\delta > 0$ keeps the same results up to the $\delta$ factor in the final rate for the simple approach.
> We believe that larger stepsizes ($\gamma = 1/2$) lead to faster convergence, but $\gamma = 1/4$ gives a better tradeoff between practical and theoretical performance.

---

> > ### Comment · Area_Chair_MLFq · 2024-11-24
> >
> > Dear Reviewer pFMj,
> >
> > The author discussion phase will be ending soon. Could you please reply to the authors whether they have addressed your concern and whether you will keep or modify your assessment of this submission?
> >
> > Thanks.
> >
> > Area Chair

---

> > ### Comment · Reviewer_pFMj · 2024-11-24
> >
> > Thank you for your clarifications. I agree with the rebuttal and keep my initial rating.

---

### Official Review · Reviewer_RdU1 · 2024-11-04

**Soundness:** 3
**Presentation:** 3
**Contribution:** 1
**Rating:** 5
**Confidence:** 3

**Summary:**

This paper analyzes the iteration complexities of several algorithms targeted at convex $(L_0,L_1)$-smooth optimization. In comparison to previous works, the authors focus on a more refined analysis, including the elimination of dependence on the smoothness parameter $L$ (although it is not the dominant term) for variants of Gradient Descent, as well as improvements on the naive adaption of Adaptive Gradient Descent for generalized smoothness.

**Strengths:**

This paper covers existing methods for convex $(L_0,L_1)$-smooth optimization, specifically Gradient Descent with Smoothed Clipping and Polyak Stepsizes. It also provides analyses of Similar Triangles Methods and Adaptive Gradient Descent under the setting of generalized smoothness. The results give an overview of existing and adapted methods.

**Weaknesses:**

My major concern is about the significance of the results. To be specific, for the first two variants of Gradient Descent, this paper only improves the non-dominant $\mathcal{O}(\sqrt{1/\varepsilon})$ term of iteration complexity, which is inconsequential for small $\varepsilon$, i.e., when finding a solution with good quality. For the adapted versions of Similar Triangles Method and Adaptive Gradient Descent, the iteration complexity is in the form of $\mathcal{O}(\sqrt{L_0L_1A\exp(L_1A)A^2/\varepsilon^2})$, where $A$ equals either $R_0$ or $D$, which generally aligns with a specification of Li et al. (2024). Thus, the overall contribution in terms of novel theoretical guarantees appears quite limited to me at this stage.

**Questions:**

**Typos and minor suggestions:**

Line 203: Since you cite the conference version rather than the arXiv version of the paper, please refer to it as "Proposition 1".

Line 383-387 / Inequality (20-21): I suggest that the authors avoid breaking the entire inequality into two labels, which can cause ambiguity as seen in the second step of (73). Using an underbrace might be a better option.

Line 419: "$\varepsilon$-solution".

Line 1095: a multiplier of $\exp(\eta)$ is missing in the last term.

**Questions:**

* As I mentioned in the Weaknesses section, are there any practical examples (either theoretical or experimental) that can justify the importance of the *improved* complexities?
* For Theorem 3.1, it appears that you integrate the analyses from Li et al. (2024) and Koloskova et al. (2023), substituting the normalization term $\ell(G)$ in $\eta$ into something related to $\|\nabla f(x^k)\|$, so that the sequence enjoys the monotonic properties and can be analyzed in two cases. Could you elaborate on the intuition behind this approach?
* You mention new technical results for $(L_0,L_1)$-smooth functions in Section 1.3. Could you specify these results for reference?



Haochuan Li, Jian Qian, Yi Tian, Alexander Rakhlin, and Ali Jadbabaie. Convex and non-convex optimization under generalized smoothness. In Advances in Neural Information Processing Systems 36, 2024.

Anastasia Koloskova, Hadrien Hendrikx, and Sebastian U Stich. Revisiting gradient clipping: Stochastic bias and tight convergence guarantees. In Proceedings of the 40th International Conference on Machine Learning, 2023.

---

> ### Author Response · Authors · 2024-11-20
> **Respose to Reviewer RdU1: Part 1**
>
> >**My major concern is about the significance of the results. To be specific, for the first two variants of Gradient Descent, this paper only improves the non-dominant $\mathcal{O}(\sqrt{1/\varepsilon})$ term of iteration complexity, which is inconsequential for small $\varepsilon$, i.e., when finding a solution with good quality.**
>
> >**As I mentioned in the Weaknesses section, are there any practical examples (either theoretical or experimental) that can justify the importance of the improved complexities?**
>
> We emphasize that our results hold without $L$-smoothness assumption, while the existing results for Clipped GD and GD with Polyak Stepsize do require standard L-smoothness, and, in particular, the mentioned $\mathcal{O}(\sqrt{1/\varepsilon})$ term is proportional to $\sqrt{L}$. Of course, one can show that for small enough stepsize parameters, the methods do not escape the ball centered at the solution and have radius $\|\| x^0 - x^\ast \|\|$. However, the smoothness constant is dependent on the size of this ball. This means that the $\mathcal{O}(\sqrt{1/\varepsilon})$ term can be the leading one even for very small values of $\varepsilon$.
> To illustrate this, consider the function from Example 1.1: $f(x) = \|\| x \|\|^{2n}$. As we show in Appendix B, it is $(2n, 2n-1)$-smooth, but it is not smooth for $n \geq 2$ on the whole space. However, on the ball centered at $0$ and radius $\|\| x^0 \|\|$ the smoothness constant equals $2n(2n-1)\|\| x^0 \|\|^{2n-2}$. For example, if $n=5$ and $\|\| x^0 -x^\ast \|\| = 10$, then $L_0 = 10, L_1 = 9$ and the smoothness constant on the mentioned ball equals $L = 9\cdot 10^9$. Plugging these values in the bound $\mathcal{O}(\max\{L_0R_0^2/\varepsilon, \sqrt{R_0^4 LL_1^2/\varepsilon}\})$ term shown by (Koloskova et al., 2023; Takezawa et al., 2024), we observe that the second term is dominant even for small values of $\varepsilon = 10^{-6}$. In contrast, the leading term in our bound $\mathcal{O}(\max\{L_0R_0^2/\varepsilon, L_1^2 R_0^2\})$ is the first one and it is significantly smaller than $\mathcal{O}(\sqrt{R_0^4 LL_1^2/\varepsilon})$ term even for reasonably small values of $\varepsilon$. Moreover, as one can see from Lemma 2.1, the smoothness constant on the described ball is proportional to $\exp(L_1\|\| x^0 - x^\ast \|\|)$ in the worst case, i.e., it can be exponentially large.
> We also highlight that our result for $(L_0, L_1)$-GD provides a refined characterization of the method’s behavior: we prove that the norm of the gradient is non-increasing and that the method’s convergence has two clear phases (large gradient and small gradient regimes). Previous results do not provide such detailed characterization.
> Considering all of these aspects, we believe that our results make a noticeable and significant improvement over the previous ones.
>
> >**For the adapted versions of Similar Triangles Method and Adaptive Gradient Descent, the iteration complexity is in the form of $\mathcal{O}(\sqrt{L_0L_1A\exp(L_1A)A^2/\varepsilon^2})$, where $A$ equals either $R_0$ or $D$, which generally aligns with a specification of Li et al. (2024).**
>
> Indeed, as we explain in lines 409-416, our result for $(L_0, L_1)$-STM matches the result derived by Li et al. (2024) in the case of $(L_0, L_1)$-smooth problems, though we consider a different method, which is of the interest on its own. However, Adaptive Gradient Descent (AdGD) is not considered by Li et al. (2024), and, in particular, our Theorem 6.1 gives the result of a different form. To achieve this, we modify the proof of AdGD significantly – see the proof of Theorem G.2 in Appendix G.2. This new approach to the analysis of AdGD allows us to show a better bound than the one that directly follows from $(L_0, L_1)$-smoothness and existing analysis by Malitsky & Mishchenko (2020) (see more details in lines 424-450 and the discussion after Theorem 6.1).
>
> >**Thus, the overall contribution in terms of novel theoretical guarantees appears quite limited to me at this stage.**
>
> We hope that our above responses clarify the novelty of the derived results. Moreover, we also highlight that in Section 7, we provide the results for the stochastic extensions of $(L_0, L_1)$-GD and GD-PS. These results are novel, and the proof technique is also also new for stochastic optimization, e.g., see lines 2004-2051 – we are unaware of similar tricks used in prior works on stochastic optimization.
>
> >**You mention new technical results for $(L_0,L_1)$-smooth functions in Section 1.3. Could you specify these results for reference?**
>
> Lemma 2.2 is new and provides several useful inequalities. In particular, we are not aware of analogs of inequality (10) in the literature on generalized smoothness. Moreover, in the revised version, we expanded the discussion about the relation to the known analogs of inequalities (8) and (11) from (Koloskova et al., 2023; Li et al., 2024a).

---

> ### Author Response · Authors · 2024-11-20
> **Respose to Reviewer RdU1: Part 2**
>
> >**For Theorem 3.1, it appears that you integrate the analyses from Li et al. (2024) and Koloskova et al. (2023), substituting the normalization term $\ell(G)$ in $\eta$ into something related to $\|\|\nabla f(x^k)\|\|$, so that the sequence enjoys the monotonic properties and can be analyzed in two cases. Could you elaborate on the intuition behind this approach?**
>
> Our proof follows the one from (Koloskova et al., 2023), which follows the analysis of standard GD. Then, similarly to (Koloskova et al., 2023), we consider two possible situations: either $\|\| \nabla f(x^k) \|\| \geq L_0 / L_1$ or $\|\| \nabla f(x^k) \|\| < L_0 / L_1$. In the second case, the gradient is small, and $L_1$-term in $(L_0, L_1)$-smoothness is dominated by $L_0$-term, i.e., in inequality (8), the denominator of the left-hand side is $\mathcal{O}(L_0)$. In this case, the method behaves as standard GD for $L$-smooth problems with $L = \mathcal{O}(L_0)$. However, when $\|\| \nabla f(x^k) \|\| \geq L_0/L_1$, the $L_1$-term in $(L_0, L_1)$-smoothness is the leading one and this is the crucial difference between our proof and the one obtained by Koloskova et al. (2023): to handle this case, we use (8) and show that such situations lead to the decrease of $\|\| x^{k+1} - x^\ast \|\|^2$ by some positive constant $\nu\eta / (8L_1^2)$. In contrast, Koloskova et al. (2023) use traditional smoothness, which leads to worse complexity, as we explained in our first response. Moreover, Lemma 3.2 shows that the gradient norm is non-increasing along the trajectory of the method – similar to standard GD (e.g., see Lemma C.3 from [1]).
>
> Next, Li et al. (2024a) show that under $(r,\ell)$-smoothness, the function is locally smooth, and thus, the standard analysis of GD for smooth problems can be applied under the assumption that the stepsize is sufficiently small. However, this approach leads to a worse dependency on $L_0$ and $L_1$ than we have in our bounds for $(L_0, L_1)$-GD, as we explain in lines 147-153.
>
>
> >**Typos and minor suggestions**
>
> We thank the reviewer for spotting the typos and for the suggestions. We incorporated all of them in the revised version. All the changes are highlighted in red.
>
> —
> References
>
> [1] Gorbunov et al. Extragradient Method: O (1/K) Last-Iterate Convergence for Monotone Variational Inequalities and Connections With Cocoercivity. AISTATS 2022

---

> > ### Comment · Reviewer_RdU1 · 2024-11-25
> >
> > Thank you for your response, and apologies for my late reply. I have four follow-up remarks that follows.
> >
> > > **The importance of improved complexities.**
> >
> > I generally agree with your point that the assumption of $L$-smoothness is somewhat undesirable for analyses under generalized smoothness, even though constant $L$ only appears in the non-dominant term of the iteration complexity. The example you raised is a bit extreme, since loss functions are usually not in the form of powers. The real issue is that $L$ can be thousands of times larger than $L_0$ and $L_1$ in practice. Nonetheless, it is apparently better to explicitly remove the dependence on $L$, and I recognize your efforts in this direction.
> >
> > > **The complexities of accelerated methods.**
> >
> > As you remarked after Theorem 3.1, the result of Li et al. (2024) is not satisfactory due to its dependence on $\Vert\nabla f(x^0)\|$, which scales up to $L_0 R_0 \exp(L_1 R_0)$ according to Lemma 2.1. The current complexities of accelerated methods still leave an open question regarding the possibility of a more refined results (without exponential dependence) that makes the current results less strong, although the remaining question may be highly non-trivial. I think your modification of AdGD is interesting, yet the result still does not make an improvement over the best-known complexity.
> >
> > > **Interpretation of Lemma 2.2.**
> >
> > After further checking the revised version and comparing it with the literature, I now understand that Lemma 2.2 has not been previously introduced. However, I must note that the difference is not particularly significant. As you mentioned, Equation (8) differs from Koloskova et al. (2023) due to the difference in the definition of $(L_0,L_1)$-smoothness. Similarly, cocoercivity and monotony of the gradient have been previously introduced (Li et al., 2023), and thus your Equation (10) seems like a natural extension. Moreover, since the requirement in Equation (9) is equivalent to $\Vert x-y\Vert\leq \nu/L_1$, your Equation (10) and (11) still utilize local properties, which is similar to prior studies.
> >
> > > **Overall contribution.**
> >
> > In my opinion, the paper is generally good in presentation and soundness, but the significance of results is still marginal in terms of novel bounds. Indeed, the authors provide a substantial amount of analysis for different algorithms and extensions, and I appreciate your efforts. If the open problem that you raised is addressed, I will certainly be inclined to accept the paper. However, the present contribution still does not convince me to assign a higher rating.

---

> > > ### Author Response · Authors · 2024-11-28
> > >
> > > We thank the reviewer for the response and the acknowledgment of the importance of removing the dependence on $L$ from the bounds. Below, we would like to provide further clarifications.
> > >
> > > **Complexity bounds for $(L_0,L_1)$-STM and AdGD.** We agree with the reviewer (and also explicitly mentioned in the text) that the tightness of the derived bounds for $(L_0,L_1)$-STM and AdGD is an open question. However, to the best of our knowledge, this question was not raised in prior works. Though Li et al. (2024) also derive accelerated rates (for a different method), they do not raise the question of the optimality of the derived results in the case of $(L_0,L_1)$-smoothness. Moreover, we are not aware of any prior convergence results for AdGD or any other parameter-free method under $(L_0,L_1)$-smoothness. Our analysis of $(L_0,L_1)$-STM and AdGD illustrates the importance and non-triviality of the open questions we formulated in the text. We believe such observations are also important for the community. We also highlight that we see the results from Sections 3, 4, and 7 as the main results of this paper, so our contribution is not limited to formulating those open questions.
> > >
> > > **Lemma 2.2.** This lemma indeed resembles existing results. However, we do not see this lemma as the main contribution of our paper.
> > >
> > > **Contributions.** We would like to highlight the significance of the derived results.
> > >
> > > - The main results of our paper are Theorems 3.1 and 4.1. As the reviewer acknowledged, these results remove the dependence on $L$ from the bounds (and also the need to assume $L$-smoothness), which is an important improvement. In the revised version of our paper, we also explain the technical differences between our proofs and existing ones (see Appendix D.1). We believe that presented bounds and proof techniques are sufficiently novel given the importance of the considered setup (generalized smoothness) and the importance of deriving tight convergence bounds without unnecessary assumptions, which is one of the ultimate goals in Optimization.
> > >
> > > - The results of Theorems 7.1 and 7.2 are new and provide important generalizations of the derived bounds for $(L_0,L_1)$-GD and GD-PS to the stochastic case. We also emphasize that our proofs are quite non-standard (see lines 2004-2051 – we are not aware of similar tricks used in prior works on stochastic optimization).
> > >
> > > - Although we agree with the limitations of our results for $(L_0,L_1)$-STM and AdGD, our analysis of these methods and the fact that we brought the attention of the community to the open questions about the accelerated and parameter-free methods for $(L_0,L_1)$-smooth problems might be very important and useful for the community.
> > >
> > > **We kindly ask the reviewer to provide their opinion on the above clarifications. If the reviewer has any further concerns/questions, we are happy to address them as soon as possible.**

---

> ### Comment · Area_Chair_MLFq · 2024-11-24
>
> Dear  Reviewer RdU1,
>
> The author discussion phase will be ending soon. The authors have provided detailed responses. Could you please reply to the authors whether they have addressed your concern and whether you will keep or modify your assessment of this submission?
>
> Thanks.
>
> Area Chair

---

### Author Response · Authors · 2024-11-20
**General comment to all reviewers**

We thank the reviewers for their feedback and time. In particular, we appreciate that the reviewers acknowledged the multiple strengths of our work. In particular, they write that

The paper:
- is well-written (Reviewer 4rMB)
- makes a technical contribution (Reviewer jtK8)
- improves and derives fine-grained existing convergence results (Reviewer jtK8and pFMj)
- provides new convergence results for AdGD (Reviewer pFMj)

We have updated our manuscript, **highlighting the changes in red**. Additionally, we have addressed the reviewers' questions, comments, and concerns in individual responses, **referencing line numbers from the revised manuscript**. We remain committed to promptly addressing any further questions or comments and are happy to engage in back-and-forth discussions as needed.

---

### Meta-Review · Area_Chair_MLFq · 2024-12-08

**Metareview:**

This paper studies the optimization algorithm for strongly convex and (L0,L1)-smooth functions. They derived the improved convergence rates for multiple algorithm including Gradient Descent with Gradient Clipping, Gradient Descent with Polyak stepsizes, and Adaptive Gradient Descent. This work fills a gap in the analysis of convex (L0,L1)-smooth functions. The main weakness is that there are not many practical examples that are  (L0,L1)-smooth but not C2.

**Additional Comments On Reviewer Discussion:**

I gave a lower weight to Reviewer pFMj who gave score 8 to this paper. This is because the review of Reviewer pFMj does not provide enough details, especially on the technical parts.

Reviewer jtK8 and the authors had a long and back-and-forth debate over if an exponential factor in the complexity should be called “local” result. In my opinion, no matter who is correct, this is really a minor issue and is just about how to explain this factor. Here, Reviewer jtK8 does not criticize on the value of contribution but just would like the authors to fairly claim their contribution and better position this work. I think no reviewers raised any critical issues. Two reviewers increased their scores after rebuttal.

---

### Decision · Program_Chairs · 2025-01-22

Accept (Poster)